# TAMING SCORE-BASED DENOISERS IN ADMM: A CONVERGENT PLUG-AND-PLAY FRAMEWORK

**Rajesh Shrestha**
School of EECS
Oregon State University
shresthr@oregonstate.edu

**Xiao Fu**
School of EECS
Oregon State University
xiao.fu@oregonstate.edu

## ABSTRACT

While score-based generative models have emerged as powerful priors for solving inverse problems, directly integrating them into optimization algorithms such as ADMM remains nontrivial. Two central challenges arise: i) the mismatch between the noisy data manifolds used to train the score functions and the geometry of ADMM iterates, especially due to the influence of dual variables, and ii) the lack of convergence understanding when ADMM is equipped with score-based denoisers. To address the manifold mismatch issue, we propose ADMM plug-and-play (ADMM-PnP) with the AC-DC denoiser, a new framework that embeds a three-stage denoiser into ADMM: (1) auto-correction (AC) via additive Gaussian noise, (2) directional correction (DC) using conditional Langevin dynamics, and (3) score-based denoising. In terms of convergence, we establish two results: first, under proper denoiser parameters, each ADMM iteration is a weakly nonexpansive operator, ensuring high-probability fixed-point *ball convergence* using a constant step size; second, under more relaxed conditions, the AC-DC denoiser is a bounded denoiser, which leads to convergence under an adaptive step size schedule. Experiments on a range of inverse problems demonstrate that our method consistently improves solution quality over a variety of baselines. Source code is publicly available at https://github.com/rajeshshrestha/ACDC.git.

## 1 INTRODUCTION

Inverse problems arise in many fields, including medical imaging (Song et al., 2022b; Jin et al., 2017; Arridge, 1999), remote sensing (Entekhabi et al., 1994; Combal et al., 2003), oceanography (Bennett, 1992), and computational physics (Raissi et al., 2019; Tarantola, 2005). Their solutions typically rely on incorporating prior knowledge or structural assumptions about the target signals, either through explicit regularization or data-driven models.

Classical approaches to inverse problems often rely on handcrafted regularizers, such as the $\ell_1$ norm for sparsity (Yang et al., 2010; Elad & Aharon, 2006; Dabov et al., 2007) and the nuclear norm for low-rank structure (Semerci et al., 2014; Hu et al., 2017). Deep learning introduced a new paradigm of using learned generative models—VAEs, GANs, and normalizing flows—as data-driven regularizers (Ulyanov et al., 2020; Alkhouri et al., 2024; Shah & Hegde, 2018), offering more expressive priors by capturing complex distributions. More recently, pre-trained score functions from diffusion models have gained attention for inverse problems (Song et al., 2022b; Chung et al., 2023), as they effectively approximate data distributions and align solutions with the underlying data geometry (Xiao et al., 2022).

The use of pre-trained score functions in inverse problems mainly falls into two categories. The first modifies the MCMC process of diffusion sampling to incorporate observation information, as in DPS (Chung et al., 2023) and DDRM (Kawar et al., 2022), where observations guide unconditional score functions to perform posterior sampling. The second integrates score functions into deterministic or stochastic optimization algorithms; for example, Wang et al. (2024) and Song et al. (2023) use them as "projectors" to keep iterates on the desired data manifold. Furthermore, building on Tweedie's lemma, which links score functions to signal denoising, works such as (Zhu et al., 2023; Mardani et al., 2024; Renaud et al., 2024b) employ score as denoisers in proximal-gradient-like steps.

**Challenges.** The works in (Zhu et al., 2023; Mardani et al., 2024; Li et al., 2024) present flexible "plug-and-play (PnP)" paradigms that integrate diffusion models with optimization algorithms. However, two challenges remain in this line of work. First, score functions are trained on noisy data manifolds constructed via Gaussian perturbations, whereas optimization iterates need not lie on such manifolds, leading to geometry mismatch and degraded denoising performance. Remedies such as stochastic regularization (Renaud et al., 2024b) or purification (Li et al., 2024) add Gaussian noise to the iterates, but this does not guarantee alignment with the score manifolds. Second, the theoretical understanding of these methods—particularly their convergence properties when combining the score denoisers with various optimization paradigms—remains limited.

**Contributions.** We propose to integrate score-based denoisers with the ADMM framework. Using ADMM iterates with score-based denoising is particularly challenging, as the presence of dual variables further distorts the "noise" geometry—likely explaining why score-based denoising has rarely been combined with primal–dual methods. Nevertheless, ADMM remains attractive for its flexibility in handling diverse inverse problems with multiple regularizers. Our contributions are:

▶ *Score-Based AC-DC Denoiser*: To mitigate the manifold mismatches, we propose a three-stage denoiser consisting of (1) additive Gaussian noise *auto-correction* (AC), (2) conditional Langevin dynamics-based *directional correction* (DC), and (3) score-based denoising. The AC stage pulls ADMM iterates toward neighborhoods of noise-trained manifolds, while DC refines alignment without losing signal information. This combination balances efficiency and accuracy, making score-based denoising effective within ADMM.

▶ *Convergence Analysis*: We show that, under proper AC-DC parameters, each ADMM iteration is weakly nonexpansive, ensuring convergence to a fixed-point neighborhood under constant step sizes under strongly convex losses. We further relax convexity and prove that an adaptive step-size scheme (Chan et al., 2016) guarantees convergence with high probability. These results extend prior ADMM-PnP convergence theory (Ryu et al., 2019; Chan et al., 2016) to score-based settings.

Our method is validated on diverse applications—including inpainting, phase retrieval, Gaussian and motion deblurring, super-resolution, and high dynamic range (HDR).

**Notation.** The detailed notation designation is listed in Appendix A.1.

## 2 BACKGROUND

**Inverse Problems.** We consider the typical inverse problem setting where

$$\boldsymbol{y} = \mathcal{A}(\boldsymbol{x}) + \boldsymbol{\xi} \tag{1}$$

where $\mathcal{A} : \mathbb{R}^d \to \mathbb{R}^n$ is the *measurement operator*, $n \leq d$, and $\boldsymbol{\xi}$ is additive noise. In some inverse problems, e.g., signal denoising and image deblurring, we have $n = d$; while for some other problems, e.g., data compression and recovery, we have $n < d$. The goal is to recover $\boldsymbol{x}$ from the $\boldsymbol{y}$, with the knowledge of $\mathcal{A}$. Structural regularization on $\boldsymbol{x}$ is often used to underpin the desired solution:

$$\min_{\boldsymbol{x}} \ \ell(\boldsymbol{y} \, \| \, \mathcal{A}(\boldsymbol{x})) + h(\boldsymbol{x}), \tag{2}$$

where $\ell(\boldsymbol{y} \, \| \, \mathcal{A}(\boldsymbol{x}))$ is a divergence term that measures the similarity of $\boldsymbol{y}$ and $\mathcal{A}(\boldsymbol{x})$ (e.g., $\|\boldsymbol{y} - \mathcal{A}(\boldsymbol{x})\|^2$), and $h(\boldsymbol{x})$ is a structural regularization term (e.g., $\|\boldsymbol{x}\|_1$ for sparse $\boldsymbol{x}$).

**Diffusion-Based Inverse Problem Solving.** Diffusion models can also be used for solving inverse problems, in ways more subtle than direct regularization. Consider training a diffusion model on $\boldsymbol{x}_0 \sim p_{\text{data}}$ via *denoising score matching* (Song et al., 2021), where the forward process is $\boldsymbol{x}_t | \boldsymbol{x}_0 \sim \mathcal{N}(\boldsymbol{x}_0, \sigma(t)\boldsymbol{I}), \quad t \in [0, T]$, with variance schedule $\sigma(0) = 0$ and $\sigma(t)$ increasing in $t$. After training, the model provides a score function $\boldsymbol{s}_{\boldsymbol{\theta}}(\boldsymbol{x}, \sigma(t)) \approx \nabla_{\boldsymbol{x}_t} \log p(\boldsymbol{x}_t)$, where $p(\boldsymbol{x}_t)$ is the marginal density of $\boldsymbol{x}_t$. This induces noisy data manifolds

$$\mathcal{M}_{\sigma(t)} = \text{supp}(\boldsymbol{x}_t), \quad \forall t \in [T],$$

which are continuous since $\boldsymbol{x}_t$ is generated by Gaussian perturbations of $\boldsymbol{x}_0 \sim p_{\text{data}}$. These score functions can then be leveraged in different ways to assist inverse problem solving.

▶ *Posterior Sampling*: Many works formulate inverse problems as posterior sampling from $p(\boldsymbol{x}|\boldsymbol{y}) \propto p(\boldsymbol{x})p(\boldsymbol{y}|\boldsymbol{x})$. These methods approximate $\nabla_{\boldsymbol{x}_t} \log p(\boldsymbol{y}|\boldsymbol{x}_t)$ and combine it with the learned score

$s_{\boldsymbol{\theta}}(\boldsymbol{x}_t, \sigma(t)) \approx \nabla \log p(\boldsymbol{x}_t)$ to perform stochastic sampling as $t$ decreases (Chung et al., 2023; Song et al., 2022a; Kawar et al., 2021; 2022; Wang et al., 2023). While effective, their performance is often limited by the accuracy of the approximation to $\nabla \log p(\boldsymbol{y}|\boldsymbol{x}_t)$.

▶ *Plug-and-Play (PnP) Approaches*: Instead of sampling schemes, another line of work employs deterministic or stochastic optimization to solve (2), plugging score functions into the updates as structural regularizers. A representative example is DiffPIR (Zhu et al., 2023), which adopts a variable-splitting reformulation, $\min_{\boldsymbol{x},\boldsymbol{z}} \ell(\boldsymbol{y}\|\mathcal{A}(\boldsymbol{x})) + \frac{\mu}{2}\|\boldsymbol{x} - \boldsymbol{z}\|_2^2 + h(\boldsymbol{z})$, where the $\boldsymbol{z}$-subproblem at iteration $k$ reduces to a standard denoising step:

$$\textbf{Denoising: } \boldsymbol{z}^{(k+1)} = \arg\min_{\boldsymbol{z}} \frac{\mu}{2}\left\|\boldsymbol{x}^{(k+1)} - \boldsymbol{z}\right\|_2^2 + h(\boldsymbol{z}). \tag{3}$$

This step is then tackled using a score-based denoiser:

$$\boldsymbol{z}^{(k+1)} \leftarrow D_{\sigma^{(k)}}(\widetilde{\boldsymbol{x}}^{(k+1)}) = \widetilde{\boldsymbol{x}}^{(k+1)} + (\sigma^{(k)})^2 \boldsymbol{s}_{\boldsymbol{\theta}}(\widetilde{\boldsymbol{x}}^{(k+1)}, \sigma^{(k)}) \tag{4}$$

where $\widetilde{\boldsymbol{x}}^{(k+1)} = \boldsymbol{x}^{(k+1)} + \sigma^{(k)}(\zeta(\widetilde{\boldsymbol{x}}^{(k)} - \boldsymbol{x}^{(k+1)})/\sigma^{(k)} + (1-\zeta)\boldsymbol{n})$ with $\boldsymbol{n} \sim \mathcal{N}(\boldsymbol{0}, \boldsymbol{I})$ and $\zeta \in [0, 1]$; also see (Li et al., 2024) for a similar method. These denoisers are designed following the *Tweedie's lemma* (Robbins, 1992) (see Appendix A). The construction of $\widetilde{\boldsymbol{x}}^{(k)}$ is meant to make the inputs to the score function closer to a certain $\mathcal{M}_{\sigma^{(k)}}$. Using score-based denoising, $h(\boldsymbol{z})$ is implicitly reflected in the denoising process and thus does not need to be specified analytically.

Another line of approaches explicitly construct regularizers $h(\boldsymbol{x})$ whose gradients correspond to applying the score function. Examples include RED-diff (Mardani et al., 2024) and SNORE (Renaud et al., 2024a). In SNORE, the regularizer is defined as $h(\boldsymbol{x}) = \mathbb{E}_{\widetilde{\boldsymbol{x}}|\boldsymbol{x}} \log p_\sigma(\widetilde{\boldsymbol{x}})$, $\widetilde{\boldsymbol{x}} \leftarrow \boldsymbol{x} + \sigma\boldsymbol{\epsilon}$, $\boldsymbol{\epsilon} \sim \mathcal{N}(\boldsymbol{0}, \boldsymbol{I})$ and taking its gradient in (2) yields:

$$\textbf{SNORE update: } \boldsymbol{x}^{(k+1)} \leftarrow \boldsymbol{x}^{(k)} - \delta\nabla\ell(\boldsymbol{y}\|\mathcal{A}(\boldsymbol{x}^{(k)})) - \eta(\widetilde{\boldsymbol{x}}^{(k)} - D_{\sigma_i}(\widetilde{\boldsymbol{x}}^{(k)})), \tag{5}$$

where, again, $D_\sigma(\widetilde{\boldsymbol{x}}) = \widetilde{\boldsymbol{x}} + \sigma^2 \boldsymbol{s}_\theta(\widetilde{\boldsymbol{x}}, \sigma)$ by the Tweedie's lemma.

**Challenges — Manifold Mismatch and Convergence.** Score-based PnP methods face two main challenges. First, the score is not trained on algorithm-induced iterates (e.g., $\boldsymbol{x}^{(k+1)}$ in (3)). While both $\boldsymbol{x}_t$ and $\boldsymbol{x}^{(k)}$ can be seen as noisy versions of $\boldsymbol{x} \sim p_{\text{data}}$, $\boldsymbol{x}_t$ follows Gaussian noise whereas the distribution of $\boldsymbol{x}^{(k)}$ is unclear. Many works attempt to bridge this gap by injecting Gaussian noise before applying the score function (cf. $\widetilde{\boldsymbol{x}}^{(k+1)}$ in (3), $\widetilde{\boldsymbol{x}}^{(k)}$ in (5)), or by purification-based schemes (Nie et al., 2022; Alkhouri et al., 2023; Meng et al., 2022). Yet noise injection alone is insufficient, and overfitting to measurement noise remains an issue (Wang et al., 2024). Second, the understanding to convergence of score-based PnP remains limited. Unlike classical denoisers with established theory, the geometry mismatch above makes it unclear whether iterates stabilize or under what conditions convergence can be ensured. Existing analyses mostly cover primal algorithms (see, e.g., (Renaud et al., 2024a)). Primal–dual methods such as ADMM offer greater flexibility for handling multiple regularizers and constraints, but their convergence with score-based denoisers remains unclear, as dual variables further complicate the manifold geometry of the iterates.

## 3 PROPOSED APPROACH

In this section, we propose our score-based denoiser and embed it in the ADMM framework. While we focus on ADMM due to its flexibility, the denoiser can be plugged into any other proximal operator based schemes (e.g. proximal gradient or variable-splitting as in DiffPIR).

**Preliminaries of ADMM-PnP.** ADMM-based inverse problem solvers start by rewriting (2) as

$$\min_{\boldsymbol{x},\boldsymbol{z}} \ell(\boldsymbol{y}\|\mathcal{A}(\boldsymbol{x})) + \gamma h(\boldsymbol{z}) \text{ s.t. } \boldsymbol{x} = \boldsymbol{z} \tag{6}$$

The augmented Lagrangian of (6) is given by $L_\rho(\boldsymbol{x}, \boldsymbol{y}, \boldsymbol{\lambda}; \boldsymbol{y}) = \ell(\boldsymbol{y}\|\mathcal{A}(\boldsymbol{x})) + \gamma h(\boldsymbol{z}) + \boldsymbol{\lambda}^T(\boldsymbol{x} - \boldsymbol{z}) + \frac{\rho}{2}\|\boldsymbol{x} - \boldsymbol{z}\|_2^2$, where $\boldsymbol{\lambda}$ is the dual variable and $\rho > 0$ is the penalty parameter. At iteration $k$, ADMM updates are as follows:

$$\boldsymbol{x}^{(k+1)} = \arg\min_{\boldsymbol{x}} \frac{1}{\rho}\ell(\boldsymbol{y}\|\mathcal{A}(\boldsymbol{x})) + \frac{1}{2}\left\|\boldsymbol{x} - \boldsymbol{z}^{(k)} + \boldsymbol{u}^{(k)}\right\|_2^2 \tag{7a}$$

$$\boldsymbol{z}^{(k+1)} = \arg\min_{\boldsymbol{z}} \frac{\gamma}{\rho}h(\boldsymbol{z}) + \frac{1}{2}\left\|\boldsymbol{x}^{(k+1)} - \boldsymbol{z} + \boldsymbol{u}^{(k)}\right\|_2^2 \tag{7b}$$

$$\boldsymbol{u}^{(k+1)} = \boldsymbol{u}^{(k)} + (\boldsymbol{x}^{(k+1)} - \boldsymbol{z}^{(k+1)}) \tag{7c}$$

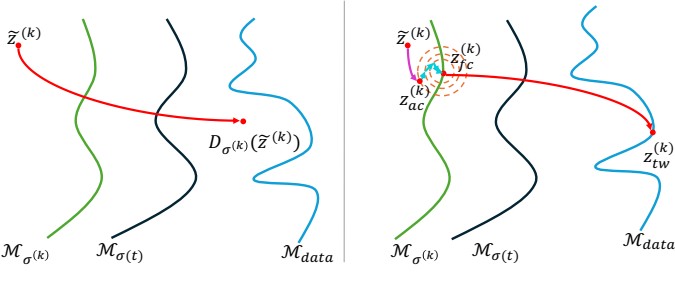

$s_\theta$ trained over data from $\{\mathcal{M}_{\sigma(t)}\}_{t=0}^T$

Figure 1: Left: direct denoising of $\widetilde{z}^{(k)}$ using score functions could lead to unnatural recovered signals with artifacts. Right: AC-DC denoising brings $\widetilde{z}^{(k)}$ closer to $\mathcal{M}_{\sigma^{(k)}}$, and then uses the score function to bring $\widetilde{z}^{(k)}$ to the data manifold $\mathcal{M}_{\text{data}}$.

---

**Algorithm 1** AC-DC Denoiser at iteration $k$ of ADMM in (7)

1: **auto correction (AC):** $\quad z_{\text{ac}}^{(k)} \leftarrow \widetilde{z}^{(k)} + \sigma^{(k)}n, \ n \sim \mathcal{N}(0, I)$
2: **directional correction (DC):**
3: $\quad w^{(k,0)} \leftarrow z_{\text{ac}}^{(k)}$
4: **for** $j = 0$ to $J - 1$ **do**
5: $\quad\quad w^{(k,j+1)} \leftarrow w^{(k,j)} + \eta^{(k)} \left( \frac{1}{\sigma_{s(k)}^2}(z_{\text{ac}}^{(k)} - w^{(k,j)}) + s_\theta(w^{(k,j)}, \sigma^{(k)}) \right) + \sqrt{2\eta^{(k)}}n,$
6: **end for**
7: $\quad z_{\text{dc}}^{(k)} \leftarrow w^{(k,J)}$
8: **Denoising** : $\quad z_{\text{tw}}^{(k)} \leftarrow \mathbb{E}[z_0|z_t = z_{\text{dc}}^{(k)}] = z_{\text{dc}}^{(k)} + (\sigma^{(k)})^2 s_\theta(z_{\text{dc}}^{(k)}, \sigma^{(k)})$ $\quad\quad$ (Tweedie Denoising)
$\quad$ (Alternative: $z_{\text{ode}}^{(k)} \leftarrow z_0$ by solving: $\frac{dz_t}{dt} = \lambda(t)s_\theta(z_t, t)$, with $z_{\sigma^{(k)}} = z_{\text{dc}}^{(k)}$) $\quad\quad$ (ODE Denoising)

---

where $u^{(k)} = \lambda^{(k)}/\rho$ is the scaled dual variable. The subproblem (7b) is a denoising problem, and thus can be replaced by

$$z^{(k+1)} = \text{Prox}_{\frac{\gamma}{\rho}h}(x^{(k+1)} + u^{(k)}) = D_{\sigma^{(k)}}(\widetilde{z}^{(k)}). \tag{8}$$

The above is the classical ADMM-PnP method; see (Chan et al., 2016; Ryu et al., 2019). Same as before, Eq. (8) can be replaced by the score-based denoising following the Tweedie's lemma. However, as $\widetilde{z}^{(k)} = x^{(k+1)} + u^{(k)}$ could be in any of the manifolds $\mathcal{M}_{\sigma(t)}$ on which the score function was trained, such naive replacement does not ensure effective denoising. The existence of the dual variable $u^{(k)}$ makes the noise distribution in $\widetilde{z}^{(k)}$ even harder to understand.

**Proposed Approach: The AC-DC Denoiser.** To address the manifold mismatch issues, we propose a three-stage denoiser. To be specific, in the $k$th iteration of the ADMM algorithm, we use the denoising process shown in Algorithm 1. Note that the Tweedie's lemma step (line 8) can also be substituted by a score ODE based process (Karras et al., 2022) initialized at $z_{\text{dc}}^{(k)}$. Our algorithm using these two different denoisers will be referred to as Ours-tweedie and Ours-ode, respectively.

The rationale of the AC-DC denoiser is illustrated in Fig.1. Recall that the score function is most effective on the noisy data manifolds $\{\mathcal{M}_{\sigma(t)}\}_{t=1}^T$, as it is trained over them. Since ADMM-induced iterates $\widetilde{z}^{(k)}$ need not lie on these manifolds, directly applying score-based denoising may be ineffective. The AC step addresses this by adding Gaussian noise, making $z_{\text{ac}}^{(k)}$ closer to some $\mathcal{M}_{\sigma(t)}$ (see AppendixB). This idea is related to the "purification" step in (Nie et al., 2022; Alkhouri et al., 2023; Li et al., 2024) and noise-added denoising in (Mardani et al., 2024; Renaud et al., 2024a; Zhu et al., 2023) (cf. (5) and (3)). However, AC alone does not guarantee manifold alignment. The proposed DC step, based on Langevin dynamics, further refines $z_{\text{dc}}^{(k)}$ toward $\mathcal{M}_{\sigma^{(k)}}$.

To see the idea, let us break down the three steps. First, the AC step gives

$$z_{\text{ac}}^{(k)} = z_{\sigma^{(k)}} + \tilde{s}^{(k)}, \quad s^{(k)} = \widetilde{z}^{(k)} - z_\natural^{(k)}, \ z_\natural^{(k)} \sim p_{\text{data}} \tag{9}$$

where $z_{\natural}^{(k)}$ is denoised signal of $\widetilde{z}^{(k)}$, $z_{\sigma^{(k)}} = z_{\natural}^{(k)} + \sigma^{(k)} n_1$, $\tilde{s}^{(k)} = \sqrt{2}\sigma^{(k)} n_2 + s^{(k)}$, and $n_1, n_2 \overset{\text{i.i.d}}{\sim} \mathcal{N}(\mathbf{0}, \boldsymbol{I}_d)$. Given a sufficiently large $\sigma^{(k)}$, $z_{\text{ac}}^{(k)}$ would have dominated by Gaussian noise—but not necessarily on any of $\mathcal{M}_{\sigma_t}$ where the score was trained. Starting from $z_{\text{ac}}^{(k)}$, the DC step runs a few iterations of *Langevin dynamics* targeting the distribution $p(z_{\sigma^{(k)}}|z_{\text{ac}}^{(k)})$. This is because $\text{supp}(z_{\sigma^{(k)}}|z_{\text{ac}}^{(k)}) \subseteq \text{supp}(z_{\sigma^{(k)}}) = \mathcal{M}_{\sigma^{(k)}}$. In addition, $p(z_{\sigma^{(k)}}|z_{\text{ac}}^{(k)})$ at the same time retains the information of $z_{\text{ac}}^{(k)}$ (thereby the information from the measurements). Assume that the forward process used in training the score has sufficiently small time intervals, $\mathcal{M}_{\sigma^{(k)}}$ is approximately contained in $\{\mathcal{M}_{\sigma_t}\}_{t=1}^T$. This way, when applying Tweedie's lemma for denoising, the step is expected to be effective, as the score was trained over $\{\mathcal{M}_{\sigma_t}\}_{t=1}^T$.

Note that the conditional score for the Langevin dynamics step can be expressed as follows

$$\nabla \log p(z_{\sigma^{(k)}}|z_{\text{ac}}^{(k)}) = s_{\boldsymbol{\theta}}(z_{\sigma^{(k)}}, \sigma^{(k)}) + \nabla \log p(z_{\text{ac}}^{(k)}|z_{\sigma^{(k)}}). \tag{10}$$

Ideally, one would use the exact $\nabla \log p(z_{\text{ac}}^{(k)}|z_{\sigma^{(k)}})$ for the DC step—which is unavailable. In practice, we approximate $p(z_{\text{ac}}^{(k)}|z_{\sigma^{(k)}})$ using a Gaussian distribution. Note that under proper scheduling of $\sigma^{(k)}$ and mild regularity conditions on $s^{(k)}$, e.g., when $\text{Var}(s^{(k)})^{1/2} \ll \sigma^{(k)}$, the likelihood can be well-approximated by a locally quadratic form, leading to $\nabla \log p(z_{\text{ac}}^{(k)}|z_{\sigma^{(k)}}) \approx -1/\sigma_{s^{(k)}}^2(z_{\sigma^{(k)}} - z_{\text{ac}}^{(k)})$ and the DC step in Algorithm 1.

## 4 CONVERGENCE ANALYSIS

### 4.1 CONVERGENCE OF UNDER WEAKLY NON-EXPANSIVE RESIDUALS

Following the established convention in ADMM-PnP, e.g., (Buzzard et al., 2018; Sun et al., 2019; Chan, 2019; Teodoro et al., 2017), we aim at understanding the convergence properties when the AC-DC denoiser is used. We will use the following definitions:

**Definition 1** (Fixed point convergence). *Let $T : \mathcal{X} \to \mathcal{X}$ be the update map of an iterative algorithm, and let $x^{(0)} \in \mathcal{X}$ be arbitrary initialization. The algorithm is said to converge to a fixed point $x^*$ if for any $\delta > 0$ there exists $K_\delta > 0$ such that the sequence generated by the algorithm $\{x^{(k)}\}_{k \in \mathbb{N}+}$ satisfies $\left\|x^{(k)} - x^*\right\|_2 < \delta$ for all $k \geq K_\delta$. Equivalently, $\lim_{k\to\infty} x^{(k)} = x^*$ with $T(x^*) = x^*$.*

**Definition 2** (Sequence convergence to a $\delta$-ball). *For a certain $\delta > 0$, a sequence $\{x^{(k)}\}_{k \in \mathbb{N}+}$ is said to converge within a $\delta$-ball if there exists $K > 0$ and $x^*$ such that the following holds for all $k \geq K$.*

$$\left\|x^{(k)} - x^*\right\|_2 \leq \delta \tag{11}$$

Comparing the two definitions, Definition 2 is a weaker statement; that is, even when $k \to \infty$, $x^{(k)} \to x^*$ does not necessarily happen. Nonetheless, convergence to a $\delta$-ball is still meaningful. The notion of $\delta$-ball convergence is often used in numerical analysis for stability characterization; see, e.g., Ren & Argyros (2021); Ren & Wu (2009); Liang (2007).

**Definition 3** (ADMM convergence to a $\delta$-ball). *ADMM is said to converge within a $\delta$ ball if the sequences $\{x^{(k)}\}_{k \in \mathbb{N}+}$ and $\{u^{(k)}\}_{k \in \mathbb{N}+}$ obtained from ADMM converges within a $\delta$-ball.*

To proceed, consider the following assumption:

**Assumption 1.** *For a certain $\delta > 0$ there exists $\epsilon \leq 1$ such that for all $\widetilde{z}_1, \widetilde{z}_2 \in \mathbb{R}^d$, the following holds:*

$$\|R_\sigma(\widetilde{z}_1) - R_\sigma(\widetilde{z}_2)\|_2^2 \leq \epsilon^2 \|\widetilde{z}_1 - \widetilde{z}_2\|_2^2 + \delta^2 \tag{12}$$

*where $R_\sigma(\widetilde{z}) = (D_{\sigma^{(k)}} - I)(\widetilde{z})$ with $I$ being the identity function (i.e., $I(z) = z$).*

Here, the notation $(D_{\sigma^{(k)}} - I)(\widetilde{z})$ denotes the residual of $D_{\sigma^{(k)}}$ i.e. $D_{\sigma^{(k)}}(\widetilde{z}) - \widetilde{z}$. The next theorem extends the fixed point convergence of ADMM-PnP in Ryu et al. (2019). Unlike Ryu et al. (2019) where $R_\sigma$ needs to be strictly contractive, our result allows $R_\sigma$ to be weakly contractive:

**Theorem 1.** *Under Assumption 1, assume that $\ell$ is $\mu$-strongly convex. Then, there exists $x^*$, $u^*$ and $K > 0$ such that the sequences $\{x^{(k)}\}_{k \in \mathbb{N}+}$ and $\{u^{(k)}\}_{k \in \mathbb{N}+}$ generated by ADMM-PnP using a fixed*

*step size $\rho$ satisfies $\left\|\boldsymbol{u}^{(k)} - \boldsymbol{u}^*\right\|_2 \leq r$ and $\left\|\boldsymbol{x}^{(k)} - \boldsymbol{x}^*\right\|_2 \leq r$ with $r = (1 + \frac{\rho}{\rho+\mu})\bar{\delta}/\sqrt{1 - \bar{\epsilon}^2}$ for all $k \geq K$ when $\epsilon/\mu(1+\epsilon-2\epsilon^2) < 1/\rho$ where $\bar{\delta}^2 = \frac{\delta^2\bar{\epsilon}}{\epsilon}$ and $\bar{\epsilon} = \frac{\rho+\rho\epsilon+\mu\epsilon+2\mu\epsilon^2}{\rho+\mu+2\mu\epsilon}$.*

The proof is relegated to Appendix C. Note that when $\delta = 0$, it implies the result in Ryu et al. (2019).

## 4.2 CONVERGENCE UNDER WEAKLY NON-EXPANSIVENESS WITH AC-DC

In this subsection, we will show that the AC-DC denoiser satisfies Assumption 1 under mild conditions. To this end, consider the following:

**Assumption 2** (Smoothness of $\log p_{\text{data}}$). *The log data density $\log p_{\text{data}}$ is $M$-smooth for a constant $M > 0$, i.e., $\|\nabla \log p_{\text{data}}(\boldsymbol{x}) - \nabla \log p_{\text{data}}(\boldsymbol{y})\|_2 \leq M\|\boldsymbol{x} - \boldsymbol{y}\|_2$ for all $\boldsymbol{x}, \boldsymbol{y} \in \mathcal{X}$.*

**Assumption 3** (Coercivity for $-\log p_{\text{data}}$). *There exists constants $c_1 > 0$ and $c_2 \geq 0$ such that*

$$\|\nabla \log p_{\text{data}}(\boldsymbol{x})\|_2^2 \geq -c_1 \log p_{\text{data}} - c_2, \quad \|\boldsymbol{x}\|_2 \leq -c_1 \log p_{\text{data}}(\boldsymbol{x}) + c_2, \quad \forall \boldsymbol{x} \in \mathcal{X} \quad (13)$$

This coercivity assumption means the negative log-density grows sufficiently fast at infinity, which prevents the Langevin dynamics from "escaping to infinity". This assumption guarantees stability and ensure ergodicity leading to convergence to the stationary distribution (Mattingly et al., 2002).

**Theorem 2.** *Suppose that the assumptions in Theorem 1, Assumption 2 and Assumption 3 hold. Further, assume that the DC step reaches the stationary distribution for each $k$. Let $D_{\sigma^{(k)}} : \widetilde{\boldsymbol{z}}^{(k)} \mapsto \boldsymbol{z}_{\text{tw}}^{(k)}$ denote the AC-DC denoiser. Then, we have:*

*(a) With probability at least $1 - 2e^{-\nu_k}$, the following holds for iteration $k$ of ADMM-PnP:*

$$\|(D_{\sigma^{(k)}} - I)(\boldsymbol{x}) - (D_{\sigma^{(k)}} - I)(\boldsymbol{y})\|_2^2 \leq \epsilon_k^2 \|\boldsymbol{x} - \boldsymbol{y}\|_2^2 + \delta_k^2 \quad (14)$$

*for any $\boldsymbol{x}, \boldsymbol{y} \in \mathcal{X}$ and $k \in \mathbb{N}^+$ when $\sigma_{\boldsymbol{s}^{(k)}}^2 + (\sigma^{(k)})^2 < 1/M$ with*

$$\epsilon_k^2 = 3((\sqrt{2}M\sigma_{\boldsymbol{s}^{(k)}}^2/1-\sigma_{\boldsymbol{s}^{(k)}}^2 M)^2 + (\sigma^{(k)})^4 M^2) \quad (15)$$

$$\delta_k^2 = 3(2(\sigma^{(k)})^2(d + 2\sqrt{d\nu_k} + 2\nu_k) + 32d\sigma_{\boldsymbol{s}^{(k)}}^2/(1-M\sigma_{\boldsymbol{s}^{(k)}}^2) \log 2/\nu_k). \quad (16)$$

*In other words, with $\nu_k = \ln 2\pi/6\eta + 2\ln k$, the denoiser $D_{\sigma^{(k)}}$ satisfies part (a) for all $k \in \mathbb{N}^+$ with probability at least $1 - \eta$.*

*(b) Assume that $\sigma^{(k)}$ is scheduled such that $\lim_{k\to\infty}(\sigma^{(k)})^2\nu_k = 0$ for $\nu_k = \ln 2\pi/6\eta + 2\ln k$, $\epsilon < 1$, and $\epsilon/\mu(1+\epsilon-2\epsilon^2) < 1/\rho$ all hold, where $\epsilon = \lim_{k\to\infty} \sup \epsilon_k$ with $\epsilon_k$ defined in (15). Consequently, $\delta = \lim_{k\to\infty} \sup \delta_k$ is finite and ADMM-PnP with the AC-DC denoiser converges to an $r$-ball (see $r$ in Theorem 1) with probability at least $1 - \eta$.*

The proof is relegated to Appendix D. Theorem 2 (a) establishes that the AD-DC denoiser is weakly non-expansive with probability $1 - 2e^{-\nu_k}$ in iteration $k$. The (b) part states that when $\sigma^{(k)}$ is carefully scheduled to approach zero as $k$ grows, then, with high probability, all the iterations satisfy the weakly non-expansiveness together—this leads to the convergence of the ADMM-PnP algorithm.

## 4.3 CONVERGENCE WITHOUT CONVEXITY OF $\ell$

The weakly non-expansiveness based convergence analysis holds under fixed step size (i.e., $\rho$) of the ADMM-PnP algorithm, which is consistent with practical implementations in many cases. However, the assumption that the $\ell$ term is $\mu$-strongly convex is only met by some inverse problems, e.g., signal denoising and deblurring, but not met by others such as signal compression/recovery and data completion. In this subsection, we remove the convexity assumption and analyze the AC-DC denoiser's properties under the adaptive $\rho$-scheme following that in (Chan et al., 2016).

**Theorem 3.** *Suppose that Assumptions 2-3 hold. Let $D := \text{diam}(\mathcal{X}) = \sup_{\boldsymbol{x},\boldsymbol{y}\in\mathcal{X}} \|\boldsymbol{x} - \boldsymbol{y}\|_2 < \infty$, $S := \inf_{\boldsymbol{x}\in\mathcal{X}} \|\nabla \log p_{\text{data}}(\boldsymbol{x})\|_2 < \infty$ and define $L := MD + S$. Let $D_{\sigma^{(k)}} : \widetilde{\boldsymbol{z}}^{(k)} \mapsto \boldsymbol{z}_{\text{tw}}^{(k)}$ denote the AC-DC denoiser. Also, assume that the DC step reaches the stationary distribution for each $k$[1]. Then, the following hold:*

---

[1]Note that Theorems 2 and 3 use this stationary distribution assumption for notation conciseness. For their counterparts removing this assumption, see Appendix E.2.

*(a) (**Boundedness**) With probability at least $1 - 2e^{-\nu_k}$, the denoiser $D_{\sigma^{(k)}}$ is bounded at each iteration $k$ i.e. $\frac{1}{d}\|(D_{\sigma^{(k)}} - I)(\boldsymbol{x})\|_2^2 \le c_k^2$ whenever $\sigma_{\boldsymbol{s}^{(k)}}^2 + (\sigma^{(k)})^2 < 1/M$, where $c_k = (\sigma^{(k)})^2(2 + 4\sqrt{\nu_k} + 4\nu_k) + \frac{16\sigma_{\boldsymbol{s}^{(k)}}^2}{1 - M\sigma_{\boldsymbol{s}^{(k)}}^2}\log 2/\nu_k + 2\sigma_{\boldsymbol{s}^{(k)}}^4 L^2 + 2(\sigma^{(k)})^4 L^2$, and $\nu_k > 0$.*

*Let $\nu_k = \ln\frac{2\pi^2}{6\eta} + 2\ln k$ with $\eta \in (0, 1]$. Consequently, the denoiser $D_{\sigma^{(k)}}$ is bounded for all $k \in \mathbb{N}_+$ with corresponding $c_k$ and probability at least $1 - \eta$.*

*(b) (**Convergence**) Assume there exists $R < \infty$ such that $\|\nabla\ell(\boldsymbol{x})\|_2/\sqrt{d} \le R$. Apply the $\rho$-increasing rule in (Chan et al., 2016) and schedule $(\sigma^{(k)}, \sigma_{\boldsymbol{s}^{(k)}})$ such that $\lim_{k\to\infty}(\sigma^{(k)})^2(2 + 4\sqrt{\nu_k} + 4\nu_k) = 0$, $\lim_{k\to\infty}\frac{\sigma_{\boldsymbol{s}^{(k)}}^2}{1 - M\sigma_{\boldsymbol{s}^{(k)}}^2}\log\frac{2}{\nu_k} = 0$, $\lim_{k\to\infty}\sigma^{(k)} = 0, \lim_{k\to\infty}\sigma_{\boldsymbol{s}^{(k)}} = 0$ $\sigma_{\boldsymbol{s}^{(k)}}^2 + (\sigma^{(k)})^2 < 1/M$, $\forall k \in \mathbb{N}_+$ for $\nu_k = \ln\frac{2\pi^2}{6\eta} + 2\ln k$ with $\eta \in (0, 1]$. Then, the solution sequence converges to a fixed point with probability at least $1 - \eta$.*

The proof is relegated to Appendix E. Theorem 3 (a) shows that, with high probability the denoiser is bounded uniformly across all iterations $k$. Part (b) further shows that, under the proper scheduling of $(\sigma^{(k)}, \sigma_{\boldsymbol{s}^{(k)}})$, the AD-DC ADMM-PnP algorithm converges to a fixed point with high probability.

The condition of $D < \infty$ implies the data space $\mathcal{X}$ has bounded support, which is natural in practice: for images, pixel intensities typically lie within a bounded range such as $[0, 1]$. Additionally, the condition $S < \infty$ ensures that there exists at least one point in $\mathcal{X}$ where the score norm is finite. This prevents pathological cases where the score diverges everywhere (making the distribution degenerate). Together, these conditions guarantee that the score is "well-behaved".

A remark is that all theoretical results in this section focus on fixed-point convergence, which is not the strongest form of convergence guarantees. Establishing stronger convergence results, e.g., stationary-point convergence, for PnP approaches is considered challenging as the objective function is implicit (more specifically, $h(\cdot)$ in (2) is implicit). Nonetheless, in recent years, some efforts have been made towards establishing stationary-point convergence for PnP methods under certain types of denoisers (see, e.g., (Hurault et al., 2022a;b; Wei et al., 2025; Xu et al., 2025)); more discussions are in Sec. 5.

## 5 RELATED WORKS

ADMM-PnP has gained much popularity, due to access to data-driven effective denoisers. It has been used in various applications like image restoration (Chan et al., 2016), data compression (Yuan et al., 2022), hyperspectral imaging (Liu et al., 2022), and medical imaging (Ahmad et al., 2020).

Theoretical understanding of PnP algorithms with general "black-box" denoisers remains limited. Unlike classical proximal operators, the implicit regularizer $h(\cdot)$ in (2) handled by data-driven denoisers is typically unknown. Hence, many results are therefore restricted to fixed-point convergence; see, e.g., (Ryu et al., 2019; Chan et al., 2016). Nonetheless, in certain cases where the denoisers have interesting structures, stronger convergence results can be established. For example, (Xu et al., 2025) used classical results from image denoising connecting linear denoisers with quadratic $h(\cdot)$ to show that when linear denoisers are employed, ADMM-PnP converges to KKT points. Hurault et al. (2022a) showed stationary-point convergence of PnP methods gradient-type denoisers, leveraging the fact that this type of denoisers can be written as a proximal operator of a special function (see the nonconvex counterpart in (Hurault et al., 2022b)); Wei et al. (2025) trained denoisers to satisfy a cocoercive conservativity condition, which also ensures convergence of PnP to stationary points associated with an implicit convex $h(\cdot)$. Nonetheless, these results do not cover diffusion-based denoisers. In this work, we generalize the fixed-point convergence proofs in (Chan et al., 2016; Ryu et al., 2019) to accommodate the diffusion score-based AC–DC denoiser.

Recent advances in score-based generative modeling have motivated their integration into PnP algorithms. One line of work directly replaces the proximal denoiser with a pre-trained scores (Zhu et al., 2023; Li et al., 2024). Alternatively, others embed the score function as an explicit regularizer with task-specific loss (Mardani et al., 2024; Renaud et al., 2024a). Deterministic version PnP have also been considered. For example, Wang et al. (2024); Song et al. (2023) use unrolled ODE and consistency model-distilled one-step representation to express the target signal, respectively. These methods are similar to (Bora et al., 2017), but with diffusion-driven parameterization.

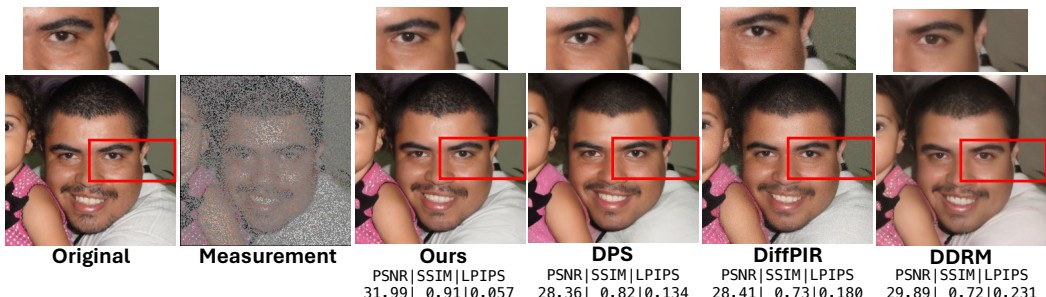

Figure 2: Inpainting under random missings.

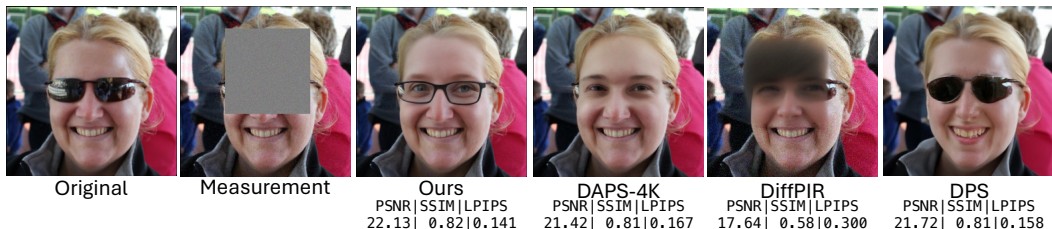

Figure 3: Inpainting under box missing.

Prior works have emphasized the importance of matching the residual noise to the operating range of the PnP denoiser. D-AMP (Metzler et al., 2016; Eksioglu & Tanc, 2018) achieve this via the Onsager correction, which approximately Gaussianizes the residual under compressive sensing problem structures. Wei et al. (2021) learns a reinforcement learning-based policy to automatically tune all internal parameters, including denoising strength. Unlike AC–DC that provides a generic correction mechanism for a variety inverse problems, these methods either problem specific or require training additional models. Score-based inverse problem solvers have also attempted to "bring" iterates to noisy data manifolds used during training. The work Chung et al. (2022) uses a manifold constraint based on gradient of data-fidelity, while He et al. (2024) uses an off-the-shelf pretrained neural network to impose a manifold constraint. On the other hand, Zirvi et al. (2025) uses the projection of measurement guidance to low-rank subspace, using SVD on the intermediate diffusion state, for similar purposes.

The idea of adding noise before evaluating score functions during optimization procedures (similar to our AC step) has been widely considered (Li et al., 2024; Graikos et al., 2022; Renaud et al., 2024b; Mardani et al., 2024). A variant of this called estimation-correction idea proposed in (Karras et al., 2022) is used in (Zhu et al., 2023) for this purpose.

## 6 EXPERIMENTS

**Dataset and Evaluation Metrics.** For all these tasks, we use two datasets: FFHQ $256 \times 256$ (Karras et al., 2021) and ImageNet $256 \times 256$ (Deng et al., 2009). During testing, we randomly sample 100 images from the validation set of each dataset. All the methods use the pre-trained score model in Chung et al. (2023). We use *Peak Signal-to-Noise Ratio* (PSNR) as a pixel-wise similarity metric, and *Structural Similarity Index* (SSIM) and *Learned Perceptual Image Patch Similarity* (LPIPS) (Zhang et al., 2018) as perceptual similarity metrics. We report these metrics averaged over the 100 test images for each method and inverse problem.

**Task Description.** We consider $\boldsymbol{\xi} \sim \mathcal{N}(\mathbf{0}, \sigma_n^2 \boldsymbol{I})$ with $\sigma_n = 0.05$ for all the tasks. (a) For *super-resolution*, we use cubic interpolation method with kernel size 4 for downsampling the resolution by 4 times. (b) For *recovery under Gaussian blurring* (Gaussian deblurring), a kernel of size 61 and standard deviation 3 is used. (c) As for *recovery under motion blurring* (motion deblurring), a kernel of size 61 and standard deviation of 0.5 is used. (d) In *inpainting under box mask* (box inpainting), an approximately centered mask of size $128 \times 128$ is sampled in image while maintaining the 32

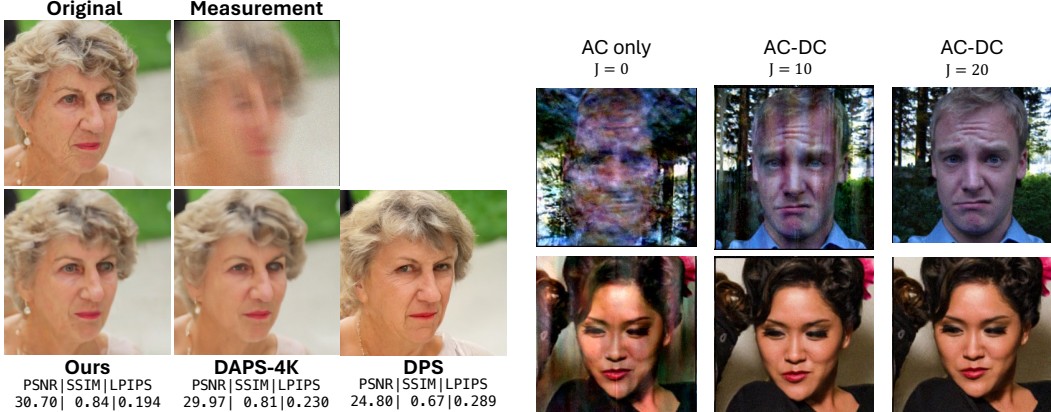

Figure 4: Recovery under motion blurring.    Figure 5: Influence of DC steps in the denoiser.

pixel margin in both spatial dimensions of the input image. (e) For *inpainting under random missings* (random inpainting), $70\%$ of the pixels are uniformly sampled to be masked, and a scaling of 2 was used in high dynamic range (HDR) before clipping the values. (f) For *phase retrieval*, similar as in prior works (Wu et al., 2024; Mardani et al., 2024), we use oversampling by factor of 2. g) For deblurring under nonlinear blurring, we use the operator in (Tran et al., 2021) with default settings.

**Baselines.** We use a set of baselines, namely, DPS (Chung et al., 2023), DAPS (Zhang et al., 2024), DDRM (Kawar et al., 2022), DiffPIR (Zhu et al., 2023), RED-diff (Mardani et al., 2024), DPIR (Zhang et al., 2022), DCDP (Li et al., 2025), PMC (Sun et al., 2024).

**Hyperparameter Settings.** We adopt a linear schedule for $\sigma^{(k)}$ with range $[0.1, 10]$ over $W$ decay window i.e. $\sigma^{(k)} = \max(0.1, 10 - (10 - 0.1) \cdot k/W)$. The maximal number of iterations for our proposed method is set to $K = W + 10$. At iteration $k$, we use $J = 10$ DC steps, and the schedules $\eta^{(k)} = 5 \times 10^{-4}\sigma^{(k)}$ and $\sigma_{\boldsymbol{s}^{(k)}} = {}^{0.1}/\sqrt{\sigma^{(k)}}$. We use gradient descent with Adam optimizer (Kingma & Ba, 2015) for solving each regularized maximum likelihood subproblem (7a). This subproblem is optimized for maximum of 1000 iterations with convergence detected when the loss value increases more than $\Delta_{\text{tol}} = 1 \times 10^{-1}$ consecutively for 3 iterations window. We conduct our experiment with two variants based on the third stage: using Tweedie's lemma (denoted as "Ours-tweedie") and a 10-step ODE based denoiser (Zhang et al., 2024; Karras et al., 2022) (denoted as "Ours-ode"). We use the preconditioning in Karras et al. (2022) while using the pretrained diffusion models.

**Qualitative Performance.** Figs 2, 3 and 4 show reconstructions under inpainting under random missings, inpainting under box missing, and motion deblurring. It can be seen that our method is able to recover the image that is comparatively natural looking with less noise and artifacts, while being consistent with the measurements. On the other hand, images recovered with DiffPIR appears to suffer from noise and artifacts, whereas DPS leads to measurement-inconsistent reconstructions.

Our method outperforms others while other methods appear to either be blurred or contain noisy artifacts in the recovered image. Recovery by DPS is less consistent with the original image; the pattern on the child's clothing is completely lost.

**Quantitative Performance.** Table 1 summarizes PSNR, SSIM and LPIPS averaged over 100 images on FFHQ and Imagenet datasets. In almost all of the inverse problems, both of our variants (Ours-tweedie and Ours-ode) achieve the best or second-best performance in terms of all metrics. Our method significantly outperforms other PnP baseline methods considered, namely, DDRM, DiffPIR and RED-diff. This demonstrates the effectiveness of our AC-DC denoiser.

**Effectiveness of DC.** To perform ablation study on the DC stage, we consider the challenging phase retrieval problem. Fig. 5 shows the output of ADMM-PnP with our AC-DC denoiser with different numbers of DC iterations $J$. With $J = 0$ (disabling DC step), artifacts remain severe. Increasing $J$ progressively results in cleaner images.

**More Details and Additional Experiments.** More details and experiments are in appendices.

Table 1: Reconstruction metrics (100 images) on FFHQ / ImageNet. **Bold**: best, blue: 2nd best.

| Task | Method | FFHQ PSNR↑ | SSIM↑ | LPIPS↓ | ImageNet PSNR↑ | SSIM↑ | LPIPS↓ |
|---|---|---|---|---|---|---|---|
| Superresolution (4×) | Ours–tweedie | **30.439** | **0.857** | 0.178 | **27.318** | **0.717** | 0.280 |
| | Ours–ode | 29.991 | 0.845 | **0.156** | 26.919 | 0.700 | 0.276 |
| | DAPS | 29.529 | 0.814 | 0.167 | 26.653 | 0.680 | **0.266** |
| | DPS | 24.828 | 0.705 | 0.257 | 22.785 | 0.549 | 0.411 |
| | DDRM | 27.145 | 0.782 | 0.261 | 26.105 | 0.683 | 0.306 |
| | DiffPIR | 26.771 | 0.749 | 0.208 | 23.884 | 0.543 | 0.336 |
| | RED-diff | 16.833 | 0.422 | 0.547 | 18.662 | 0.309 | 0.519 |
| | DPIR | 28.849 | 0.826 | 0.254 | 26.524 | 0.699 | 0.334 |
| | DCDP | 27.761 | 0.639 | 0.332 | 24.517 | 0.525 | 0.361 |
| | PMC | 23.774 | 0.421 | 0.407 | 22.534 | 0.334 | 0.456 |
| Inpainting (Random) | Ours–tweedie | **32.844** | **0.906** | 0.122 | **29.564** | **0.817** | 0.184 |
| | Ours–ode | 32.127 | 0.894 | **0.095** | 28.733 | 0.795 | **0.148** |
| | DAPS | 31.652 | 0.847 | 0.124 | 28.137 | 0.751 | 0.162 |
| | DPS | 29.084 | 0.828 | 0.181 | 26.049 | 0.678 | 0.318 |
| | DDRM | 28.969 | 0.847 | 0.178 | 27.883 | 0.778 | 0.203 |
| | DiffPIR | 28.558 | 0.709 | 0.230 | 26.923 | 0.639 | 0.222 |
| | RED-diff | 20.361 | 0.630 | 0.275 | 20.948 | 0.464 | 0.315 |
| | PMC | 23.289 | 0.755 | 0.263 | 25.965 | 0.636 | 0.342 |
| Motion Deblur | Ours–tweedie | **30.003** | **0.854** | 0.179 | **27.149** | **0.717** | 0.280 |
| | Ours–ode | 29.648 | 0.841 | **0.154** | 26.615 | 0.694 | 0.275 |
| | DAPS | 29.051 | 0.815 | 0.175 | 26.571 | 0.689 | 0.276 |
| | DPS | 23.257 | 0.663 | 0.265 | 19.613 | 0.451 | 0.451 |
| | PMC | 19.480 | 0.590 | 0.426 | 21.608 | 0.480 | 0.510 |

| Task | Method | FFHQ PSNR↑ | SSIM↑ | LPIPS↓ | ImageNet PSNR↑ | SSIM↑ | LPIPS↓ |
|---|---|---|---|---|---|---|---|
| Gaussian Blur | Ours-tweedie | **30.402** | **0.853** | 0.175 | **27.199** | **0.705** | 0.281 |
| | Ours-ode | 30.019 | 0.841 | 0.158 | 26.899 | 0.690 | 0.282 |
| | DAPS | 29.790 | 0.813 | **0.157** | 26.886 | 0.678 | **0.260** |
| | DPS | 26.106 | 0.730 | 0.207 | 23.995 | 0.575 | 0.328 |
| | DiffPIR | 25.148 | 0.699 | 0.230 | 22.756 | 0.508 | 0.374 |
| | DPIR | 28.875 | 0.833 | 0.228 | 26.702 | 0.700 | 0.314 |
| | DCDP | 16.821 | 0.171 | 0.642 | 15.102 | 0.136 | 0.620 |
| | PMC | 20.172 | 0.638 | 0.344 | 24.103 | 0.545 | 0.415 |
| Inpainting (Box) | Ours-tweedie | 24.025 | **0.859** | **0.131** | 21.626 | **0.789** | 0.222 |
| | Ours-ode | 23.342 | 0.837 | 0.136 | 20.618 | 0.743 | 0.227 |
| | DAPS | 23.643 | 0.815 | 0.146 | 21.303 | 0.774 | 0.199 |
| | DPS | 23.488 | 0.817 | 0.164 | 19.933 | 0.677 | 0.309 |
| | DiffPIR | 20.934 | 0.561 | 0.294 | 19.565 | 0.562 | 0.342 |
| | RED-diff | 18.713 | 0.523 | 0.364 | 18.075 | 0.499 | 0.371 |
| | DCDP | 25.230 | 0.754 | 0.163 | 20.991 | 0.727 | **0.195** |
| | PMC | 14.828 | 0.697 | 0.318 | 15.550 | 0.666 | 0.326 |
| Phase Retrieval | Ours-tweedie | **27.944** | **0.793** | **0.209** | **17.770** | **0.440** | **0.471** |
| | Ours-ode | 27.095 | 0.757 | 0.237 | 16.013 | 0.339 | 0.539 |
| | DAPS | 26.707 | 0.749 | 0.230 | 16.444 | 0.395 | 0.512 |
| | DPS | 11.627 | 0.366 | 0.658 | 9.434 | 0.216 | 0.768 |
| | RED-diff | 15.411 | 0.490 | 0.480 | 12.852 | 0.204 | 0.695 |
| | DCDP | 20.026 | 0.540 | 0.424 | 12.257 | 0.212 | 0.665 |
| | PMC | 10.421 | 0.287 | 0.783 | 8.636 | 0.129 | 0.890 |

# 7 CONCLUSION

We introduced the AC-DC denoiser, a score-based denoiser designed for integration within the ADMM-PnP framework. The denoiser adopts a three-stage structure aimed at mitigating the mismatch between ADMM iterates and the noisy manifolds on which score functions are trained. We established convergence guarantees for ADMM-PnP with the AC-DC denoiser under both fixed and adaptive step size schedules. Empirical results across a range of inverse problems demonstrate that the proposed method consistently improves solution quality over existing baselines.

**Limitations.** While our analysis provides initial insights, several aspects merit deeper understanding. The second convergence result relaxes convexity by allowing adaptive step sizes, though such schedules are arguably less appealing in practice. Our experiments, however, suggest that constant step sizes also perform well for nonconvex objectives; it is therefore desirable to establish convergence guarantees for constant step sizes in such settings. In addition, our result ensures the *stability* of the ADMM method, but does not directly explain the reason *why* the AC–DC denoiser attains high-quality recovery; recoverability and estimation error analyses are also desirable. On the implementation side, the noise schedules used in the AC and DC stages are currently guided by empirical heuristics. Designing problem-adaptive scheduling strategies may further improve both convergence speed and robustness. Additionally, each iteration of AC–DC denoiser needs multiple score evaluations. Reducing the required NFEs could significantly improve its efficiency.

**Ethics Statement**: This work focuses exclusively on the theory and methodology of solving inverse problems. It does not involve human subjects, personal data, or any sensitive procedures.

**Reproducibility Statement**: The source code is provided as a part of the supplementary material. All assumptions, derivations and necessary details regarding the theory and experiments are included in the appendices.

**Acknowledgment**: This work was supported in part by the National Science Foundation (NSF) under Project NSF CCF-2210004. It was also supported in part by the SciRIS seed funding from the College of Science and College of Engineering at Oregon State University.

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

# A PRELIMINARIES

## A.1 NOTATION

Table 2: Summary of notation

| Symbol | Description |
|---|---|
| $\boldsymbol{x} \in \mathbb{R}^d$ | The unknown signal or image to be recovered |
| $\boldsymbol{y} \in \mathbb{R}^n$ | Measurements, with $n \leq d$ |
| $y_i$ | $i$th element of measurement $\boldsymbol{y}$ |
| $\mathcal{A} : \mathbb{R}^d \to \mathbb{R}^n$ | Measurement operator |
| $\mathcal{X}$ | Support of $\boldsymbol{x}$ |
| $\mathcal{X}_t$ | Support of $\boldsymbol{x}_t$ |
| $\boldsymbol{\xi}$ | Additive measurement noise |
| $\boldsymbol{x}_t$ | noisy data by using forward diffusion process with noise $\sigma(t)$ |
| $\boldsymbol{x}_{\sigma^{(k)}}$ | noisy data by using forward diffusion process with noise $\sigma^{(k)}$ |
| $\ell(\boldsymbol{y} \| \mathcal{A}(\boldsymbol{x}))$ | data-fidelity loss (e.g. $\|\boldsymbol{y} - \mathcal{A}(\boldsymbol{x})\|_2^2$) |
| $h(\boldsymbol{x})$ | structural regularization prior (enforced via denoiser) |
| $\rho > 0$ | ADMM penalty parameter |
| $\mathrm{Prox}(\cdot)$ | Proximal operator |
| $\boldsymbol{u}^{(k)}$ | Scaled dual variable in iteration $k$ of ADMM |
| $\boldsymbol{z}^{(k)}$ | Auxiliary variable in iteration $k$ of ADMM |
| $\widetilde{\boldsymbol{z}}^{(k)} = \boldsymbol{x}^{(k+1)} + \boldsymbol{u}^{(k)}$ | Pre-denoising input to the PnP denoiser |
| $\sigma^{(k)}$ | Noise level schedule for the AC-DC denoiser |
| $\sigma_{\boldsymbol{s}^{(k)}}$ | Variance parameter in the AC-DC prior for directional correction. |
| $\boldsymbol{n} \sim \mathcal{N}(\boldsymbol{0}, \boldsymbol{I})$ | Multivariate standard gaussian random variable |
| $\boldsymbol{s}_{\boldsymbol{\theta}}(\boldsymbol{x}, \sigma) \approx \nabla_{\boldsymbol{x}} \log p(\boldsymbol{x} + \sigma \boldsymbol{n})$ | Pretrained score function |
| $K$ | Maximum iteration of ADMM |
| $J$ | Total iteration of directional correction at for each denoising |
| $M$ | Smoothness constant of $\nabla \log p_{\mathrm{data}}$ (Assumption 2) |
| $M_t$ | Smoothness constant of $\nabla \log p_t$ |
| $T > 0$ | Maximum time steps used for diffusion |
| $\mathcal{M}_{\sigma(t)}$ | Manifold of $\boldsymbol{x}_t$ |
| $\mathcal{M}_{\sigma^{(k)}}$ | Manifold of $\boldsymbol{x}_t$ where $t \in [0, T]$ such that $\sigma(t) = \sigma^{(k)}$ |
| $D_{\sigma^{(k)}}(\boldsymbol{z})$ | AC-DC denoiser at $k$th iteration |
| $R_{\sigma^{(k)}}(\boldsymbol{z}) = D_{\sigma^{(k)}}(\boldsymbol{z}) - \boldsymbol{z}$ | Residual of AC-DC denoiser at $k$th iteration |
| $I(\boldsymbol{x}) = \boldsymbol{x}$ | Identity mapping function |
| $\boldsymbol{I}$ | Identity matrix |
| $\boldsymbol{0}$ | vector of values 0 |
| $T_1 \circ T_2(\boldsymbol{z}) = T_1(T_2(\boldsymbol{z}))$ | Concatenation of two functions $T_1$ and $T_2$ |
| $\|\boldsymbol{x}\|_2$ | 2-norm of a vector $\boldsymbol{x}$ |
| $\mathrm{Cov}(\cdot)$ | Covariance matrix |

## A.2 DEFINITIONS

**$\mu$-strongly convex function (Boyd & Vandenberghe, 2004).** A differentiable function $f : \mathbb{R}^m \to \mathbb{R}$ is *$\mu$-strongly convex* for a certain $\mu > 0$ if

$$f(\boldsymbol{y}) \geq f(\boldsymbol{x}) + \nabla f(\boldsymbol{x})^T (\boldsymbol{y} - \boldsymbol{x}) + \frac{\mu}{2} \|\boldsymbol{y} - \boldsymbol{x}\|_2^2 \tag{17}$$

for all $\boldsymbol{x}, \boldsymbol{y} \in \mathbb{R}^m$.

The notion of of *nonexpansive* and *averaged nonexpansive* have been widely used in the convergence analysis of various nonlinear problems (Combettes & Yamada, 2015; Yao et al., 2008; Eckstein &

Bertsekas, 1992). We use the generalized form of both *nonexpansive* and *averaged nonexpansive* operator for establishing the ball convergence in our method.

**Nonexpansive function  (Browder, 1965).**  A function $T : \mathbb{R}^d \to \mathbb{R}^d$ is *nonexpansive* if $T$ is *nonexpansive* function if there exists $\epsilon \in [0, 1]$ such that

$$\|T(\boldsymbol{x}) - T(\boldsymbol{y})\|_2^2 \leq \epsilon^2 \|\boldsymbol{x} - \boldsymbol{y}\|_2^2 \tag{18}$$

for all $\boldsymbol{x}, \boldsymbol{y} \in \mathbb{R}^d$.

**$\theta$-averaged function  (Combettes & Yamada, 2015).**  A mapping $T : \mathbb{R}^d \to \mathbb{R}^d$ is defined to be $\theta$-*averaged* for a constant $\theta \in (0, 1)$ if there exists a nonexpansive operator $R : \mathbb{R}^d \to \mathbb{R}^d$ such that $T = (1 - \theta)I + \theta R$.

The notion of relaxed bound $\|T_k(\boldsymbol{x}) - T_k(\boldsymbol{y})\| \leq \epsilon^{(k)} \|\boldsymbol{x} - \boldsymbol{y}\| + \delta^{(k)}$, $\forall \boldsymbol{x}, \boldsymbol{y} \in \mathcal{X}$ was used to study and show the convergence in Yao et al. (2008) when $\sum_{k=1}^{\infty} |\delta^{(k)}| < \infty$. We define a similar weaker form of nonexpansive function and $\theta$-averaged functions below.

**$\delta$-weakly nonexpansive function.**  A mapping $T : \mathbb{R}^d \to \mathbb{R}^d$ is said to be $\delta$-*weakly nonexpansive* for $\delta \geq 0$ if there exists $\epsilon \in [0, 1]$ such that

$$\|T(\boldsymbol{x}) - T(\boldsymbol{y})\|_2^2 \leq \epsilon^2 \|\boldsymbol{x} - \boldsymbol{y}\|_2^2 + \delta^2 \tag{19}$$

for all $\boldsymbol{x}, \boldsymbol{y} \in \mathbb{R}^d$.

**$\delta$-weakly $\theta$-averaged function.**  A mapping $T : \mathbb{R}^d \to \mathbb{R}^d$ is defined to be $\delta$-*weakly $\theta$-averaged* function for a certain $\delta \geq 0$ and $\theta \in (0, 1)$, if there exists a $\delta$-weakly nonexpansive function $R : \mathbb{R}^d \to \mathbb{R}^d$ such that $T = \theta R + (1 - \theta)I$.

**Sub-Gaussian random vector.**  A random vector $\boldsymbol{x} \in \mathbb{R}^d$ (with mean $\mathbb{E}[\boldsymbol{x}]$) is called *sub-Gaussian with parameter* $\sigma^2$ if its Euclidean norm satisfies a sub-Gaussian tail bound:

$$\Pr(\|\boldsymbol{x} - \mathbb{E}[\boldsymbol{x}]\|_2 > \varepsilon) \leq 2 \exp\left(-\frac{\varepsilon^2}{2\sigma^2}\right), \quad \forall \varepsilon > 0. \tag{20}$$

**2-Wasserstein Distance.**  Let $\mu$ and $\nu$ be probability measures on $\mathbb{R}^d$ with finite second moments. The 2-Wasserstein distance between $\mu$ and $\nu$ is defined as

$$W_2(\mu, \nu) = \left( \inf_{\gamma \in \Gamma(\mu, \nu)} \int_{\mathbb{R}^d \times \mathbb{R}^d} \|x - y\|_2^2 \, d\gamma(x, y) \right)^{1/2}, \tag{21}$$

where $\Gamma(\mu, \nu)$ denotes the set of all couplings of $\mu$ and $\nu$, i.e.,

$$\Gamma(\mu, \nu) = \left\{ \gamma \in \mathcal{P}(\mathbb{R}^d \times \mathbb{R}^d) : \gamma(A \times \mathbb{R}^d) = \mu(A), \ \gamma(\mathbb{R}^d \times B) = \nu(B), \ \forall A, B \subseteq \mathbb{R}^d \text{ measurable} \right\}. \tag{22}$$

### A.3  Supporting Lemmas

Tweedie's lemma establishes an important connection between the score of the marginal distribution and expectation of posterior when the likelihood function is gaussian. This allows the score function of the diffusion model to be used as a *minimum-mean-square-error* (MMSE) denoiser.

**Lemma 1** (Tweedie's lemma (Robbins, 1992)). *Let $p_0(\boldsymbol{x}_0)$ be the prior distribution and then $\boldsymbol{x}_t \sim \mathcal{N}(\boldsymbol{x}_0, \boldsymbol{\Sigma})$ be observed with $\boldsymbol{\Sigma}$ known. Suppose $p_t(\boldsymbol{x}_t)$ be the marginal distribution of $\boldsymbol{x}_t$. Then, Tweedie's lemma computes the posterior expectation of $\boldsymbol{x}_0$ given $\boldsymbol{x}_t$ as*

$$\mathbb{E}[\boldsymbol{x}_0 | \boldsymbol{x}_t] = \boldsymbol{x}_t + \boldsymbol{\Sigma} \nabla \log p_t(\boldsymbol{x}_t) \tag{23}$$

The lemmas related to $\theta$-averaged from Combettes & Yamada (2015) are used to show fixed point convergence in Ryu et al. (2019). In the following, we extend all these lemmas to a more general $\delta$-weakly $\theta$-averaged cases that will be used later to show our ball convergence.

**Lemma 2.** $T : \mathbb{R}^d \to \mathbb{R}^d$ *be a function. Then, the following statements are equivalent:*

(a) *$T$ is $\delta$-weakly $\theta$-averaged for $\delta \geq 0$ and $\theta \in (0, 1)$.*

(b) *$\|T(\boldsymbol{x}) - T(\boldsymbol{y})\|_2^2 + (1 - 2\theta)\|\boldsymbol{x} - \boldsymbol{y}\|_2^2 - 2(1 - \theta)\langle T(\boldsymbol{x}) - T(\boldsymbol{y}), \boldsymbol{x} - \boldsymbol{y}\rangle \leq \delta^2 \theta^2$, for all $\boldsymbol{x}, \boldsymbol{y} \in \mathbb{R}^d$.*

(c) *$(1 - 1/\theta)I + (1/\theta)T$ is $\delta$-weakly nonexpansive.*

(d) *$\|T(\boldsymbol{x}) - T(\boldsymbol{y})\|_2^2 \leq \|\boldsymbol{x} - \boldsymbol{y}\|_2^2 - \frac{1-\theta}{\theta}\|(I - T)(\boldsymbol{x}) - (I - T)(\boldsymbol{y})\|_2^2 + \delta^2 \theta$, for all $\boldsymbol{x}, \boldsymbol{y} \in \mathbb{R}^d$*

*Proof.* Equivalence between (a) and (b): Provided $T$ is $\delta$-*weakly $\theta$-averaged*, let's find the LHS - RHS in (b)

$$
\begin{aligned}
&(1 - 2\theta)\|\boldsymbol{x} - \boldsymbol{y}\|_2^2 + \|T(\boldsymbol{x}) - T(\boldsymbol{y})\|_2^2 - 2(1 - \theta)\langle \boldsymbol{x} - \boldsymbol{y}, T(\boldsymbol{x}) - T(\boldsymbol{y})\rangle \\
\leq& (1 - 2\theta)\|\boldsymbol{x} - \boldsymbol{y}\|_2^2 + \|\theta R(\boldsymbol{x}) - \theta R(\boldsymbol{y}) + (1 - \theta)(\boldsymbol{x} - \boldsymbol{y})\|_2^2 \\
&\quad - 2(1 - \theta)\langle \boldsymbol{x} - \boldsymbol{y}, \theta(R(\boldsymbol{x}) - R(\boldsymbol{y})) + (1 - \theta)(\boldsymbol{x} - \boldsymbol{y})\rangle \\
=& (1 - 2\theta)\|\boldsymbol{x} - \boldsymbol{y}\|_2^2 + \theta^2\|R(\boldsymbol{x}) - R(\boldsymbol{y})\|_2^2 + (1 - \theta)^2\|\boldsymbol{x} - \boldsymbol{y}\|_2^2 - 2(1 - \theta)^2\|\boldsymbol{x} - \boldsymbol{y}\|_2^2 \\
&\quad + 2\theta(1 - \theta)\langle \boldsymbol{x} - \boldsymbol{y}, R(\boldsymbol{x}) - R(\boldsymbol{y})\rangle - 2(1 - \theta)\theta\langle \boldsymbol{x} - \boldsymbol{y}, R(\boldsymbol{x}) - R(\boldsymbol{y})\rangle \\
\leq& (1 - 2\theta)\|\boldsymbol{x} - \boldsymbol{y}\|_2^2 + \theta^2\|\boldsymbol{x} - \boldsymbol{y}\|_2^2 + \theta^2\delta^2 + (1 - \theta)^2\|\boldsymbol{x} - \boldsymbol{y}\|_2^2 - 2(1 - \theta)^2\|\boldsymbol{x} - \boldsymbol{y}\|_2^2 \\
&\quad + 2\theta(1 - \theta)\langle \boldsymbol{x} - \boldsymbol{y}, R(\boldsymbol{x}) - R(\boldsymbol{y})\rangle - 2(1 - \theta)\theta\langle \boldsymbol{x} - \boldsymbol{y}, R(\boldsymbol{x}) - R(\boldsymbol{y})\rangle \\
=& (1 - 2\theta + \theta^2 + (1 - \theta)^2 - 2(1 - \theta)^2)\|\boldsymbol{x} - \boldsymbol{y}\|_2^2 + \theta^2\delta^2 \\
=& \theta^2\delta^2
\end{aligned}
$$

For another direction, let us suppose $T$ satisfies (a). Let $R = \frac{1}{\theta}(T - (1 - \theta)I)$ so that we have $T = \theta R + (1 - \theta)I$. Now, we need to show that $\|R(\boldsymbol{x}) - R(\boldsymbol{y})\|_2^2 \leq \|\boldsymbol{x} - \boldsymbol{y}\|_2^2 + \delta^2$ for all $\boldsymbol{x}, \boldsymbol{y} \in \mathbb{R}^d$ i.e. $\delta$-*weakly nonexpansive*.

$$
\begin{aligned}
&\|R(\boldsymbol{x}) - R(\boldsymbol{y})\|_2^2 \\
=& \frac{1}{\theta^2}\|T(\boldsymbol{x}) - T(\boldsymbol{y}) - (1 - \theta)(\boldsymbol{x} - \boldsymbol{y})\|_2^2 \\
=& \frac{1}{\theta^2}\left(\|T(\boldsymbol{x}) - T(\boldsymbol{y})\|_2^2 + (1 - \theta)^2\|\boldsymbol{x} - \boldsymbol{y}\|_2^2 - 2(1 - \theta)\langle T(\boldsymbol{x}) - T(\boldsymbol{y}), \boldsymbol{x} - \boldsymbol{y}\rangle\right) \\
=& \frac{1}{\theta^2}\left(\|T(\boldsymbol{x}) - T(\boldsymbol{y})\|_2^2 + (1 - 2\theta)\|\boldsymbol{x} - \boldsymbol{y}\|_2^2 - 2(1 - \theta)\langle T(\boldsymbol{x}) - T(\boldsymbol{y}), \boldsymbol{x} - \boldsymbol{y}\rangle + \theta^2\|\boldsymbol{x} - \boldsymbol{y}\|_2^2\right) \\
\leq& \frac{1}{\theta^2}\left(\theta^2\delta^2 + \theta^2\|\boldsymbol{x} - \boldsymbol{y}\|_2^2\right) \\
=& \|\boldsymbol{x} - \boldsymbol{y}\|_2^2 + \delta^2
\end{aligned}
$$

where, the inequality is due to $T$ satisfying (b).

Equivalence between (a) and (c): Note that $T$ is $\delta$-weakly $\theta$-averaged $\iff T = \theta R + (1 - \theta)I$ with $R$ being $\delta$-weakly nonexpansive function. Now, we have

$$
\begin{aligned}
(1 - 1/\theta)I + (1/\theta)T &= (1 - 1/\theta)I + 1/\theta \cdot (\theta R + (1 - \theta)I) \\
&= (1 - 1/\theta)I + R - (1 - 1/\theta)I \\
&= R
\end{aligned}
$$

Hence, $T$ being $\delta$-weakly $\theta$-average is equivalent to $(1 - 1/\theta)I + (1/\theta)T$ being $\delta$-weakly nonexpansive.

Equivalence between (a) and (d): From equivalence between (a) and (b), we have

$$\|T(\boldsymbol{x}) - T(\boldsymbol{y})\|_2^2 + (1 - 2\theta)\|\boldsymbol{x} - \boldsymbol{y}\|_2^2 - 2(1 - \theta)\langle T(\boldsymbol{x}) - T(\boldsymbol{y}), \boldsymbol{x} - \boldsymbol{y}\rangle \le \delta^2\theta^2$$

$$\iff \|T(\boldsymbol{x}) - T(\boldsymbol{y})\|_2^2 + (1 - 2\theta)\|\boldsymbol{x} - \boldsymbol{y}\|_2^2 -$$
$$(1 - \theta)\left(\|T(\boldsymbol{x}) - T(\boldsymbol{y})\|_2^2 + \|\boldsymbol{x} - \boldsymbol{y}\|_2^2 - \|(T - I)(\boldsymbol{x}) - (T - I)(\boldsymbol{y})\|_2^2\right) \le \delta^2\theta^2$$

$$\iff \theta\|T(\boldsymbol{x}) - T(\boldsymbol{y})\|_2^2 - \theta\|\boldsymbol{x} - \boldsymbol{y}\|_2^2 \le \delta^2\theta^2 - (1 - \theta)\|(T - I)(\boldsymbol{x}) - (T - I)(\boldsymbol{y})\|_2^2$$

$$\iff \|T(\boldsymbol{x}) - T(\boldsymbol{y})\|_2^2 \le \|\boldsymbol{x} - \boldsymbol{y}\|_2^2 - \frac{1 - \theta}{\theta}\|(I - T)(\boldsymbol{x}) - (I - T)(\boldsymbol{y})\|_2^2 + \delta^2\theta$$

$$\square$$

**Lemma 3** (Concatenation of $\delta$-weakly $\theta$-averaged functions). *Assume $T_1 : \mathbb{R}^d \to \mathbb{R}^d$ and $T_2 : \mathbb{R}^d \to \mathbb{R}^d$ are $\delta_1$-weakly $\theta_1$-averaged and $\delta_2$-weakly $\theta_2$-averaged respectively. Then, $T_1 \circ T_2$ is $\delta$-weakly $\theta$-averaged, with $\theta = \frac{\theta_1 + \theta_2 - 2\theta_1\theta_2}{1 - \theta_1\theta_2}$, and $\delta^2 = \frac{1}{\theta}(\delta_1^2\theta_1 + \delta_2^2\theta_2)$.*

*Proof.* Here, we follow the proof structure of Combettes & Yamada (2015). Since $\theta_1(1 - \theta_2) \le (1 - \theta_2)$, we have $\theta_1 + \theta_2 \le 1 + \theta_1\theta_2$, and therefore, $\theta = \frac{\theta_1 + \theta_2 - 2\theta_1\theta_2}{1 - \theta_1\theta_2} \in (0, 1)$, and let $\delta^2 = \frac{\delta_1^2\theta_1 + \delta_2^2\theta_2}{\theta}$

Now, from Lemma 2, for $i \in \{1, 2\}$, we have,

$$\|T_i(\boldsymbol{x}) - T_i(\boldsymbol{y})\|_2^2 \le \|\boldsymbol{x} - \boldsymbol{y}\|_2^2 - \frac{1 - \theta_i}{\theta_i}\|(I - T_i)(\boldsymbol{x}) - (I - T_i)(\boldsymbol{y})\|_2^2 + \delta_i^2\theta_i \qquad (24)$$

Then, let us evaluate the composition function using this property.

$$\|T_1 \circ T_2(\boldsymbol{x}) - T_1 \circ T_2(\boldsymbol{y})\|_2^2$$
$$\le \|T_2(\boldsymbol{x}) - T_2(\boldsymbol{y})\|_2^2 - \frac{1 - \theta_1}{\theta_1}\|(I - T_1)(T_2(\boldsymbol{x})) - (I - T_1)(T_2(\boldsymbol{y})\|_2^2$$
$$\le \|\boldsymbol{x} - \boldsymbol{y}\|_2^2 - \frac{1 - \theta_2}{\theta_2}\|(I - T_2)(\boldsymbol{x}) - (I - T_2)(\boldsymbol{y})\|_2^2 + \delta_2^2\theta_2$$
$$- \frac{1 - \theta_1}{\theta_1}\|(I - T_1)(T_2(\boldsymbol{x})) - (I - T_1)(T_2(\boldsymbol{y})\|_2^2 + \delta_1^2\theta_1$$

From Bauschke et al. (2017)[Corollary 2.15], we have, for $\alpha \in \mathbb{R}$,

$$\|\alpha\boldsymbol{u} + (1 - \alpha)\boldsymbol{v}\|_2^2 + \alpha(1 - \alpha)\|\boldsymbol{u} - \boldsymbol{v}\|_2^2 = \alpha\|\boldsymbol{u}\|_2^2 + (1 - \alpha)\|\boldsymbol{v}\|_2^2$$
$$\implies \alpha(1 - \alpha)\|\boldsymbol{u} + \boldsymbol{v}\|_2^2 \le \alpha\|\boldsymbol{u}\|_2^2 + (1 - \alpha)\|\boldsymbol{v}\|_2^2$$

Now, let $\boldsymbol{u} = (I - T_2)(\boldsymbol{x}) - (I - T_2)(\boldsymbol{y})$, $\boldsymbol{v} = (I - T_1)(T_2(\boldsymbol{x})) - (I - T_1)(T_2(\boldsymbol{y}))$, $a = \frac{1 - \theta_2}{\theta_2}$, and $b = \frac{1 - \theta_1}{\theta_1}$.

$$\|T_1 \circ T_2(\boldsymbol{x}) - T_1 \circ T_2(\boldsymbol{y})\|_2^2$$
$$\le \|\boldsymbol{x} - \boldsymbol{y}\|_2^2 - a\|\boldsymbol{u}\|_2^2 - b\|\boldsymbol{v}\|_2^2 + \delta_1^2\theta_1 + \delta_2^2\theta_2$$
$$= \|\boldsymbol{x} - \boldsymbol{y}\|_2^2 - (a + b)\left(\frac{a}{a + b}\|\boldsymbol{u}\|_2^2 + \frac{b}{a + b}\|\boldsymbol{v}\|_2^2\right) + \delta_1^2\theta_1 + \delta_2^2\theta_2$$
$$= \|\boldsymbol{x} - \boldsymbol{y}\|_2^2 - (a + b)\left(\frac{a}{a + b}\|\boldsymbol{u}\|_2^2 + \left(1 - \frac{a}{a + b}\right)\|\boldsymbol{v}\|_2^2\right) + \delta_1^2\theta_1 + \delta_2^2\theta_2$$

Using the above results, we get,

$$\|T_1 \circ T_2(\boldsymbol{x}) - T_1 \circ T_2(\boldsymbol{y})\|_2^2$$
$$\le \|\boldsymbol{x} - \boldsymbol{y}\|_2^2 - (a + b)\frac{ab}{(a + b)^2}\|\boldsymbol{u} + \boldsymbol{v}\|_2^2 + \delta_1^2\theta_1 + \delta_2^2\theta_2$$
$$= \|\boldsymbol{x} - \boldsymbol{y}\|_2^2 - \frac{ab}{(a + b)}\|(I - T_1 \circ T_2)(\boldsymbol{x}) - (I - T_1 \circ T_2)(\boldsymbol{y})\|_2^2 + \delta_1^2\theta_1 + \delta_2^2\theta_2$$

Let $\theta = \frac{\theta_1 + \theta_2 - 2\theta_1\theta_2}{1 - \theta_1\theta_2}$. Then, we can see that $\frac{ab}{a+b} = \frac{1-\theta}{\theta}$.

$$\|T_1 \circ T_2(\boldsymbol{x}) - T_1 \circ T_2(\boldsymbol{y})\|_2^2 \leq \|\boldsymbol{x} - \boldsymbol{y}\|_2^2 - \frac{1-\theta}{\theta}\|(I - T_1 \circ T_2)(\boldsymbol{x}) - (I - T_1 \circ T_2)(\boldsymbol{y})\|_2^2$$
$$+ \delta^2\theta \qquad (25)$$

where, $\delta^2\theta = \delta_1^2\theta_1 + \delta_2^2\theta_2$. This implies that $T_1 \circ T_2$ is $\delta$-weakly $\theta$-averaged with

$$\theta = \frac{\theta_1 + \theta_2 - 2\theta_1\theta_2}{1 - \theta_1\theta_2}, \ \delta^2 = \frac{1}{\theta}(\delta_1^2\theta_1 + \delta_2^2\theta_2) \qquad (26)$$

$\square$

**Lemma 4** (Proposition 5.4 of Giselsson (2015)). *Assume $\ell$ is $\mu$-strongly convex, closed, and proper. Then, $-(2\mathrm{Prox}_{\frac{1}{\rho}\ell} - I)$ is $\frac{\rho}{\rho+\mu}$-averaged.*

**Lemma 5** (Pardo (2018)). *The KL divergence between two gaussian distributions $q_1 = \mathcal{N}(\boldsymbol{\mu}_1, \boldsymbol{\Sigma}_1)$ and $q_2 = \mathcal{N}(\boldsymbol{\mu}_2, \boldsymbol{\Sigma}_2)$ in $\mathbb{R}^d$ space is given by*

$$KL(q_1\|q_2) = \frac{1}{2}\left[\log\frac{|\boldsymbol{\Sigma}_2|}{|\boldsymbol{\Sigma}_1|} - d + tr(\boldsymbol{\Sigma}_2^{-1}\boldsymbol{\Sigma}_1) + (\boldsymbol{\mu}_2 - \boldsymbol{\mu}_1)^T\boldsymbol{\Sigma}_2^{-1}(\boldsymbol{\mu}_2 - \boldsymbol{\mu}_1)\right] \qquad (27)$$

*where $|\cdot|$ denotes the determinant, and tr denotes the trace of the matrix.*

# B  INFLUENCE OF AC-STEP ON BRINGING CLOSE TO $\{\mathcal{M}_{\sigma(t)}\}_{t=0}^T$

Given a noisy image $\widetilde{\boldsymbol{z}}^{(k)}$ at each iteration $k$, the denoising aims to recover the underlying clean image $\boldsymbol{z}_\natural^{(k)} \sim p_0(\boldsymbol{z})$ such that $\widetilde{\boldsymbol{z}}^{(k)} = \boldsymbol{z}_\natural^{(k)} + \boldsymbol{s}^{(k)}$, where $\boldsymbol{s}^{(k)}$ is the noise contained in $\widetilde{\boldsymbol{z}}^{(k)}$. The AC-step aims to bring $\boldsymbol{z}_{\mathrm{ac}}^{(k)}$ closer to the noisy distribution $\mathcal{M}_{\sigma^{(k)}}$ on which the $\boldsymbol{s}_{\boldsymbol{\theta}}(\cdot, \sigma^{(k)})$ was trained on. Lemma 6 shows that the AC-step tries to match with the distribution induced by the forward diffusion process.

**Lemma 6.** *The KL divergence between the target distribution $p(\boldsymbol{z}_{\sigma^{(k)}}|\boldsymbol{z}_\natural^{(k)})$ for correction steps and the distribution $p(\boldsymbol{z}_{\mathrm{ac}}^{(k)}|\widetilde{\boldsymbol{z}}^{(k)})$ induced by the approximate correction step in Algorithm 1 is given by*

$$KL(p(\boldsymbol{z}_{\sigma^{(k)}}|\boldsymbol{z}_\natural^{(k)})\|p(\boldsymbol{z}_{\mathrm{ac}}^{(k)}|\widetilde{\boldsymbol{z}}^{(k)})) = \frac{1}{2\left(\sigma^{(k)}\right)^2}\left\|\boldsymbol{s}^{(k)}\right\|_2^2 \qquad (28)$$

*Proof.*

$$\textbf{Target distribution: } p\left(\boldsymbol{z}_{\sigma^{(k)}}|\boldsymbol{z}_\natural^{(k)}\right) = \mathcal{N}\left(\boldsymbol{z}_\natural^{(k)}, \left(\sigma^{(k)}\right)^2 \boldsymbol{I}\right) \qquad (29)$$

$$\textbf{AC induced distribution: } p\left(\boldsymbol{z}_{\mathrm{ac}}^{(k)}|\widetilde{\boldsymbol{z}}^{(k)}\right) = \mathcal{N}\left(\widetilde{\boldsymbol{z}}^{(k)}, \left(\sigma^{(k)}\right)^2 \boldsymbol{I}\right) \qquad (30)$$

The KL divergence between these two distribution can be computed in closed form using Lemma 5.

$$\mathrm{KL}(q_1\|q_2) = \frac{1}{2}\left(\log 1 - d + \mathrm{tr}(\boldsymbol{I}) + (\widetilde{\boldsymbol{z}}^{(k)} - \boldsymbol{z}_\natural^{(k)})^T\left(\sigma^{(k)}\right)^{-2}\boldsymbol{I}(\widetilde{\boldsymbol{z}}^{(k)} - \boldsymbol{z}_\natural^{(k)})\right)$$
$$= \frac{1}{2}\left(0 - d + d + \left(\sigma^{(k)}\right)^{-2}\left\|\widetilde{\boldsymbol{z}}^{(k)} - \boldsymbol{z}_\natural^{(k)}\right\|_2^2\right)$$
$$= \frac{1}{2\left(\sigma^{(k)}\right)^2}\left\|\boldsymbol{s}^{(k)}\right\|_2^2$$

where, $\boldsymbol{s}^{(k)} = \widetilde{\boldsymbol{z}}^{(k)} - \boldsymbol{z}_\natural^{(k)}$.  $\square$

Lemma 6 shows that KL-gap of our approximate AC update; as long as $\sigma^{(k)}$ is sufficiently large, the two distributions remain close. Alkhouri et al. (2023)[Theorem 1] showed a result with a similar

flavor. Larger noise $\sigma^{(k)}$ makes the posterior nearly indistinguishable, but it also washes out fine structural details originally present (low *Signal-to-Noise Ratio* with larger $\sigma^{(k)}$). Existing works often use annealed scheduling $\sigma^{(k)} \downarrow 0$ (Zhu et al., 2023; Renaud et al., 2024a; Wang et al., 2024) to preserve image details, implicitly assuming $\left\|s^{(k)}\right\|_2^2$ decays at least as fast as $(\sigma^{(k)})^2$. With just the use of annealing $\sigma^{(k)}$ schedule, it is not sufficient to ensure that $z_{\text{ac}}^{(k)}$ lands in a desired manifold in each ADMM iteration. To bridge this gap, we propose to use DC-step in addition to the widely used annealing $\sigma^{(k)}$ schedule that explicitly corrects this gap.

## C    PROOF OF THEOREM 1

The proof involves showing that the each iteration of ADMM-PnP is weakly non-expansive when the denoiser satisfies Assumption 1. This weakly nonexpansiveness of each step leads to ball convergence of the algorithm.

Recall that the subproblems at $k$th iteration of ADMM-PnP is given by:

$$
\begin{aligned}
\boldsymbol{x}^{(k+1)} &= \arg\min_{\boldsymbol{x}} \frac{1}{\rho}\ell(\boldsymbol{y}\|\mathcal{A}(\boldsymbol{x})) + \frac{1}{2}\left\|\boldsymbol{x} - \boldsymbol{z}^{(k)} + \boldsymbol{u}^{(k)}\right\|_2^2 \\
&= \text{Prox}_{\frac{1}{\rho}\ell}(\boldsymbol{z}^{(k)} - \boldsymbol{u}^{(k)}) 
\end{aligned}
\tag{31a}
$$

$$
\begin{aligned}
\boldsymbol{z}^{(k+1)} &= \arg\min_{\boldsymbol{z}} \frac{\gamma}{\rho}h(\boldsymbol{z}) + \frac{1}{2}\left\|\boldsymbol{x}^{(k+1)} - \boldsymbol{z} + \boldsymbol{u}^{(k)}\right\|_2^2 \\
&= \text{Prox}_{\frac{\gamma}{\rho}h}(\boldsymbol{x}^{(k+1)} + \boldsymbol{u}^{(k)}) \\
&= D_{\sigma^{(k)}}(\boldsymbol{x}^{(k+1)} + \boldsymbol{u}^{(k)}) 
\end{aligned}
\tag{31b}
$$

$$
\boldsymbol{u}^{(k+1)} = \boldsymbol{u}^{(k)} + (\boldsymbol{x}^{(k+1)} - \boldsymbol{z}^{(k+1)})
\tag{31c}
$$

**Lemma 7** (Ryu et al. (2019)). *The steps of ADMM-PnP in (31) can be expressed as $\boldsymbol{v}^{(k+1)} = T(\boldsymbol{v}^{(k)})$ with $\boldsymbol{v}^{(k)} = \boldsymbol{z}^{(k)} - \boldsymbol{u}^{(k)}$ and*

$$
T = \frac{1}{2}I + \frac{1}{2}(2D_{\sigma^{(k)}} - I)(2\text{Prox}_{\frac{1}{\rho}\ell} - I)
\tag{32}
$$

**Lemma 8.** $D_\sigma : \mathbb{R}^d \to \mathbb{R}^d$ *satisfies Assumption 1 if and only if $\frac{1}{1+2\epsilon}(2D_{\sigma^{(k)}} - I)$ is $\Delta$-weakly $\theta$-averaged with $\theta = \frac{2\epsilon}{1+2\epsilon}$ and $\Delta^2 = 4\delta^2\frac{(1-\theta)^2}{\theta^2}$.*

*Proof.* We follow the similar proof structure as in Ryu et al. (2019). Let $\theta = \frac{2\epsilon}{1+2\epsilon}$ which implies $\epsilon = \frac{\theta}{2(1-\theta)}$. Here, we can clearly see that $\theta \in [0, 1)$. Let us define $G = \frac{1}{1+2\epsilon}(2D_{\sigma^{(k)}} - I)$ which implies $D_{\sigma^{(k)}} = \frac{1}{2(1-\theta)}G + \frac{1}{2}I$. Then,

$$
\begin{aligned}
&\|(D_{\sigma^{(k)}} - I)(\boldsymbol{x}) - (D_{\sigma^{(k)}} - I)(\boldsymbol{y})\|^2 - \epsilon^2\|\boldsymbol{x} - \boldsymbol{y}\|_2^2 \\
=&\|(D_{\sigma^{(k)}}(\boldsymbol{x}) - D_{\sigma^{(k)}}(\boldsymbol{y})) - (\boldsymbol{x} - \boldsymbol{y})\|_2^2 - \epsilon^2\|\boldsymbol{x} - \boldsymbol{y}\|_2^2 \\
=&\left\|\left(D_{\sigma^{(k)}} - \frac{1}{2}I\right)(\boldsymbol{x}) - \left(D_{\sigma^{(k)}} - \frac{1}{2}I\right)(\boldsymbol{y})\right\|_2^2 + \frac{1}{4}\|\boldsymbol{x} - \boldsymbol{y}\|_2^2 - \frac{\theta^2}{4(1-\theta)^2}\|\boldsymbol{x} - \boldsymbol{y}\|_2^2 \\
&- 2\left\langle\left(D_{\sigma^{(k)}} - \frac{1}{2}I\right)(\boldsymbol{x}) - \left(D_{\sigma^{(k)}} - \frac{1}{2}\right)(\boldsymbol{y}), \frac{1}{2}(\boldsymbol{x} - \boldsymbol{y})\right\rangle \\
=&\frac{1}{4(1-\theta)^2}\|G(\boldsymbol{x}) - G(\boldsymbol{y})\|_2^2 + \frac{1}{4}\left(1 - \frac{\theta^2}{(1-\theta)^2}\right)\|\boldsymbol{x} - \boldsymbol{y}\|_2^2 \\
&- \frac{1}{2(1-\theta)}\langle G(\boldsymbol{x}) - G(\boldsymbol{y}), \boldsymbol{x} - \boldsymbol{y}\rangle \\
=&\frac{1}{4(1-\theta)^2}\left(\|G(\boldsymbol{x}) - G(\boldsymbol{y})\|_2^2 - 2(1-\theta)\langle G(\boldsymbol{x}) - G(\boldsymbol{y}), \boldsymbol{x} - \boldsymbol{y}\rangle + (1-2\theta)\|\boldsymbol{x} - \boldsymbol{y}\|_2^2\right)
\end{aligned}
$$

Now,

$$\frac{1}{4(1-\theta)^2}\left(\|G(\boldsymbol{x})-G(\boldsymbol{y})\|_2^2 - 2(1-\theta)\langle G(\boldsymbol{x})-G(\boldsymbol{y}), \boldsymbol{x}-\boldsymbol{y}\rangle + (1-2\theta)\|\boldsymbol{x}-\boldsymbol{y}\|_2^2\right) \le \delta^2$$

$$\Leftrightarrow \|G(\boldsymbol{x})-G(\boldsymbol{y})\|_2^2 - 2(1-\theta)\langle G(\boldsymbol{x})-G(\boldsymbol{y}), \boldsymbol{x}-\boldsymbol{y}\rangle + (1-2\theta)\|\boldsymbol{x}-\boldsymbol{y}\|_2^2 \le 4\delta^2(1-\theta)^2$$

$$\Leftrightarrow \|G(\boldsymbol{x})-G(\boldsymbol{y})\|_2^2 - 2(1-\theta)\langle G(\boldsymbol{x})-G(\boldsymbol{y}), \boldsymbol{x}-\boldsymbol{y}\rangle + (1-2\theta)\|\boldsymbol{x}-\boldsymbol{y}\|_2^2 \le \Delta^2\theta^2$$

where, $\Delta^2 = 4\delta^2\frac{(1-\theta)^2}{\theta^2}$. From Lemma 2, this is equivalent to $G$ being $\Delta$-weakly $\theta$-averaged. $\quad\square$

### C.1 PROOF OF THE THEOREM

We follow the procedures in Ryu et al. (2019) and expand the results in the $\delta$-weakly expansive denoisers. We show that each iteration of PnP ADMM is also weakly nonexpansive when the denoiser satisfies Assumption 1.

*Proof.* From Assumption 1.

$$\|(D_{\sigma^{(k)}} - I)(\boldsymbol{x}) - (D_{\sigma^{(k)}} - I)(\boldsymbol{y})\|_2^2 \le \epsilon^2 \|\boldsymbol{x}-\boldsymbol{y}\|_2^2 + \delta^2 \tag{33}$$

From Lemma 4, we have $-(2\mathrm{Prox}_{\frac{1}{\rho}\ell} - I)$ is $\frac{\rho}{\rho+\mu}$-averaged.

Then, from Lemma 8, we have

$$\frac{1}{1+2\epsilon}\left(2D_{\sigma^{(k)}} - I\right) \tag{34}$$

is $\delta_1$-weakly $\theta$-averaged with $\theta = \frac{2\epsilon}{1+2\epsilon}$ and $\delta_1^2 = 4\delta^2\frac{(1-\theta)^2}{\theta^2}$.

By Lemma 3, it implies

$$-\frac{1}{1+2\epsilon}(2D_{\sigma^{(k)}} - I)(2\mathrm{Prox}_{\frac{1}{\rho}\ell} - I) \tag{35}$$

is $\delta_\circ$-weakly $\theta_\circ$-averaged with $\theta_\circ = \frac{\rho+2\mu\epsilon}{\rho+\mu+2\mu\epsilon}$ and $\delta_\circ^2 = \frac{1}{\theta_\circ}\cdot\frac{4\delta^2}{2\epsilon(1+2\epsilon)}$.

Now, using the definition of $\delta_\circ$-weakly $\theta_\circ$-averagedness, we have

$$(2D_{\sigma^{(k)}} - I)(2\mathrm{Prox}_{\frac{1}{\rho}\ell} - I) = -(1+2\epsilon)\left((1-\theta_\circ)I + \theta_\circ R\right)$$

$$= -(1+2\epsilon)\left(\frac{\mu}{\rho+\mu+2\mu\epsilon}I + \frac{\rho+2\mu\epsilon}{\rho+\mu+2\mu\epsilon}R\right)$$

where, $R$ is a certain $\delta_\circ$-weakly nonexpansive function.

Plugging this result into ADMM-PnP operator (Lemma 7), we get

$$T = \frac{1}{2}I + \frac{1}{2}(2D_{\sigma^{(k)}} - I)(2\mathrm{Prox}_{\frac{1}{\rho}\ell} - I)$$

$$= \frac{1}{2}I - \frac{1}{2}(1+2\epsilon)\left(\frac{\mu}{\rho+\mu+2\mu\epsilon}I + \frac{\rho+2\mu\epsilon}{\rho+\mu+2\mu\epsilon}R\right)$$

$$= \underbrace{\frac{\rho}{2(\rho+\mu+2\mu\epsilon)}}_{a}I - \underbrace{\frac{(1+2\epsilon)(\rho+2\mu\epsilon)}{2(\rho+\mu+2\mu\epsilon)}}_{b}R$$

where, clearly $a > 0$ and $b > 0$.

Now,

$$\|T(\boldsymbol{x})-T(\boldsymbol{y})\|_2^2 = a^2\|\boldsymbol{x}-\boldsymbol{y}\|_2^2 + b^2\|R(\boldsymbol{x})-R(\boldsymbol{y})\|_2^2 - 2\langle a(\boldsymbol{x}-\boldsymbol{y}), b(R(\boldsymbol{x})-R(\boldsymbol{y}))\rangle \tag{36}$$

From Young's inequality, for any $\gamma > 0$, we have

$$\langle a(\boldsymbol{x}-\boldsymbol{y}), b(R(\boldsymbol{x})-R(\boldsymbol{y}))\rangle \le \frac{1}{2\gamma}a^2\|\boldsymbol{x}-\boldsymbol{y}\|_2^2 + \frac{\gamma b^2}{2}\|R(\boldsymbol{x})-R(\boldsymbol{y})\|_2^2 \tag{37}$$

Plugging this, we get,

$$\|T(\boldsymbol{x}) - T(\boldsymbol{y})\|_2^2 \leq a^2\left(1 + \frac{1}{\gamma}\right)\|\boldsymbol{x} - \boldsymbol{y}\|_2^2 + b^2(1+\gamma)\|R(\boldsymbol{x}) - R(\boldsymbol{y})\|_2^2 \tag{38}$$

$$\leq \left(a^2\left(1 + \frac{1}{\gamma}\right) + b^2(1+\gamma)\right)\|\boldsymbol{x} - \boldsymbol{y}\|_2^2 + b^2(1+\gamma)\delta_\circ^2 \tag{39}$$

where, the second inequality is due to $\delta_\circ$-weak nonexpansiveness of $R$.

Note, that this holds for any $\gamma > 0$. When $\gamma = \frac{a}{b}$, we have

$$\left(a^2\left(1 + \frac{1}{\gamma}\right) + b^2(1+\gamma)\right) = (a+b)^2 \tag{40}$$

$$\begin{aligned}
&\|T(\boldsymbol{x}) - T(\boldsymbol{y})\|_2^2 \\
&\leq (a+b)^2\|\boldsymbol{x} - \boldsymbol{y}\|_2^2 + b^2\left(1 + \frac{a}{b}\right)\delta_\circ^2 \\
&= \underbrace{\left(\frac{\rho + \rho\epsilon + \mu\epsilon + 2\mu\epsilon^2}{\rho + \mu + 2\mu\epsilon}\right)^2}_{\epsilon_T^2}\|\boldsymbol{x} - \boldsymbol{y}\|_2^2 + \underbrace{\frac{(\rho + \rho\epsilon + \mu\epsilon + 2\mu\epsilon^2)\delta^2}{\epsilon(\rho + \mu + 2\mu\epsilon)}}_{\delta_T^2}
\end{aligned} \tag{41}$$

Hence, we have

$$\|T(\boldsymbol{x}) - T(\boldsymbol{y})\|_2^2 \leq \epsilon_T^2\|\boldsymbol{x} - \boldsymbol{y}\|_2^2 + \delta_T^2 \tag{42}$$

This shows that when $\epsilon_T \leq 1$, then, $\exists_{\boldsymbol{v}^*} \in \mathbb{R}^d$, and $K > 0$ such that $\forall k \geq K$ the following holds:

$$\left\|\boldsymbol{v}^{(k)} - \boldsymbol{v}^*\right\|_2^2 \leq \epsilon_T^{2k}\left\|\boldsymbol{v}^{(0)} - \boldsymbol{v}^*\right\|_2^2 + \frac{\delta_T^2}{1 - \epsilon_T^2}$$

$$\implies \lim_{k\to\infty}\left\|\boldsymbol{v}^{(k)} - \boldsymbol{v}^*\right\| \leq \frac{\delta_T}{\sqrt{1 - \epsilon_T^2}} \tag{43}$$

Hence with this we have the sequence $\{\boldsymbol{v}^{(k)} = \boldsymbol{z}^{(k)} - \boldsymbol{u}^{(k)}\}_{k\in\mathbb{N}+}$ converges within a ball of radius $\frac{\delta_T}{\sqrt{1-\epsilon_T^2}}$. Since, $-(2\text{Prox}_{\frac{1}{\rho}\ell} - I)$ is $\frac{\rho}{\rho+\mu}$-averaged, this implies

$$\text{Prox}_{\frac{1}{\rho}\ell} = \frac{1}{2}\frac{\rho}{\rho + \mu}(I - R) \tag{44}$$

for some nonexpansive function $R$. With this, we have

$$\lim_{k\to\infty}\left\|\text{Prox}_{\frac{1}{\rho}\ell}(\boldsymbol{v}^{(k)}) - \text{Prox}_{\frac{1}{\rho}\ell}(\boldsymbol{v}^*)\right\|_2^2 \leq \left(\frac{\rho}{\rho+\mu}\right)^2\left\|\boldsymbol{v}^{(k)} - \boldsymbol{v}^*\right\|_2^2$$

$$\implies \lim_{k\to\infty}\left\|\boldsymbol{x}^{(k)} - \boldsymbol{x}^*\right\|_2 \leq \frac{\rho}{\rho+\mu}\frac{\delta_T}{\sqrt{1 - \epsilon_T^2}} \tag{45}$$

where $\boldsymbol{x}^* = \text{Prox}_{\frac{1}{\rho}\ell}(\boldsymbol{v}^*)$.

With these results, we know there exists $\boldsymbol{u}^*$ such that

$$\lim_{k\to\infty}\left\|\boldsymbol{u}^k - \boldsymbol{u}^*\right\|_2 \leq \left(1 + \frac{\rho}{\rho+\mu}\right)\frac{\delta_T}{\sqrt{1 - \epsilon_T^2}} \tag{46}$$

$\square$

## D  PROOF OF THEOREM 2

Here, we show that our 3-step AC-DC denoiser satisfies Assumption 1 for constants $\epsilon$ and $\delta$. In the following, we first show that each step satisfies the weakly nonexpansive assumption. Therefore, the concatenation of these 3 steps meets Assumption 1.

**Lemma 9.** *Assume the* Variance Exploding (VE) scheduling *(Karras et al., 2022) is used in the diffusion model. Given that the log-density* $\log p_0$ *(i.e.* $\log p_{\text{data}}$) *is $M$-smooth (Assumption 2), the intermediate noisy log-densities* $\{\log p_t\}$ *are $M_t$-smooth for $t \in [0, T]$ i.e.* $\|\nabla \log p_t(\boldsymbol{x}) - \nabla \log p_t(\boldsymbol{y})\|_2 \leq M_t \|\boldsymbol{x} - \boldsymbol{y}\|_2$ *for all* $\boldsymbol{x}, \boldsymbol{y} \in \mathcal{X}$. *For $t$ such that $\sigma^2(t) < 1/M$, the smoothness constant $M_t$ can be upper bounded as*

$$M_t \leq \frac{M}{1 + M\sigma^2(t)} \leq M \tag{47}$$

*Proof.* From Tweedie's lemma, we have,

$$\implies \nabla_{\boldsymbol{x}_t} \log p_t(\boldsymbol{x}_t) = -\frac{1}{\sigma^2(t)} \left( \boldsymbol{x}_t - \mathbb{E}[\boldsymbol{x}_0 | \boldsymbol{x}_t] \right)$$

$$\implies \nabla_{\boldsymbol{x}_t}^2 \log p_t(\boldsymbol{x}_t) = -\frac{1}{\sigma^2(t)} (\boldsymbol{I} - \nabla_{\boldsymbol{x}_t} \mathbb{E}[\boldsymbol{x}_0 | \boldsymbol{x}_t]) \tag{48}$$

Now, let us evaluate the Jacobian $\nabla_{\boldsymbol{x}_t} \mathbb{E}[\boldsymbol{x}_0 | \boldsymbol{x}_t]$,

$$\nabla_{\boldsymbol{x}_t} \mathbb{E}[\boldsymbol{x}_0 | \boldsymbol{x}_t] = \nabla_{\boldsymbol{x}_t} \int_{\boldsymbol{x}_0 \in \mathcal{X}} \boldsymbol{x}_0 p(\boldsymbol{x}_0 | \boldsymbol{x}_t) d\boldsymbol{x}_0$$

$$= \int_{\boldsymbol{x}_0 \in \mathcal{X}} \boldsymbol{x}_0 \left( \nabla_{\boldsymbol{x}_t} \frac{p(\boldsymbol{x}_t | \boldsymbol{x}_0)}{p_t(\boldsymbol{x}_t)} \right) p_0(\boldsymbol{x}_0) d\boldsymbol{x}_0$$

$$= \int_{\boldsymbol{x}_0 \in \mathcal{X}} \boldsymbol{x}_0 \left( \frac{1}{p_t(\boldsymbol{x}_t)} \nabla_{\boldsymbol{x}_t} p(\boldsymbol{x}_t | \boldsymbol{x}_0) - p(\boldsymbol{x}_t | \boldsymbol{x}_0) \frac{1}{p_t^2(\boldsymbol{x}_t)} \nabla_{\boldsymbol{x}_t} p_t(\boldsymbol{x}_t) \right) p_0(\boldsymbol{x}_0) d\boldsymbol{x}_0 \tag{49}$$

Given $\boldsymbol{x}_t | \boldsymbol{x}_0 \sim \mathcal{N}(\boldsymbol{x}_0, \sigma(t)^2 \boldsymbol{I})$ and $p(\boldsymbol{x}_t) = \int_{\boldsymbol{x}_0 \in \mathcal{X}} p(\boldsymbol{x}_0) p(\boldsymbol{x}_t | \boldsymbol{x}_0) d\boldsymbol{x}_0$, we can compute their gradient (similar to Peng et al. (2024)) as:

$$\nabla_{\boldsymbol{x}_t} p(\boldsymbol{x}_t | \boldsymbol{x}_0) = -\frac{1}{\sigma^2(t)} p(\boldsymbol{x}_t | \boldsymbol{x}_0)(\boldsymbol{x}_t - \boldsymbol{x}_0)$$

$$\nabla_{\boldsymbol{x}_t} p(\boldsymbol{x}_t) = \int_{\boldsymbol{x}_0 \in \mathcal{X}} p_0(\boldsymbol{x}_0) \nabla_{\boldsymbol{x}_t} p(\boldsymbol{x}_t | \boldsymbol{x}_0) d\boldsymbol{x}_0$$

$$= \int_{\boldsymbol{x}_0 \in \mathcal{X}} p_0(\boldsymbol{x}_0) \left( -\frac{1}{\sigma^2(t)} p(\boldsymbol{x}_t | \boldsymbol{x}_0)(\boldsymbol{x}_t - \boldsymbol{x}_0) \right) d\boldsymbol{x}_0$$

$$= -\frac{1}{\sigma^2(t)} p_t(\boldsymbol{x}_t) \mathbb{E}[\boldsymbol{x}_t - \boldsymbol{x}_0 | \boldsymbol{x}_t] \tag{50}$$

Plugging these results in (49) and using integration by parts,

$$\nabla_{\boldsymbol{x}_t} \mathbb{E}[\boldsymbol{x}_0 | \boldsymbol{x}_t] = \frac{1}{p(\boldsymbol{x}_t)} \int_{\boldsymbol{x}_0 \in \mathcal{X}} \boldsymbol{x}_0 \nabla_{\boldsymbol{x}_t} p(\boldsymbol{x}_t | \boldsymbol{x}_0) p(\boldsymbol{x}_0) d\boldsymbol{x}_0 - \frac{\nabla_{\boldsymbol{x}_t} p(\boldsymbol{x}_t)}{p(\boldsymbol{x}_t)} \mathbb{E}[\boldsymbol{x}_0 | \boldsymbol{x}_t] \tag{51}$$

Substituting $\nabla_{\boldsymbol{x}_t} p(\boldsymbol{x}_t | \boldsymbol{x}_0) = -1/(\sigma^2(t))(\boldsymbol{x}_t - \boldsymbol{x}_0) p(\boldsymbol{x}_t | \boldsymbol{x}_0)$, we have

$$\nabla_{\boldsymbol{x}_t} \mathbb{E}[\boldsymbol{x}_0 | \boldsymbol{x}_t] = -\frac{1}{\sigma^2(t) p(\boldsymbol{x}_t)} \int_{\boldsymbol{x}_0 \in \mathcal{X}} \boldsymbol{x}_0 (\boldsymbol{x}_t - \boldsymbol{x}_0) p(\boldsymbol{x}_t | \boldsymbol{x}_0) p(\boldsymbol{x}_0) d\boldsymbol{x}_0 - \frac{\nabla_{\boldsymbol{x}_t} p(\boldsymbol{x}_t)}{p(\boldsymbol{x}_t)} \mathbb{E}[\boldsymbol{x}_0 | \boldsymbol{x}_t]$$

$$= \frac{-1}{\sigma^2(t)} \mathbb{E}[\boldsymbol{x}_0 (\boldsymbol{x}_t - \boldsymbol{x}_0) | \boldsymbol{x}_t] - \frac{\nabla_{\boldsymbol{x}_t} p(\boldsymbol{x}_t)}{p(\boldsymbol{x}_t)} \mathbb{E}[\boldsymbol{x}_0 | \boldsymbol{x}_t] \tag{52}$$

Substituting $\nabla_{\boldsymbol{x}_t} p(\boldsymbol{x}_t) = \int_{\boldsymbol{x}_0 \in \mathcal{X}} p(\boldsymbol{x}_0) \nabla_{\boldsymbol{x}_t} p(\boldsymbol{x}_t | \boldsymbol{x}_0) d\boldsymbol{x}_0 = -p_t(\boldsymbol{x}_t)/\sigma^2(t) \mathbb{E}[\boldsymbol{x}_t - \boldsymbol{x}_0 | \boldsymbol{x}_t]$, we have

$$\nabla_{\boldsymbol{x}_t} \mathbb{E}[\boldsymbol{x}_0 | \boldsymbol{x}_t] = \frac{-1}{\sigma^2(t)} \mathbb{E}[\boldsymbol{x}_0 (\boldsymbol{x}_t - \boldsymbol{x}_0) | \boldsymbol{x}_t] + \frac{1}{\sigma^2(t)} \mathbb{E}[\boldsymbol{x}_0 | \boldsymbol{x}_t] \mathbb{E}[\boldsymbol{x}_t - \boldsymbol{x}_0 \boldsymbol{x}_t]$$

$$= -\frac{1}{\sigma^2(t)} \text{Cov}(\boldsymbol{x}_0, \boldsymbol{x}_t - \boldsymbol{x}_0)$$

$$= \frac{1}{\sigma^2(t)} \text{Cov}(\boldsymbol{x}_0 | \boldsymbol{x}_t) \tag{53}$$

Note that this result in (53) is similar to Dytso et al. (2021)[Proposition 1], and has also been derived in Hatsell & Nolte (1971); Palomar & Verdu (2006).

From Assumption 2, we have

$$-M\boldsymbol{I} \preceq \nabla^2_{\boldsymbol{x}_0} \log p_0(\boldsymbol{x}_0) \preceq M\boldsymbol{I}, \ \forall \boldsymbol{x}_0 \in \mathcal{X} \tag{54}$$

where, $M > 0$ is a constant.

Then, let's analyze the hessian of the log of posterior distribution $p(\boldsymbol{x}_0|\boldsymbol{x}_t)$,

$$\nabla^2_{\boldsymbol{x}_0} \log p(\boldsymbol{x}_0|\boldsymbol{x}_t) = \nabla^2_{\boldsymbol{x}_0} \log p_0(\boldsymbol{x}_0) + \nabla^2_{\boldsymbol{x}_0} \log p(\boldsymbol{x}_t|\boldsymbol{x}_0) \tag{55}$$

$$\implies -M\boldsymbol{I} + \frac{1}{\sigma^2(t)}\boldsymbol{I} \preceq \nabla^2_{\boldsymbol{x}_0} \log p(\boldsymbol{x}_0|\boldsymbol{x}_t) \preceq M\boldsymbol{I} + \frac{1}{\sigma^2(t)}\boldsymbol{I} \tag{56}$$

When $M < \dfrac{1}{\sigma^2(t)}$, then the distribution $\log p(\boldsymbol{x}_0|\boldsymbol{x}_t)$ is strongly log-concave. In this case, the covariance of distribution $p(\boldsymbol{x}_0|\boldsymbol{x}_t)$ can be bounded (Brascamp & Lieb, 1976) as

$$\left(M\boldsymbol{I} + \frac{1}{\sigma^2(t)}\boldsymbol{I}\right)^{-1} \preceq \mathrm{Cov}(\boldsymbol{x}_0|\boldsymbol{x}_t) \preceq \left(-M\boldsymbol{I} + \frac{1}{\sigma^2(t)}\boldsymbol{I}\right)^{-1} \tag{57}$$

Combining result from equations (48), (53) and (57), we get

$$\nabla^2_{\boldsymbol{x}_t} \log p_t(\boldsymbol{x}_t) = -\frac{1}{\sigma^2(t)}\left(\boldsymbol{I} - \nabla_{\boldsymbol{x}_t}\mathbb{E}[\boldsymbol{x}_0|\boldsymbol{x}_t]\right)$$

$$= -\frac{1}{\sigma^2(t)}\left(\boldsymbol{I} - \frac{1}{\sigma^2(t)}\mathrm{Cov}(\boldsymbol{x}_0|\boldsymbol{x}_t)\boldsymbol{I}\right)$$

$$\left\|\nabla^2_{\boldsymbol{x}_t} \log q_t(\boldsymbol{x}_t)\right\|_2 \leq \frac{1}{\sigma^2(t)}\left|\left(1 - \frac{1}{\sigma^2(t)}\cdot\frac{\sigma^2(t)}{M\sigma^2(t)+1}\right)\right|$$

$$= \frac{1}{\sigma^2(t)} \cdot \frac{M\sigma^2(t)}{1 + M(\sigma^2(t))}$$

$$= \frac{M}{1 + M\sigma^2(t)} \tag{58}$$

Hence, the smoothness constant of $\log q(\boldsymbol{x}_t)$ is upper bounded as $M_t \leq \dfrac{M}{1 + M\sigma^2(t)}$ i.e. $M_t \leq M$.

$\qquad\qquad\qquad\qquad\qquad\qquad\qquad\qquad\qquad\qquad\qquad\qquad\qquad\qquad\qquad\qquad\qquad\qquad\qquad\qquad\quad \square$

**Lemma 10.** *Let $H^{(k)}_{\mathrm{ac}} : \widetilde{\boldsymbol{z}}^{(k)} \mapsto \boldsymbol{z}^{(k)}_{\mathrm{ac}}$ denote the function corresponding to approximate correction in Algorithm 1. Then, with probability at least $1 - e^{-\nu_k}$, the following holds for any $\boldsymbol{x}, \boldsymbol{y} \in \mathcal{X}$*

$$\left\|(H^{(k)}_{\mathrm{ac}} - I)(\boldsymbol{x}) - (H^{(k)}_{\mathrm{ac}} - I)(\boldsymbol{y})\right\|^2_2 \leq (\delta^{(k)}_{\mathrm{ac}})^2 + (\epsilon^{(k)}_{\mathrm{ac}})^2\|\boldsymbol{x} - \boldsymbol{y}\|^2_2 \tag{59}$$

*where, ${\delta^{(k)}_{\mathrm{ac}}}^2 = 2(\sigma^{(k)})^2(d + 2\sqrt{d\nu_k} + 2\nu_k)$, and $(\epsilon^{(k)}_{\mathrm{ac}})^2 = 0$.*

*Proof.* For any $\boldsymbol{x}, \boldsymbol{y} \in \mathcal{X}$, we have the residuals $R^{(k)}_{\mathrm{ac}}(\boldsymbol{x}) = (H^{(k)}_{\mathrm{ac}} - I)(\boldsymbol{x}) = \sigma^{(k)}\boldsymbol{n}_1$ and $R^{(k)}_{\mathrm{ac}}(\boldsymbol{y}) = (H^{(k)}_{\mathrm{ac}} - I)(\boldsymbol{y}) = \sigma^{(k)}\boldsymbol{n}_2$ where, $\boldsymbol{n}_1, \boldsymbol{n}_2 \overset{\mathrm{i.i.d.}}{\sim} \mathcal{N}(\boldsymbol{0}, \boldsymbol{I})$.

Then, we can bound the norm of difference of these two residuals as

$$\left\|R^{(k)}_{\mathrm{ac}}(\boldsymbol{x}) - R^{(k)}_{\mathrm{ac}}(\boldsymbol{y})\right\|^2_2 = \left\|\sigma^{(k)}\boldsymbol{n}_1 - \sigma^{(k)}\boldsymbol{n}_2\right\|^2_2$$

$$= (\sigma^{(k)})^2\|\boldsymbol{n}_{12}\|^2_2$$

$$= 2(\sigma^{(k)})^2\chi^2_d \tag{60}$$

where, $\boldsymbol{n}_{12} = \boldsymbol{n}_1 - \boldsymbol{n}_2 \sim \mathcal{N}(\boldsymbol{0}, 2\boldsymbol{I})$ and $\chi_d^2$ is standard chi-square distribution with $d$ degree of freedom.

From Laurent & Massart (2000)[Lemma 1], the following holds with probability at least $1 - e^{-\nu_k}$

$$\chi_d^2 \leq d + 2\sqrt{d\nu_k} + 2\nu_k \tag{61}$$

Plugging this in proves the lemma.

$\square$

**Lemma 11.** *Let $H_{\mathrm{dc}}^{(k)} : \boldsymbol{z}_{\mathrm{ac}}^{(k)} \mapsto \boldsymbol{z}_{\mathrm{dc}}^{(k)}$ denote the function corresponding to fine correction as defined in Algorithm 1. Then, with probability at least $1 - e^{-\nu_k}$, the following holds for any $\boldsymbol{x}, \boldsymbol{y} \in \mathcal{X}$ if $\sigma_{\boldsymbol{s}^{(k)}}^2 < \frac{1}{M_t}$:*

$$\left\| (H_{\mathrm{dc}}^{(k)} - I)(\boldsymbol{x}) - (H_{\mathrm{dc}}^{(k)} - I)(\boldsymbol{y}) \right\|_2^2 \leq (\delta_{\mathrm{dc}}^{(k)})^2 + (\epsilon_{\mathrm{dc}}^{(k)})^2 \|\boldsymbol{x} - \boldsymbol{y}\|_2^2 \tag{62}$$

*where, $(\delta_{\mathrm{dc}}^{(k)})^2 = \frac{32 d \sigma_{\boldsymbol{s}^{(k)}}^2}{(1 - M_t \sigma_{\boldsymbol{s}^{(k)}}^2)} \log \frac{2}{\nu_k}$, and $(\epsilon_{\mathrm{dc}}^{(k)})^2 = \left( \frac{\sqrt{2} M_t \sigma_{\boldsymbol{s}^{(k)}}^2}{1 - \sigma_{\boldsymbol{s}^{(k)}}^2 M_t} \right)^2$.*

*Proof.* Recall that the target distribution for this step is given by

$$\log p(\boldsymbol{z}_{\sigma^{(k)}} | \boldsymbol{z}_{\mathrm{ac}}^{(k)}) \propto \log p(\boldsymbol{z}_{\mathrm{ac}}^{(k)} | \boldsymbol{z}_{\sigma^{(k)}}) + \log p(\boldsymbol{z}_{\sigma^{(k)}}) \tag{63}$$

where, $p(\boldsymbol{z}_{\mathrm{ac}}^{(k)} | \boldsymbol{z}_{\sigma^{(k)}}) = \mathcal{N}(\boldsymbol{z}_{\sigma^{(k)}}, \sigma_{\boldsymbol{s}^{(k)}}^2 \boldsymbol{I})$.

Under Assumptions 2 and 3, the target distribution $p(\boldsymbol{z}_{\sigma^{(k)}} | \boldsymbol{z}_{\mathrm{ac}}^{(k)})$ also inherits smoothness and coercivity properties. These conditions imply the ergodicity of corresponding Langevin diffusion (Mattingly et al., 2002; Chen et al., 2020). In particular, Fokker-Planck equation (Uhlenbeck & Ornstein, 1930) characterizes $p(\boldsymbol{z}_{\sigma^{(k)}} | \boldsymbol{z}_{\mathrm{ac}}^{(k)})$ as its unique stationary distribution. Consequently, the iterates $\boldsymbol{z}_{\mathrm{dc}}^{(k)}$ obtained through Langevin dynamics converge to this distribution as the step size $\eta^{(k)} \to 0$ and the number of iterations $J \to \infty$. The gradient and hessian of the log of this desired distribution are given by

$$\nabla_{\boldsymbol{z}_{\mathrm{dc}}^{(k)}} \log p(\boldsymbol{z}_{\mathrm{dc}}^{(k)} | \boldsymbol{z}_{\mathrm{ac}}^{(k)}) = \frac{1}{\sigma_{\boldsymbol{s}^{(k)}}^2} (\boldsymbol{z}_{\mathrm{ac}}^{(k)} - \boldsymbol{z}_{\mathrm{dc}}^{(k)}) + \nabla_{\boldsymbol{z}_{\mathrm{dc}}^{(k)}} \log p_t(\boldsymbol{z}_{\mathrm{dc}}^{(k)}) \tag{64}$$

$$\nabla_{\boldsymbol{z}_{\mathrm{dc}}^{(k)}}^2 \log p(\boldsymbol{z}_{\mathrm{dc}}^{(k)} | \boldsymbol{z}_{\mathrm{ac}}^{(k)}) = -\frac{1}{\sigma_{\boldsymbol{s}^{(k)}}^2} + \nabla_{\boldsymbol{z}_{\mathrm{dc}}^{(k)}}^2 \log p_t(\boldsymbol{z}_{\mathrm{dc}}^{(k)}) \tag{65}$$

Here, $t$ refers to the noise level such that $\sigma(t) = \sigma^{(k)}$. By the $M_t$-smoothness of $\log p_t$ distribution (Lemma 9), we have

$$-\left( M_t + \frac{1}{\sigma_{\boldsymbol{s}^{(k)}}^2} \right) \boldsymbol{I} \preceq \nabla_{\boldsymbol{z}_{\mathrm{dc}}^{(k)}}^2 \log p(\boldsymbol{z}_{\mathrm{dc}}^{(k)} | \boldsymbol{z}_{\mathrm{ac}}^{(k)}) \preceq \left( M_t - \frac{1}{\sigma_{\boldsymbol{s}^{(k)}}^2} \right) \boldsymbol{I} \tag{66}$$

When $M_t < \frac{1}{\sigma_{\boldsymbol{s}^{(k)}}^2}$, the hessian is negative semi-definite that implies the distribution being log concave. This also implies that when $M_t \ll \frac{1}{\sigma_{\boldsymbol{s}^{(k)}}^2}$, the likelihood term dominates in the posterior (63).

Using (Wainwright, 2019)[Theorem 3.16], the following holds with probability at least $1 - \nu_k$

$$\left\| \boldsymbol{z}_{\mathrm{dc}}^{(k)} - \mathbb{E}[\boldsymbol{z}_{\mathrm{dc}}^{(k)} | \boldsymbol{z}_{\mathrm{ac}}^{(k)}] \right\|_2 \leq \sqrt{\frac{4}{\lambda_t} \log \frac{2}{\nu_k}} \tag{67}$$

where $\lambda_t = -M_t + \frac{1}{\sigma_{\boldsymbol{s}^{(k)}}^2}$.

Now, using Tweedie's lemma, we have

$$\mathbb{E}[\boldsymbol{z}_{\mathrm{dc}}^{(k)} | \boldsymbol{z}_{\mathrm{ac}}^{(k)}] = \boldsymbol{z}_{\mathrm{ac}}^{(k)} + \sigma_{\boldsymbol{s}^{(k)}}^2 \nabla_{\boldsymbol{z}_{\mathrm{ac}}^{(k)}} \log p(\boldsymbol{z}_{\mathrm{ac}}^{(k)})$$

$$\implies \mathbb{E}[\boldsymbol{z}_{\mathrm{dc}}^{(k)} | \boldsymbol{z}_{\mathrm{ac}}^{(k)}] - \boldsymbol{z}_{\mathrm{ac}}^{(k)} = \sigma_{\boldsymbol{s}^{(k)}}^2 \nabla_{\boldsymbol{z}_{\mathrm{ac}}^{(k)}} \log p(\boldsymbol{z}_{\mathrm{ac}}^{(k)}) \tag{68}$$

Combining the above two results, with probability at least $1 - 2\nu_k$, the difference of residual for $\boldsymbol{x}$ and $\boldsymbol{y}$ can be bounded as

$$
\begin{aligned}
\left\| R_{\mathrm{dc}}^{(k)}(\boldsymbol{x}) - R_{\mathrm{dc}}^{(k)}(\boldsymbol{y}) \right\|_2^2 &\leq 2 \left( 2\sqrt{\frac{4}{\lambda_t} \log \frac{2}{\nu_k}} \right)_2^2 + \left\| \sigma_{\boldsymbol{s}^{(k)}}^2 \left( \nabla_{\boldsymbol{x}} \log p_{\boldsymbol{z}_{\mathrm{ac}}^{(k)}}(\boldsymbol{x}) - \nabla_{\boldsymbol{y}} \log p_{\boldsymbol{z}_{\mathrm{ac}}^{(k)}}(\boldsymbol{y}) \right) \right\|_2^2 \\
&\leq \frac{32d}{\lambda_t} \log \frac{2}{\nu_k} + 2 \left\| \sigma_{\boldsymbol{s}^{(k)}}^2 \left( \nabla_{\boldsymbol{x}} \log p_{\boldsymbol{z}_{\mathrm{ac}}^{(k)}}(\boldsymbol{x}) - \nabla_{\boldsymbol{y}} \log p_{\boldsymbol{z}_{\mathrm{ac}}^{(k)}}(\boldsymbol{y}) \right) \right\|^2 \\
&\leq \frac{32d}{\lambda_t} \log \frac{2}{\nu_k} + 2\sigma_{\boldsymbol{s}^{(k)}}^4 M_{\boldsymbol{z}_{\mathrm{ac}}^{(k)}}^2 \|\boldsymbol{x} - \boldsymbol{y}\|^2
\end{aligned}
\tag{69}
$$

where, $M_{\boldsymbol{z}_{\mathrm{ac}}^{(k)}}$ is the smoothness constant of $\log p_{\boldsymbol{z}_{\mathrm{ac}}^{(k)}}$ that can be derived using a similar proof procedure as Lemma 9, and $\lambda_t = -M_t + \frac{1}{\sigma_{\boldsymbol{s}^{(k)}}^2}$.

Following the similar procedure as in proof of Lemma 9, we get,

$$
M_{\boldsymbol{z}_{\mathrm{ac}}^{(k)}} \leq \frac{M_t}{1 - \sigma_{\boldsymbol{s}^{(k)}}^2 M_t}
\tag{70}
$$

Plugging this leads to the lemma.

$\square$

**Lemma 12.** *Let $H_{\mathrm{tw}}^{(k)} : \boldsymbol{z}_{\mathrm{dc}}^{(k)} \mapsto \boldsymbol{z}_{\mathrm{tw}}^{(k)}$ denote the projection function using Tweedie's lemma defined in Algorithm 1. Then, we have the following*

$$
\left\| (H_{\mathrm{tw}}^{(k)} - I)(\boldsymbol{x}) - (H_{\mathrm{tw}}^{(k)} - I)(\boldsymbol{y}) \right\|_2^2 \leq (\epsilon_{\mathrm{tw}}^{(k)})^2 \|\boldsymbol{x} - \boldsymbol{y}\|_2^2 + {\delta_{\mathrm{tw}}^{(k)}}^2
\tag{71}
$$

*for any $\boldsymbol{x}, \boldsymbol{y} \in \mathcal{X}$, where $(\epsilon_{\mathrm{tw}}^{(k)})^2 = (\sigma^{(k)})^4 M_t^2$, and ${\delta_{\mathrm{tw}}^{(k)}}^2 = 0$.*

*Proof.* From Tweedie's lemma, we have

$$
\begin{aligned}
\boldsymbol{z}_{\mathrm{tw}}^{(k)} &= \mathbb{E}[\boldsymbol{z}_0 | \boldsymbol{z}_t = \boldsymbol{z}_{\mathrm{dc}}^{(k)}] \\
&= \boldsymbol{z}_{\mathrm{dc}}^{(k)} + (\sigma^{(k)})^2 \nabla \log p_t(\boldsymbol{z}_{\mathrm{dc}}^{(k)})
\end{aligned}
$$

where, $t \in [0, T]$ such that $\sigma^{(k)} = \sigma(t)$.

Then, the residuals are given by

$$
R_{\mathrm{tw}}^{(k)}(\boldsymbol{x}) = (\sigma^{(k)})^2 \nabla \log p_t(\boldsymbol{x})
\tag{72}
$$

$$
R_{\mathrm{tw}}^{(k)}(\boldsymbol{y}) = (\sigma^{(k)})^2 \nabla \log p_t(\boldsymbol{y})
\tag{73}
$$

Now, the norm of the difference of the residuals can be written as

$$
\left\| R_{\mathrm{tw}}^{(k)}(\boldsymbol{x}) - R_{\mathrm{tw}}^{(k)}(\boldsymbol{y}) \right\|_2^2 = (\sigma^{(k)})^4 \|\nabla \log p_t(\boldsymbol{x}) - \nabla \log p_t(\boldsymbol{y})\|_2^2
\tag{74}
$$

$$
\leq (\sigma^{(k)})^4 M_t^2 \|\boldsymbol{x} - \boldsymbol{y}\|_2^2
\tag{75}
$$

where, $M_t$ is the smoothness constant of $\log p_t(\boldsymbol{x})$. $\square$

### D.1 MAIN PROOF:

*Proof.* **Part (a)** Using Lemma 10, 11, and 12, with probability at least $1 - 2e^{-\nu_k}$, we have

$$
\begin{aligned}
&\|R_{\sigma^{(k)}}(\boldsymbol{x}) - R_{\sigma^{(k)}}(\boldsymbol{y})\|_2^2 \\
\leq &3 \left\| R_{\mathrm{ac}}^{(k)}(\boldsymbol{x}) - R_{\mathrm{ac}}^{(k)}(\boldsymbol{y}) \right\|_2^2 + 3 \left\| R_{\mathrm{dc}}^{(k)}(\boldsymbol{x}) - R_{\mathrm{dc}}^{(k)}(\boldsymbol{y}) \right\|_2^2 + 3 \left\| R_{\mathrm{tw}}^{(k)}(\boldsymbol{x}) - R_{\mathrm{tw}}^{(k)}(\boldsymbol{y}) \right\|_2^2 \\
\leq &3((\epsilon_{\mathrm{ac}}^{(k)})^2 + (\epsilon_{\mathrm{dc}}^{(k)})^2 + (\epsilon_{\mathrm{tw}}^{(k)})^2) \|\boldsymbol{x} - \boldsymbol{y}\|_2^2 + 3({\delta_{\mathrm{ac}}^{(k)}}^2 + {\delta_{\mathrm{dc}}^{(k)}}^2 + {\delta_{\mathrm{tw}}^{(k)}}^2) \\
\leq &\epsilon_k^2 \|\boldsymbol{x} - \boldsymbol{y}\|_2^2 + \delta_k^2
\end{aligned}
\tag{76}
$$

where,

$$\epsilon_k^2 = 3\left(\left(\frac{\sqrt{2}M_t\sigma_{\boldsymbol{s}^{(k)}}^2}{1 - \sigma_{\boldsymbol{s}^{(k)}}^2 M_t}\right)^2 + (\sigma^{(k)})^4 M_t^2\right)$$

$$\delta_k^2 = 3\left(2(\sigma^{(k)})^2(d + 2\sqrt{d\nu_k} + 2\nu_k) + \frac{32d\sigma_{\boldsymbol{s}^{(k)}}^2}{(1 - M_t\sigma_{\boldsymbol{s}^{(k)}}^2)}\log\frac{2}{\nu_k}\right)$$

Using Lemma 9 leads to the final theorem.

**Part (b).** Let us set $\nu_k = \ln\left(\frac{2\pi^2}{6\eta}\right) + 2\ln k$. With this, the above weakly nonexpansiveness holds for all $k \in \mathbb{N}^+$ with probability at least

$$1 - \sum_{k=1}^{\infty} 2e^{-\ln\left(\frac{2\pi^2}{6\eta}\right) - 2\ln k}$$

$$= 1 - \frac{6\eta}{\pi^2} \times \sum_{k=1}^{\infty} \frac{1}{k^2} \tag{77}$$

Using Riemann zeta function (Titchmarsh & Heath-Brown, 1986) at value 2, we have

$$\zeta(2) = \sum_{k=1}^{\infty} \frac{1}{k^2} = \frac{\pi^2}{6} \tag{78}$$

Plugging this in we get the probability to be at least $1 - \eta$. Now, combining the results with Theorem 1 leads to the final proof of this part. $\qquad\square$

## E    PROOF OF THEOREM 3

Here, show that our 3-step AC-DC denoiser is bounded with high probability. We first show that each of 3 steps are bounded, and then combined them to establish the boundedness of our AC-DC denoiser as a whole. And following the boundedness, we show that AC-DC ADMM-PnP converges to a fixed point with proper scheduling of $\sigma^{(k)}$ and $\sigma_{\boldsymbol{s}^{(k)}}$.

**Lemma 13** (Uniform score bound). *Suppose Assumption 2 holds. Let*

$$D := \operatorname{diam}(\mathcal{X}) = \sup_{\boldsymbol{x},\boldsymbol{y}\in\mathcal{X}} \|\boldsymbol{x} - \boldsymbol{y}\|_2 < \infty \text{ and } S := \inf_{\boldsymbol{x}\in\mathcal{X}} \|\nabla\log p_{\text{data}}(\boldsymbol{x})\|_2 < \infty.$$

*Then, with $L = MD + S$, we have*

$$\sup_{\boldsymbol{x}\in\mathcal{X}} \|\nabla\log p_{\text{data}}(\boldsymbol{x})\|_\infty \leq L.$$

*Proof.* From Assumption 2, we have

$$\|\nabla\log p_{\text{data}}(\boldsymbol{x}) - \nabla\log p_{\text{data}}(\boldsymbol{y})\|_2 \leq M\|\boldsymbol{x} - \boldsymbol{y}\|_2, \ \forall\boldsymbol{x},\boldsymbol{y}\in\mathcal{X} \tag{79}$$

Fix any $\boldsymbol{x}_0 \in \mathcal{X}$. By the triangle inequality, for all $\boldsymbol{x} \in \mathcal{X}$,

$$\|\nabla\log p_{\text{data}}(\boldsymbol{x})\|_2 \leq \|\nabla\log p_{\text{data}}(\boldsymbol{x}) - \nabla\log p_{\text{data}}(\boldsymbol{x}_0)\|_2 + \|\nabla\log p_{\text{data}}(\boldsymbol{x}_0)\|_2$$
$$\leq M\|\boldsymbol{x} - \boldsymbol{x}_0\|_2 + \|\nabla\log p_{\text{data}}(\boldsymbol{x}_0)\|_2 \tag{80}$$

Taking the supremum over $\boldsymbol{x} \in \mathcal{X}$ and then the infimum over $\boldsymbol{x}_0 \in \mathcal{X}$ yields

$$\sup_{\boldsymbol{x}\in\mathcal{X}} \|\nabla\log p_{\text{data}}(\boldsymbol{x})\|_2 \leq \sup_{\boldsymbol{x}\in\mathcal{X}} M\|\boldsymbol{x} - \boldsymbol{x}_0\|_2 + \inf_{\boldsymbol{x}_0\in\mathcal{X}} \|\nabla\log p_{\text{data}}(\boldsymbol{x}_0)\|_2 \tag{81}$$

$$\leq MD + S \tag{82}$$

Because $\|\boldsymbol{u}\|_\infty \leq \|\boldsymbol{u}\|_2$ for any vector,

$$\sup_{\boldsymbol{x}\in\mathcal{X}} \|\nabla\log p_{\text{data}}(\boldsymbol{x})\|_\infty \leq MD + S \tag{83}$$

which proves the above lemma. $\qquad\square$

**Lemma 14.** *Assuming* $\|\nabla \log p_{\text{data}}(\boldsymbol{x})\|_\infty \leq, \forall \boldsymbol{x} \in \mathcal{X}$ *, the score of intermediate noisy distributions* $\{p_t\}_{t \in [0,T]}$ *are bounded as.*

$$\|\nabla \log p_t(\boldsymbol{x})\|_2 \leq \sqrt{d}L \tag{84}$$

*for all* $\boldsymbol{x} \in \mathcal{X}$.

*Proof.* We have $\boldsymbol{x}_t = \boldsymbol{x}_0 + \sigma(t)\boldsymbol{n}$ with $\boldsymbol{n} \sim \mathcal{N}(\boldsymbol{0}, \boldsymbol{I})$. Let us denote $\boldsymbol{n}_1 = \sigma(t)\boldsymbol{n}$. Then, the marginal distribution is given by the convolution of two distributions.

$$p_t(\boldsymbol{x}) = \int_{\boldsymbol{x}_1 \in \mathcal{X}} p_{\boldsymbol{n}_1}(\boldsymbol{x}_1) p_0(\boldsymbol{x} - \boldsymbol{x}_1) d\boldsymbol{x}_1 \tag{85}$$

Then, the score is given by

$$\begin{aligned}
\nabla \log p_t(\boldsymbol{x}) &= \frac{\nabla p_t(\boldsymbol{x})}{p_t(\boldsymbol{x})} \\
&= \frac{1}{p_t(\boldsymbol{x})} \int_{\boldsymbol{x}_1 \in \mathcal{X}} p_{\boldsymbol{n}_1}(\boldsymbol{x}_1) \nabla_{\boldsymbol{x}} p_0(\boldsymbol{x} - \boldsymbol{x}_1) d\boldsymbol{x}_1 \\
&= \frac{1}{p_t(\boldsymbol{x})} \int_{\boldsymbol{x}_1 \in \mathcal{X}} p_{\boldsymbol{n}_1}(\boldsymbol{x}_1) p_0(\boldsymbol{x} - \boldsymbol{x}_1) \frac{\nabla_{\boldsymbol{x}} p_0(\boldsymbol{x} - \boldsymbol{x}_1)}{p_0(\boldsymbol{x} - \boldsymbol{x}_1)} d\boldsymbol{x}_1 \\
&= \frac{1}{p_t(\boldsymbol{x})} \int_{\boldsymbol{x}_1 \in \mathcal{X}} p_{\boldsymbol{n}_1}(\boldsymbol{x}_1) p_0(\boldsymbol{x} - \boldsymbol{x}_1) \nabla_{\boldsymbol{x}} \log p_0(\boldsymbol{x} - \boldsymbol{x}_1) d\boldsymbol{x}_1
\end{aligned} \tag{86}$$

Now the norm can be bounded as

$$\begin{aligned}
\|\nabla \log p_t(\boldsymbol{x})\|_2 &\leq \frac{1}{p_t(\boldsymbol{x})} \int_{\boldsymbol{x}_1 \in \mathcal{X}} p_{\boldsymbol{n}_1}(\boldsymbol{x}_1) p_0(\boldsymbol{x} - \boldsymbol{x}_1) \|\nabla_{\boldsymbol{x}} \log p_0(\boldsymbol{x} - \boldsymbol{x}_1)\|_2 d\boldsymbol{x}_1 \\
&\leq \sup_{\boldsymbol{x}_2 \in \mathcal{X}} \|\nabla_{\boldsymbol{x}} \log p_0(\boldsymbol{x}_2)\|_2 \frac{1}{p_t(\boldsymbol{x})} \int_{\boldsymbol{x}_1 \in \mathcal{X}} p_{\boldsymbol{n}_1}(\boldsymbol{x}_1) p_0(\boldsymbol{x} - \boldsymbol{x}_1) d\boldsymbol{x}_1 \\
&= \sqrt{d}L
\end{aligned} \tag{87}$$

The final equality is due to the fact $\|\boldsymbol{x}\|_2 \leq \sqrt{d}\|\boldsymbol{x}\|_\infty$.

$\square$

**Lemma 15.** *Let* $H_{\text{ac}}^{(k)} : \widetilde{\boldsymbol{z}}^{(k)} \mapsto \boldsymbol{z}_{\text{ac}}^{(k)}$ *denote the function corresponding to approximate correction to noise level* $\sigma^{(k)}$ *defined in Algorithm 1. Then, with probability at least* $1 - e^{-\nu}$*, the following holds for any* $\boldsymbol{x}, \boldsymbol{y} \in \mathcal{X}$

$$\frac{1}{d}\left\|(H_{\text{ac}}^{(k)} - I)(\boldsymbol{x})\right\|_2^2 \leq (\sigma^{(k)})^2(1 + 2\sqrt{\nu} + 2\nu) \tag{88}$$

*Proof.* For any $\boldsymbol{x} \in \mathcal{X}$, we have the residual $R_{\text{ac}}^{(k)}(\boldsymbol{x}) = (H_{\text{ac}}^{(k)} - I)(\boldsymbol{x}) = \sigma^{(k)}\boldsymbol{n}, \ \boldsymbol{n} \sim \mathcal{N}(\boldsymbol{0}, \boldsymbol{I})$. Then,

$$\left\|R_{\text{ac}}^{(k)}(\boldsymbol{x})\right\|_2^2 = (\sigma^{(k)})^2\|\boldsymbol{n}\|_2^2 = (\sigma^{(k)})^2\chi_d^2 \tag{89}$$

where $\chi_d^2$ is standard chi-square distribution with $d$ degrees of freedom. From Laurent & Massart (2000)[Lemma 1], the following holds with probability at least $1 - e^{-\nu}$

$$\chi_d^2 \leq d + 2\sqrt{d\nu} + 2\nu \tag{90}$$

This implies $\frac{1}{d}\left\|H_{\text{ac}}^{(k)} - I)(\boldsymbol{x})\right\|_2^2 \leq (\sigma^{(k)})^2(1 + 2\sqrt{1\nu} + 2\nu)$ with probability at least $1 - e^{-\nu}$ due to $d \geq 1$.

$\square$

**Lemma 16.** *Let* $H_{\text{dc}}^{(k)} : \boldsymbol{z}_{\text{ac}}^{(k)} \mapsto \boldsymbol{z}_{\text{dc}}^{(k)}$ *denote the function corresponding to fine correction defined in Algorithm 1. Assume* $\|\nabla \log p_{\text{data}}(\boldsymbol{x})\|_\infty \leq L$*. Then, with probability at least* $1 - e^{-\nu}$*, the following holds for any* $\boldsymbol{x}, \boldsymbol{y} \in \mathcal{X}$*:*

$$\frac{1}{d}\left\|(H_{\text{dc}}^{(k)} - I)(\boldsymbol{x})\right\|_2^2 \leq \frac{8}{\lambda_t}\log\frac{2}{\nu} + \sigma_{\boldsymbol{s}^{(k)}}^4 L^2 \tag{91}$$

*where,* $\lambda_t = -M_t + \frac{1}{\sigma_{\boldsymbol{s}^{(k)}}^2}$.

*Proof.* From (67), with probability at least $1 - 2\nu$, the norm can be bounded as

$$\left\|R_{\mathrm{dc}}^{(k)}(\boldsymbol{x})\right\|_2^2 \leq \frac{8d}{\lambda_t} \log \frac{2}{\nu} + \sigma_{\boldsymbol{s}^{(k)}}^4 \left\|\nabla_{\boldsymbol{x}} \log p_{\boldsymbol{z}_{\mathrm{ac}}^{(k)}}(\boldsymbol{x})\right\|_2^2 \tag{92}$$

Then, using Lemma 14, we have

$$\frac{1}{d}\left\|R_{\mathrm{dc}}^{(k)}(\boldsymbol{x})\right\|_2^2 \leq \frac{8}{\lambda_t} \log \frac{2}{\nu} + \sigma_{\boldsymbol{s}^{(k)}}^4 L^2 \tag{93}$$

$\square$

**Lemma 17.** *Let $H_{\mathrm{tw}}^{(k)} : \boldsymbol{z}_{\mathrm{dc}}^{(k)} \mapsto \boldsymbol{z}_{\mathrm{tw}}^{(k)}$ denote the projection function using Tweedie's lemma defined in Algorithm 1. Assume $\|\nabla \log p_{\mathrm{data}}(\boldsymbol{x})\|_\infty \leq L$. Then, we have the following*

$$\frac{1}{d}\left\|(H_{\mathrm{tw}}^{(k)} - I)(\boldsymbol{x})\right\|_2^2 \leq (\sigma^{(k)})^4 L^2 \tag{94}$$

*for any $\boldsymbol{x} \in \mathcal{X}$.*

*Proof.* From Tweedie's lemma, we have

$$\begin{aligned}
\boldsymbol{z}_{\mathrm{tw}}^{(k)} &= \mathbb{E}[\boldsymbol{z}_0 | \boldsymbol{z}_t = \boldsymbol{z}_{\mathrm{dc}}^{(k)}] \\
&= \boldsymbol{z}_{\mathrm{dc}}^{(k)} + (\sigma^{(k)})^2 \nabla \log p_t(\boldsymbol{z}_{\mathrm{dc}}^{(k)})
\end{aligned} \tag{95}$$

where, $t \in [0, T]$ such that $\sigma^{(k)} = \sigma(t)$.

Now, the norm of residual can be written as

$$\begin{aligned}
\left\|R_{\mathrm{tw}}^{(k)}(\boldsymbol{x})\right\|_2^2 &= \left\|(\sigma^{(k)})^2 \nabla \log p_t(\boldsymbol{x})\right\|_2^2 \\
&\leq (\sigma^{(k)})^4 L^2 \cdot d
\end{aligned} \tag{96}$$

where, $L$ bound of gradient from Lemma 14. $\square$

### E.1 MAIN PROOF

Combining Lemmas 15, 16 and 17 leads to the proof of part (a) of Theorem 3.

With probability at least $1 - 2e^{-\nu_k}$, the denoiser satisfies the bounded residual condition.

$$1/d\|(D_{\sigma^{(k)}} - I)(\boldsymbol{x})\|_2^2 \leq c_k^2. \tag{97}$$

Let's define the relative residue as:

$$\beta_k := \frac{1}{\sqrt{d}} \left( \left\|\boldsymbol{x}^{(k)} - \boldsymbol{x}^{(k-1)}\right\|_2 + \left\|\boldsymbol{z}^{(k)} - \boldsymbol{z}^{(k-1)}\right\|_2 + \left\|\boldsymbol{u}^{(k)} - \boldsymbol{u}^{(k-1)}\right\|_2 \right) \tag{98}$$

For any $\eta \in [0, 1)$ and a constant $\gamma > 1$, the penalty parameter $\rho_k$ is adjusted at each iteration $k$ according to following rule (Chan et al., 2016):

$$\rho_{k+1} = \begin{cases} \gamma\rho_k & \text{if } \beta_{k+1} \geq \eta\beta_k \quad (\text{Case 1}) \\ \rho_k & \text{else} \quad (\text{Case 2}) \end{cases} \tag{99}$$

The PnP-ADMM with adaptive penalty involves two cases as shown above.

At iteration $k$, if Case 1 holds, then by Lemma 18 we have

$$\beta_{k+1} \leq 6c_k + 2c_{k-1} + \frac{2R}{\rho_k} \tag{100}$$

On the other hand if Case 2 holds, then,

$$\beta_{k+1} \leq \eta\beta_k \tag{101}$$

Define $a_k = 6c_k + 2c_{k-1} + \frac{2R}{\rho_k}$. Combining two cases, we get

$$\beta_{k+1} \leq \delta\beta_k + a_k, \quad \delta = \begin{cases} \eta, & \text{if Case 2 holds at iteration } k \\ 0, & \text{if Case 1 holds at iteration } k \end{cases} \tag{102}$$

Note that for Case 1 $\rho_{k+1} = \gamma\rho_k$ and with $\gamma > 1$, we get $\lim_{k\to\infty} \frac{c}{\rho_k} = 0$. In addition, with $\nu_k = \ln\frac{2\pi^2}{6\eta} + 2\ln k$, and the scheduling of $\sigma^{(k)}$, $\sigma_{\boldsymbol{s}^{(k)}}$ that satisfies

$$\lim_{k\to\infty}(\sigma^{(k)})^2(2 + 4\sqrt{\nu_k} + 4\nu_k) = 0, \quad \lim_{k\to\infty}\frac{\sigma^2_{\boldsymbol{s}^{(k)}}}{1 - M\sigma^2_{\boldsymbol{s}^{(k)}}}\log\frac{2}{\nu_k} = 0, \quad \lim_{k\to\infty}\sigma^{(k)} = 0, \quad \lim_{k\to\infty}\sigma_{\boldsymbol{s}^{(k)}} = 0$$

results in $\lim_{k\to\infty} a_k = 0$.

As $k \to \infty$, 3 different scenarios could occur. Let's analyze each of the scenarios one by one.

- Scene 1: Case 1 occurs infinitely many times and Case 2 occurs finitely many times. When Scene 1 occurs, then, there exists a constant $K_1 > 0$ such that $\beta_{k+1} \leq a_k$ for all $k \geq K_1$. Since $\lim_{k\to\infty} a_k = 0$, this leads to $\lim_{k\to\infty}\beta_k = 0$.

- Scene 2: Case 2 occurs infinitely many times and Case 1 occurs finitely many times. Similarly, there exists a constant $K_2 > 0$ such that $\beta_{k+1} \leq \eta\beta_k$ for all $k \geq K_2$. Then, we have
$$\beta_k \leq \eta^{k-K_2}\beta_{K_2} \tag{103}$$
And with $\eta \in [0, 1)$, we have $\lim_{k\to\infty}\beta_k = 0$.

- Scene 3: Both Case 1 and Case 2 occurs infinitely many times. With the Scene 1 and Scene 2 converging, the sequence $\lim_{k\to\infty}\beta_k = 0$ under this Scene as well.

This proves the part (b) of Theorem 3.

**Lemma 18.** *For any iteration $k$ that falls into Case 1, the following holds*

$$\beta_{k+1} \leq 6c_k + 2c_{k-1} + \frac{2R}{\rho_k} \tag{104}$$

*Proof.* Consider the subproblem (7a) with adaptive penalty parameter $\rho_k$ (as defined in (99)),

$$\boldsymbol{x}^{(k+1)} = \arg\min_{\boldsymbol{x}} \frac{1}{\rho_k}\ell(\boldsymbol{y}||\mathcal{A}(\boldsymbol{x})) + \frac{1}{2}\left\|\boldsymbol{x} - \boldsymbol{z}^{(k)} + \boldsymbol{u}^{(k)}\right\|_2^2 \tag{105}$$

With the first order optimality condition, the solution $\boldsymbol{x}^{(k+1)}$ satisfies

$$\frac{1}{\rho_k}\nabla_{\boldsymbol{x}}\ell(\boldsymbol{y}||\mathcal{A}(\boldsymbol{x}))\big|_{\boldsymbol{x}=\boldsymbol{x}^{(k+1)}} + (\boldsymbol{x}^{(k+1)} - \boldsymbol{z}^{(k)} + \boldsymbol{u}^{(k)}) = \boldsymbol{0} \tag{106}$$

$$\implies \frac{1}{\sqrt{d}}\left\|\boldsymbol{x}^{(k+1)} - \boldsymbol{z}^{(k)} + \boldsymbol{u}^{(k)}\right\|_2 = \frac{1}{\rho_k\sqrt{d}}\left\|\nabla_{\boldsymbol{x}}\ell(\boldsymbol{y}||\mathcal{A}(\boldsymbol{x}))\big|_{\boldsymbol{x}=\boldsymbol{x}^{(k+1)}}\right\|_2 \tag{107}$$

Using the assumption of existence of $R < \infty$ such that $\|\nabla_{\boldsymbol{x}}\ell(\boldsymbol{y}||\mathcal{A}(\boldsymbol{x}))\|_2/\sqrt{d} \leq R$, $\forall \boldsymbol{x} \in \mathcal{X}$, we get

$$\frac{1}{\sqrt{d}}\left\|\boldsymbol{x}^{(k+1)} - \boldsymbol{z}^{(k)} + \boldsymbol{u}^{(k)}\right\|_2 \leq \frac{R}{\rho_k} \tag{108}$$

Since the denoiser $D_{\sigma^{(k)}}$ is bounded with probability at least $1 - 2e^{-\nu_k}$, we have

$$\frac{1}{\sqrt{d}}\|(D_{\sigma^{(k)}} - I)(\boldsymbol{x})\|_2 \leq c_k \tag{109}$$

Now,

$$\frac{1}{\sqrt{d}}\left\|\boldsymbol{x}^{(k+1)} - \boldsymbol{z}^{(k+1)} + \boldsymbol{u}^{(k)}\right\|_2 = \frac{1}{\sqrt{d}}\left\|\boldsymbol{x}^{(k+1)} + \boldsymbol{u}^{(k)} - D_{\sigma^{(k)}}(\boldsymbol{x}^{(k+1)} + \boldsymbol{u}^{(k)})\right\|_2 \tag{110}$$

$$= \frac{1}{\sqrt{d}}\left\|(D_{\sigma^{(k)}} - I)(\boldsymbol{x}^{(k+1)} + \boldsymbol{u}^{(k)})\right\|_2 \tag{111}$$

$$\leq c_k \tag{112}$$

Now, using triangle inequality, we can bound $\left\|\boldsymbol{z}^{(k+1)} - \boldsymbol{z}^{(k)}\right\|_2$ as

$$\frac{1}{\sqrt{d}}\left\|\boldsymbol{z}^{(k+1)} - \boldsymbol{z}^{(k)}\right\|_2 = \frac{1}{\sqrt{d}}\left\|\boldsymbol{z}^{(k+1)} - \boldsymbol{x}^{(k+1)} - \boldsymbol{u}^{(k)} + \boldsymbol{x}^{(k+1)} + \boldsymbol{u}^{(k)} - \boldsymbol{z}^{(k)}\right\|_2 \tag{113}$$

$$\leq \frac{R}{\rho_k} + c_k \tag{114}$$

Similarly, it can be shown that

$$\frac{1}{\sqrt{d}}\left\|\boldsymbol{u}^{(k+1)}\right\|_2 = \frac{1}{\sqrt{d}}\left\|\boldsymbol{u}^{(k)} + \boldsymbol{x}^{(k+1)} - \boldsymbol{z}^{(k+1)}\right\|_2 \tag{115}$$

$$= \frac{1}{\sqrt{d}}\left\|\boldsymbol{u}^{(k)} + \boldsymbol{x}^{(k+1)} - D_{\sigma^{(k)}}(\boldsymbol{x}^{(k+1)} + \boldsymbol{u}^{(k)})\right\|_2 \tag{116}$$

$$= \frac{1}{\sqrt{d}}\left\|(D_{\sigma^{(k)}} - I)(\boldsymbol{x}^{(k+1)} + \boldsymbol{u}^{(k)})\right\|_2 \tag{117}$$

$$= c_k \tag{118}$$

This implies $\frac{1}{\sqrt{d}}\left\|\boldsymbol{u}^{(k+1)} - \boldsymbol{u}^{(k)}\right\|_2 \leq 2c_k$. Finally, we use $\boldsymbol{x}^{(k+1)} = \boldsymbol{u}^{(k+1)} - \boldsymbol{u}^{(k)} + \boldsymbol{z}^{(k+1)}$ to obtain

$$\frac{1}{\sqrt{d}}\left\|\boldsymbol{x}^{(k+1)} - \boldsymbol{x}^{(k)}\right\|_2 \tag{119}$$

$$= \frac{1}{\sqrt{d}}\left\|\boldsymbol{u}^{(k+1)} - \boldsymbol{u}^{(k)} + \boldsymbol{z}^{(k+1)} - \boldsymbol{u}^{(k)} + \boldsymbol{u}^{(k-1)} - \boldsymbol{z}^{(k)}\right\|_2 \tag{120}$$

$$\leq \frac{1}{\sqrt{d}}\left\|\boldsymbol{u}^{(k+1)} - \boldsymbol{u}^{(k)}\right\|_2 + \frac{1}{\sqrt{d}}\left\|\boldsymbol{u}^{(k)} - \boldsymbol{u}^{(k-1)}\right\|_2 + \frac{1}{\sqrt{d}}\left\|\boldsymbol{z}^{(k+1)} - \boldsymbol{z}^{(k)}\right\|_2 \tag{121}$$

$$\leq 2c_k + 2c_{k-1} + \frac{R}{\rho_k} + c_k \tag{122}$$

$$= 3c_k + 2c_{k-1} + \frac{R}{\rho_k} \tag{123}$$

Combining all the bounds results using triangle inequality results in

$$\beta_{k+1} \leq 6c_k + 2c_{k-1} + \frac{2R}{\rho_k} \tag{124}$$

where, $c_k = (\sigma^{(k)})^2(2 + 4\sqrt{\nu_k} + 4\nu_k) + {16\sigma_{\boldsymbol{s}(k)}^2}/{1 - M\sigma_{\boldsymbol{s}(k)}^2} \log {2}/{\nu_k} + 2\sigma_{\boldsymbol{s}(k)}^4 L^2 + 2(\sigma^{(k)})^4 L^2$ □

**Remark 1.** *While the proposed method and its theoretical results are based on* Variance Exploding *(VE) scheduling, they can be easily extended to* Variance Preserving *(VP) scheduling case (Karras et al., 2022).*

### E.2 THEORETICAL RESULTS WITH FINITE DC STEPS $J$

**Lemma 19.** *Let $H_{\mathrm{dc}}^{(k)} : \boldsymbol{z}_{\mathrm{ac}}^{(k)} \mapsto \boldsymbol{z}_{\mathrm{dc}}^{(k)}$ denote the function corresponding to fine correction as defined in Algorithm 1 with finite J and $\eta^{(k)} \leq 2\sigma_{\boldsymbol{s}(k)}^2$. Also, let $\pi^{(k)} = p(\boldsymbol{z}_{\sigma^{(k)}}|\boldsymbol{z}_{\mathrm{ac}}^{(k)})$ be the stationary target distribution and $\tilde{\pi}_0^{(k)}$ be initial distribution used for the DC at iteration $k$. Then, with probability at least $1 - e^{-\nu_k}$, the following holds for any $\boldsymbol{x}, \boldsymbol{y} \in \mathcal{X}$ if ${1}/{\sigma_{\boldsymbol{s}(k)}^2} < M_t$:*

$$\left\|(H_{\mathrm{dc}}^{(k)} - I)(\boldsymbol{x}) - (H_{\mathrm{dc}}^{(k)} - I)(\boldsymbol{y})\right\|_2^2 \leq (\delta_{\mathrm{dc}}^{(k)})^2 + (\epsilon_{\mathrm{dc}}^{(k)})^2\|\boldsymbol{x} - \boldsymbol{y}\|_2^2 \tag{125}$$

*where, $0 < \kappa < 1$, $C > 0$, $(\delta_{\mathrm{dc}}^{(k)})^2 = \frac{64d\sigma_{\boldsymbol{s}(k)}^2}{(1 - M_t\sigma_{\boldsymbol{s}(k)}^2)} \log \frac{2}{\nu_k} + C(1 - \kappa)^{2J}\mathcal{W}_2^2(\tilde{\pi}_0^{(k)}, \pi^{(k)}) + O\left((\eta^{(k)})^2\right)$, and $(\epsilon_{\mathrm{dc}}^{(k)})^2 = \left(\frac{2\sqrt{2}M_t\sigma_{\boldsymbol{s}(k)}^2}{1 - M_t\sigma_{\boldsymbol{s}(k)}^2}\right)^2$.*

*Proof.* With finite $J$, the langevin dynamics doesn't necessarily converge to the stationary distribution $\pi^{(k)} = p(\boldsymbol{z}_{\sigma^{(k)}}|\boldsymbol{z}_{\text{ac}}^{(k)})$. Let $\tilde{\pi}_0^{(k)}$ be the initial distribution used to initialize the finite step langevin dynamics and $\tilde{\pi}^{(k)}$ be the distribution of the iterate after running finite $J$ steps of langevin dynamics. Using (Dalalyan & Karagulyan, 2019)[Theorem 1], the following holds for $\eta^{(k)} \leq 2\sigma_{\boldsymbol{s}^{(k)}}^2$

$$\mathcal{W}_2(\tilde{\pi}^{(k)}, \pi^{(k)}) \leq (1-\kappa)^J \mathcal{W}_2(\tilde{\pi}_0^{(k)}, \pi^{(k)}) + O\left(\eta^{(k)}\right) \tag{126}$$

where, $0 < \kappa < 1$ and $\mathcal{W}_2$ is the 2-Wasserstein distance.

Using Kantorovich and Rubinstein dual representation (Villani et al., 2008),

$$\|\mathbb{E}_{\tilde{\pi}^{(k)}}[\boldsymbol{z}] - \mathbb{E}_{\pi^{(k)}}[\boldsymbol{z}]\|_2 = \left\|\int \boldsymbol{z}_{\text{dc}}^{(k)} d\tilde{\pi}^{(k)}(\boldsymbol{z}) - \int \boldsymbol{z} d\pi^{(k)}(\boldsymbol{z})\right\|_2 \tag{127}$$

$$\leq \mathcal{W}_1(\tilde{\pi}^{(k)}, \pi^{(k)}) \tag{128}$$

$$\leq \mathcal{W}_2(\tilde{\pi}^{(k)}, \pi^{(k)}) \tag{129}$$

The last inequality is due to the Holder's inequality Villani et al. (2008). Using triangle inequality leads to following:

$$\left\|\boldsymbol{z}_{\text{dc}}^{(k)} - \mathbb{E}_{\pi^{(k)}}[\boldsymbol{z}]\right\|_2 \leq \left\|\boldsymbol{z}_{\text{dc}}^{(k)} - \mathbb{E}_{\tilde{\pi}^{(k)}}[\boldsymbol{z}]\right\|_2 + \|\mathbb{E}_{\tilde{\pi}^{(k)}}[\boldsymbol{z}] - \mathbb{E}_{\pi^{(k)}}[\boldsymbol{z}]\|_2 \tag{130}$$

$$\leq \left\|\boldsymbol{z}_{\text{dc}}^{(k)} - \mathbb{E}_{\tilde{\pi}^{(k)}}[\boldsymbol{z}]\right\|_2 + (1-\kappa)^J \mathcal{W}_2(\tilde{\pi}_0^{(k)}, \pi^{(k)}) + O\left(\eta^{(k)}\right) \tag{131}$$

Using this result and following the same procedure as in Lemma 11, we get the theorem.

$\square$

**Theorem 4.** *Suppose that the assumptions in Theorem 1, Assumption 2, and Assumption 3 hold. Further, assume that the DC steps finite steps $J$ and $\eta^{(k)} \leq 2\sigma_{\boldsymbol{s}^{(k)}}^2$. Also, let $\pi^{(k)} = p(\boldsymbol{z}_{\sigma^{(k)}}|\boldsymbol{z}_{\text{ac}}^{(k)})$ be the stationary target distribution and $\tilde{\pi}_0^{(k)}$ be initial distribution used for the DC at iteration $k$. Let $D_{\sigma^{(k)}} : \widetilde{\boldsymbol{z}}^{(k)} \mapsto \boldsymbol{z}_{\text{tw}}^{(k)}$ denote the AC-DC denoiser. Then, we have:*

*With probability at least $1 - 2e^{-\nu_k}$, the following holds for iteration $k$ of ADMM-PnP:*

$$\|(D_{\sigma^{(k)}} - I)(\boldsymbol{x}) - (D_{\sigma^{(k)}} - I)(\boldsymbol{y})\|_2^2 \leq \epsilon_k^2 \|\boldsymbol{x} - \boldsymbol{y}\|_2^2 + \delta_k^2 \tag{132}$$

*for any $\boldsymbol{x}, \boldsymbol{y} \in \mathcal{X}$, $k \in \mathbb{N}^+$, a constant $0 < \kappa < 1$ and $C > 0$, when $\sigma_{\boldsymbol{s}^{(k)}}^2 + (\sigma^{(k)})^2 < 1/M$ with*

$$\epsilon_k^2 = 3((2\sqrt{2}M\sigma_{\boldsymbol{s}^{(k)}}^2/(1-M\sigma_{\boldsymbol{s}^{(k)}}^2))^2 + (\sigma^{(k)})^4 M^2) \tag{133}$$

$$\delta_k^2 = 3(2(\sigma^{(k)})^2(d + 2\sqrt{d\nu_k} + 2\nu_k) + 64d\sigma_{\boldsymbol{s}^{(k)}}^2/(1-M\sigma_{\boldsymbol{s}^{(k)}}^2)\log 2/\nu_k +$$
$$C(1-\kappa)^{2J}\mathcal{W}_2^2(\tilde{\pi}_0^{(k)}, \pi^{(k)}) + O\left((\eta^{(k)})^2\right). \tag{134}$$

*In other words, with $\nu_k = \ln 2\pi/6\eta + 2\ln k$, the denoiser $D_{\sigma^{(k)}}$ satisfies part (a) for all $k \in \mathbb{N}^+$ with probability at least $1 - \eta$.*

*Proof.* By substituting Lemma 11 with Lemma 19 leads to the theorem. $\square$

**Theorem 5.** *Suppose that Assumptions 2-3 hold. Let $D := \text{diam}(\mathcal{X}) = \sup_{\boldsymbol{x},\boldsymbol{y} \in \mathcal{X}} \|\boldsymbol{x} - \boldsymbol{y}\|_2 < \infty$, $S := \inf_{\boldsymbol{x} \in \mathcal{X}} \|\nabla \log p_{\text{data}}(\boldsymbol{x})\|_2 < \infty$ and define $L := MD + S$. Let $D_{\sigma^{(k)}} : \widetilde{\boldsymbol{z}}^{(k)} \mapsto \boldsymbol{z}_{\text{tw}}^{(k)}$ denote the AC-DC denoiser. Further, assume that the DC steps finite steps $J$ and $\eta^{(k)} \leq 2\sigma_{\boldsymbol{s}^{(k)}}^2$. Also, let $\pi^{(k)} = p(\boldsymbol{z}_{\sigma^{(k)}}|\boldsymbol{z}_{\text{ac}}^{(k)})$ be the stationary target distribution and $\tilde{\pi}_0^{(k)}$ be initial distribution used for the DC at iteration $k$. Then, the following hold:*

*(**Boundedness**) With probability at least $1 - 2e^{-\nu_k}$, the denoiser $D_{\sigma^{(k)}}$ is bounded at each iteration $k$ i.e. $\frac{1}{d}\|(D_{\sigma^{(k)}} - I)(\boldsymbol{x})\|_2^2 \leq c_k^2$ whenever $\sigma_{\boldsymbol{s}^{(k)}}^2 + (\sigma^{(k)})^2 < 1/M$, where $c_k = (\sigma^{(k)})^2(2 + 4\sqrt{\nu_k} + 4\nu_k) + 32\sigma_{\boldsymbol{s}^{(k)}}^2/(1-M\sigma_{\boldsymbol{s}^{(k)}}^2)\log 2/\nu_k + C(1-\kappa)^{2J}\mathcal{W}_2^2(\tilde{\pi}_0^{(k)}, \pi^{(k)}) + O\left((\eta^{(k)})^2\right) + 4L^2\sigma_{\boldsymbol{s}^{(k)}}^4 + 2(\sigma^{(k)})^4 L^2$, $0 < \kappa < 1$, $C > 0$ and $\nu_k > 0$.*

*Let $\nu_k = \ln \frac{2\pi^2}{6\eta} + 2\ln k$ with $\eta \in (0, 1]$. Consequently, the denoiser $D_{\sigma^{(k)}}$ is bounded for all $k \in \mathbb{N}_+$ with corresponding $c_k$ and probability at least $1 - \eta$.*

*Proof.* The proof follows similar as in Theorem 3 by incorporating the effect of finite $J$ in Lemma 16 as done in Lemma 19. $\qquad\square$

# F  THEORETICAL RESULTS FOR ODE BASED DENOISER

Refer to the Zhang et al. (2024) for details on ODE based denoiser.

## F.1  THEORETICAL RESULTS EQUIVALENT TO THEOREM 2

**Lemma 20.** *Let $H_{\mathrm{ode}}^{(k)} : \boldsymbol{z}_{\mathrm{dc}}^{(k)} \mapsto \boldsymbol{z}_{\mathrm{ode}}^{(k)}$ denote the projection function using ode based denoiser (Karras et al., 2022) in Algorithm 1. Then, we have the following*

$$\left\| (H_{\mathrm{ode}}^{(k)} - I)(\boldsymbol{x}) - (H_{\mathrm{ode}}^{(k)} - I)(\boldsymbol{y}) \right\|_2^2 \leq (\epsilon_{\mathrm{ode}}^{(k)})^2 \|\boldsymbol{x} - \boldsymbol{y}\|_2^2 + \delta_{\mathrm{ode}}^{(k)\,2} \tag{135}$$

*for any $\boldsymbol{x}, \boldsymbol{y} \in \mathcal{X}$ with $(\epsilon_{\mathrm{ode}}^{(k)})^2 = 2\left( \int_{t=t_{\sigma^{(k)}}}^{0} (\sigma(t)\sigma'(t)M_t)^2\, dt \right)$, and $\delta_{\mathrm{ode}}^{(k)\,2} = 0$.*

*Proof.* Then, the difference of residual of ode projection i.e. $R_{\mathrm{ode}}^{(k)} = H_{\mathrm{ode}}^{(k)} - I$ can be bounded as

$$\begin{aligned}
\left\| R_{\mathrm{ode}}^{(k)}(\boldsymbol{x}) - R_{\mathrm{ode}}^{(k)}(\boldsymbol{y}) \right\|_2^2 &= \left\| \int_{t=t_{\sigma^{(k)}}}^{0} -\sigma(t)\sigma'(t)(\nabla \log p_t(\boldsymbol{x}) - \nabla \log p_t(\boldsymbol{y}))dt \right\|_2^2 \\
&\leq 2 \left\| \int_{t=t_{\sigma^{(k)}}}^{0} -\sigma(t)\sigma'(t)(\nabla \log p_t(\boldsymbol{x}) - \nabla \log p_t(\boldsymbol{y}))dt \right\|_2^2 \\
&\leq 2 \int_{t=t_{\sigma^{(k)}}}^{0} (\sigma(t)\sigma'(t))^2 \| (\nabla \log p_t(\boldsymbol{x}) - \nabla \log p_t(\boldsymbol{y})) \|_2^2 dt \\
&\leq 2 \int_{t=t_{\sigma^{(k)}}}^{0} (\sigma(t)\sigma'(t))^2 M_t^2 \|\boldsymbol{x} - \boldsymbol{y}\|_2^2 dt \\
&\leq 2 \left( \int_{t=t_{\sigma^{(k)}}}^{0} (\sigma(t)\sigma'(t))^2 M_t^2 dt \right) \|\boldsymbol{x} - \boldsymbol{y}\|_2^2
\end{aligned} \tag{136}$$

$\qquad\square$

**Theorem 6.** *Suppose that the assumptions in Theorem 1, Assumption 2 and Assumption 3 hold. Further, assume that the step size satisfies $\eta^{(k)} \to 0$ and the number of iterations $J \to \infty$. Let $D_{\sigma^{(k)}} : \widetilde{\boldsymbol{z}}^{(k)} \mapsto \boldsymbol{z}_{\mathrm{tw}}^{(k)}$ denote the AC-DC denoiser. Then, we have:*

*(a) With probability at least $1 - 2e^{-\nu_k}$, the following holds for iteration $k$ of ADMM-PnP:*

$$\| (D_{\sigma^{(k)}} - I)(\boldsymbol{x}) - (D_{\sigma^{(k)}} - I)(\boldsymbol{y}) \|_2^2 \leq \epsilon_k^2 \|\boldsymbol{x} - \boldsymbol{y}\|_2^2 + \delta_k^2 \tag{137}$$

*for any $\boldsymbol{x}, \boldsymbol{y} \in \mathcal{X}$ and $k \in \mathbb{N}^+$ when $\sigma_{\boldsymbol{s}^{(k)}}^2 + (\sigma^{(k)})^2 < 1/M$ with*

$$\epsilon_k^2 = 3(\sqrt{2}M\sigma_{\boldsymbol{s}^{(k)}}^2 / 1 - \sigma_{\boldsymbol{s}^{(k)}}^2 M)^2 + 6 \int_{t=t_{\sigma^{(k)}}}^{0} (\sigma(t)\sigma'(t))^2 M_t^2 dt) \tag{138}$$

$$\delta_k^2 = 3(2(\sigma^{(k)})^2(d + 2\sqrt{d\nu_k} + 2\nu_k) + 32d\sigma_{\boldsymbol{s}^{(k)}}^2 / (1 - M\sigma_{\boldsymbol{s}^{(k)}}^2) \log 2/\nu_k). \tag{139}$$

*In other words, if $\nu_k = \ln 2\pi/6\eta + 2nk$, the denoiser $D_{\sigma^{(k)}}$ satisfies part (a) for all $k \in \mathbb{N}^+$ with probability at least $1 - \eta$.*

*(b) Assume that $\sigma^{(k)}$ is scheduled such that $\lim_{k\to\infty}(\sigma^{(k)})^2\nu_k = 0$ for $\nu_k = \ln 2\pi/6\eta + 2nk$, $\epsilon < 1$, and $\epsilon/\mu(1+\epsilon-2\epsilon^2) < 1/\rho$ all hold, where $\epsilon = \lim_{k\to\infty} \sup \epsilon_k$ with $\epsilon_k$ defined in (138). Consequently, $\delta = \lim_{k\to\infty} \sup \delta_k$ is finite and ADMM-PnP with the AC-DC denoiser with ode based denoiser converges to an $r$-ball (see $r$ in Theorem 1) with probability at least $1 - \eta$.*

*Proof.* The proof follow similar to the proof of Theorem 2 in Appendix D.1 with the residual bound of Tweedie's lemma replaced by Lemma 20. $\qquad\square$

Table 3: Hyperparameter settings for each task

| Task | $\rho$ | $W$ | lr of Adam in (7a) |
|------|------|------|------|
| Superresolution ($4\times$) | 100 | 100 | $3 \times 10^{-2}$ |
| Gaussian Deblur | 100 | 100 | $5 \times 10^{-2}$ |
| HDR | 500 | 100 | $3 \times 10^{-2}$ |
| Inpainting (Random) | 500 | 100 | $1 \times 10^{-1}$ |
| Inpainting (Box) | 500 | 100 | $1 \times 10^{-1}$ |
| Motion Deblur | 100 | 100 | $1 \times 10^{-1}$ |
| Nonlinear Deblur | 300 | 400 | $3 \times 10^{-1}$ |
| Phase Retrieval | 100 | 400 | $1 \times 10^{-1}$ |

## F.2 THEORETICAL RESULTS EQUIVALENT TO THEOREM 3

**Lemma 21.** *Let $H_{\text{ode}}^{(k)} : \boldsymbol{z}_{\text{dc}}^{(k)} \mapsto \boldsymbol{z}_{\text{ode}}^{(k)}$ denote the projection function using ode based denoiser (Karras et al., 2022) in Algorithm 1. Assume $\|\nabla \log p_{\text{data}}(\boldsymbol{x})\|_\infty \leq L, \ \forall \boldsymbol{x} \in \mathcal{X}$. Then, we have the following*

$$\frac{1}{d}\left\|(H_{\text{ode}}^{(k)} - I)(\boldsymbol{x})\right\|_2^2 \leq L^2 \int_{t=t_{\sigma(k)}}^0 (\sigma(t)\sigma'(t))^2 dt \tag{140}$$

*for any $\boldsymbol{x} \in \mathcal{X}$.*

*Proof.* Then, the residual of ode projection i.e. $R_{\text{ode}}^{(k)} = H_{\text{ode}}^{(k)} - I$ can be bounded as

$$
\begin{aligned}
\left\|R_{\text{ode}}^{(k)}(\boldsymbol{x})\right\|_2^2 &= \left\|\int_{t=t_{\sigma(k)}}^0 -\sigma(t)\sigma'(t)\nabla \log p_t(\boldsymbol{x}) dt\right\|_2^2 \\
&\leq d \cdot L^2 \left\|\int_{t=t_{\sigma(k)}}^0 -\sigma(t)\sigma'(t) dt\right\|_2^2 \\
&\leq d \cdot L^2 \left\|\int_{t=t_{\sigma(k)}}^0 -\sigma(t)\sigma'(t) dt\right\|_2^2 \\
&\leq d \cdot L^2 \int_{t=t_{\sigma(k)}}^0 (\sigma(t)\sigma'(t))^2 dt
\end{aligned}
\tag{141}
$$

$\square$

A theorem analogous to Theorem 3 can also be obtained for ODE-based denoiser. The only difference lies in the expression of the constant $c_k$, which in this case becomes

$$c_k = (\sigma^{(k)})^2(2 + 4\sqrt{\nu_k} + 4\nu_k) + {}^{16\sigma_{\boldsymbol{s}(k)}^2}/_{1 - M\sigma_{\boldsymbol{s}(k)}^2} \log {}^2/_{\nu_k} + 2\sigma_{\boldsymbol{s}(k)}^4 L^2 + 2L^2 \int_{t=t_{\sigma(k)}}^0 (\sigma(t)\sigma'(t))^2 dt.$$

## G  USAGE OF LARGE LANGUAGE MODELS (LLM)

An LLM was used solely to assist with polishing the writing. LLM played no part in the experiments, results and conclusion.

## H    EXPERIMENTAL DETAILS

### H.1    DETAILS ON TASK SPECIFIC DATA-FIDELITY LOSS $\ell$

We use *mean square error* (MSE) as the data-fidelity loss for every task i.e.

$$\ell(\boldsymbol{y}\|\boldsymbol{x}) = -\log p(\boldsymbol{y}|\boldsymbol{x}) = \frac{1}{2\sigma_n^2}\|\boldsymbol{y} - \mathcal{A}(\boldsymbol{x})\|_2^2 \tag{142}$$

### H.2    DETAILS ON PRETRAINED DIFFUSION MODELS

The pretrained models provided in Chung et al. (2023) are used in our experiment. Refer to Chung et al. (2023) for more details on these pretrained models.

### H.3    BASELINE DETAILS

Unless mentioned otherwise, we conduct the experiments in the default settings of their original implementation except for maintaining consistency within the measurement operators.

- **DDRM (Kawar et al., 2021):**  We use 20 steps DDIM with $\eta = 0.85$ and $\eta_b = 1$ as specified in Kawar et al. (2022).
- **DPS (Chung et al., 2023) :** The original implementation is ran in their default settings.
- **DiffPIR (Zhu et al., 2023):** The default settings are adopted in the experiments.
- **RED-diff (Mardani et al., 2024):** We use $\lambda = 0.25$ and $lr = 0.5$ as specified in the paper.
- **DAPS (Zhang et al., 2024):** We use the best performing DAPS-4K version as proposed in the paper.
- **DPIR (Zhang et al., 2022)**: We employ "drunet_color" as PnP denoiser, while keeping all the other settings at their default values.
- **DCDP (Li et al., 2025)**: All the settings are set to their default values.
- **PMC (Sun et al., 2024)**: PMC was proposed using different score models for two different tasks with relatively high measurement SNR. For a fair comparison, we used our own implementation with the same score model checkpoints as our methods, and further tuned this method accordingly.

### H.4    EVALUATION METRICS

For all the methods, we use the implementation of PSNR, SSIM, and LPIPS provided in *piq* python package. The default settings for these metrics are used except the average pooling enabled for LPIPS.

### H.5    COMPUTATION RESOURCE DETAILS

All the experiments were run on a instance equipped with one Nvidia H100 GPU, 20 cores of 2.0 Ghz Intel Xeon Platinum 8480CL CPU, and 64 GB of RAM.

## I    ADDITIONAL EXPERIMENTAL RESULTS

### I.1    ILLUSTRATION OF PROPOSED DENOISER

Figure 6 illustrates the effect of the proposed correction-denoising procedure. The noisy input image $\widetilde{z}^{(k)}$ typically lies far away from the Gaussian noise manifold, leading to poor denoising performance if directly used. To address this mismatch, our method first performs correction to effectively gaussianize the noise which is then denoised using Tweedie's lemma or ode-style score integration, producing a high-quality clean reconstruction.

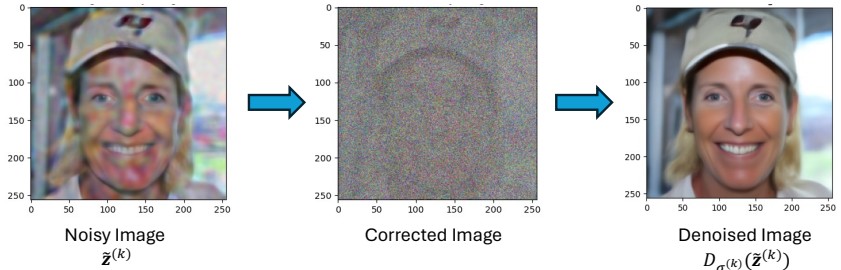

Figure 6: Illustration of correction and denoising step in proposed method.

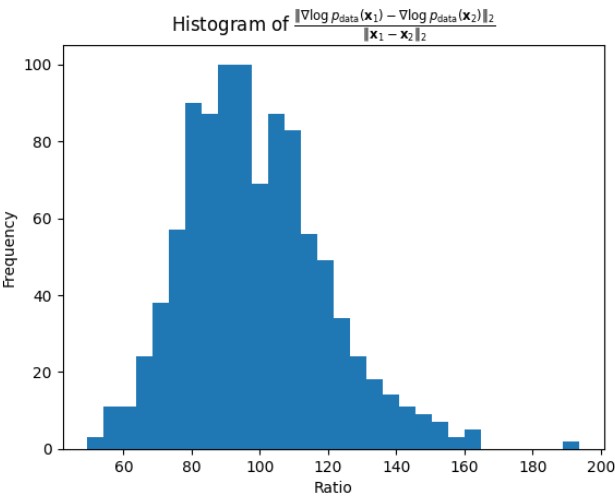

Figure 7: Histogram of score difference norm ratio: $\|\nabla \log p_{\text{data}}(\boldsymbol{x}_2) - \nabla \log p_{\text{data}}(\boldsymbol{x}_1)\|_2 / \|\boldsymbol{x}_1 - \boldsymbol{x}_2\|_2$, illustrating empirical smoothness (Assumption 2).

## I.2 EMPIRICAL VALIDATION OF ASSUMPTION 2 AND ASSUMPTION 3

To assess the practicality of the smoothness and coercivity assumptions used in Theorem 2, we conduct two diagnostic experiments using a pretrained score model on the validation split of the FFHQ dataset. These experiments are designed to evaluate (i) the empirical Lipschitz behavior of the score function $\nabla \log p_{\text{data}}(\boldsymbol{x})$ (Assumption 2), and (ii) the coercivity of the energy landscape $-\log p_{\text{data}}(\boldsymbol{x})$ (Assumption 3).

**Empirical smoothness of the score.** We randomly 1000 samples of $\boldsymbol{x}_1, \boldsymbol{x}_2$ and compute score differences $\|\nabla \log p_{\text{data}}(\boldsymbol{x}_1) - \nabla \log p_{\text{data}}(\boldsymbol{x}_2)\|_2$ and image differences $\|\boldsymbol{x}_1 - \boldsymbol{x}_2\|_2$. Figure 7 plots the histogram of their ratio. The distribution concentrates around a finite value (mostly between 50 and 160), indicating that the score behaves approximately $M$-Lipschitz with a moderate empirical constant. This supports the smoothness requirement in Assumption 2.

**Empirical coercivity.** To evaluate coercivity, we scale images by factors $c \in \{1, 1.5, 2, 3\}$ and measure the quantity $\langle \boldsymbol{x}, -\nabla \log p_{\text{data}}(\boldsymbol{x}) \rangle$ as a function of the squared image norm $\|\boldsymbol{x}\|_2^2$. As shown in Figure 8, the inner product grows approximately linearly with $\|\boldsymbol{x}\|_2^2$, indicating that the learned score consistently pulls large-norm images back toward the data manifold. This behavior is consistent with the coercivity structure assumed in Assumption 3.

Together, these empirical diagnostics demonstrate that the theoretical assumptions employed in our analysis hold approximately in practice and therefore justify the use of DC correction in our AC–DC algorithm.

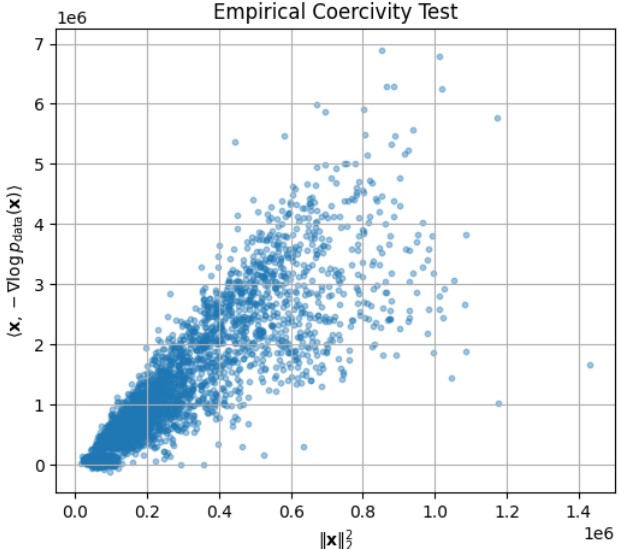

Figure 8: Empirical coercivity test: relationship between $\langle \boldsymbol{x}, -\nabla \log p_{\text{data}}(\boldsymbol{x}) \rangle$ and $\|\boldsymbol{x}\|_2^2$. The strong positive correlation indicates coercive energy behavior (Assumption 3).

Table 4: Reconstruction metrics (100 images) on FFHQ / ImageNet for additional tasks. **Bold**: best, blue: 2nd best.

| Task | Method | FFHQ | | | ImageNet | | |
|------|--------|------|------|------|------|------|------|
| | | PSNR↑ | SSIM↑ | LPIPS↓ | PSNR↑ | SSIM↑ | LPIPS↓ |
| HDR | Ours–tweedie | **27.425** | **0.853** | 0.164 | 26.515 | **0.817** | 0.182 |
| | DAPS | 26.94 | 0.852 | **0.154** | **26.848** | 0.816 | **0.172** |
| | RED-diff | 26.815 | 0.836 | 0.241 | 20.794 | 0.771 | 0.232 |
| | PMC | 21.582 | 0.707 | 0.291 | 22.745 | 0.707 | 0.290 |
| Nonlinear Deblur | Ours–tweedie | **29.326** | **0.823** | 0.185 | **27.837** | 0.725 | 0.212 |
| | DAPS | 28.598 | 0.782 | **0.172** | 27.745 | **0.739** | **0.201** |
| | DPS | 23.746 | 0.668 | 0.276 | 22.724 | 0.543 | 0.394 |
| | RED-diff | 26.9 | 0.72 | 0.234 | 25.488 | 0.72 | 0.207 |
| | PMC | 21.102 | 0.0623 | 0.354 | 22.347 | 0.533 | 0.430 |

## I.3 ILLUSTRATION OF USAGE OF ADDITIONAL REGULARIZATION

To further demonstrate the flexibility of integrating diffusion-based PnP denoisers within the ADMM framework, we present an example where we employ an additional perceptual regularization term which will be handled in the maximum-likelihood (ML) step. In particular, the $\boldsymbol{x}$-update step of ADMM with an LPIPS perceptual regularization (Zhang et al., 2018) becomes:

$$\boldsymbol{x}^{(k+1)} = \arg \min_{\boldsymbol{x}} \frac{1}{\rho} \ell(y \| \mathcal{A}(\boldsymbol{x})) + \frac{1}{2} \|\boldsymbol{x} - \boldsymbol{z}^{(k)} + \boldsymbol{u}^{(k)}\|_2^2 + \lambda_{\text{lpips}} \text{LPIPS}_{\text{VGG}}(\boldsymbol{x}, \boldsymbol{x}_{\text{ref}}), \quad (143)$$

where $\boldsymbol{x}_{\text{ref}}$ is the reference image and $\lambda_{\text{lpips}}$ controls the perceptual strength.

This example highlights the flexibility of the proposed method: unlike traditional diffusion-based PnP approaches that struggle in the presence of dual variables, our design enables seamless incorporation of additional regularization terms. In Fig. 9 we illustrate box inpainting reconstruction task with the perceptual LPIPS-VGG regularization which enhances semantic content consistency while allowing visual style transfer from the reference images.

## I.4 RESULTS ON ADDITIONAL TASKS

The results on additional two tasks: HDR and nonlinear deblurring are presented in the Table 4.

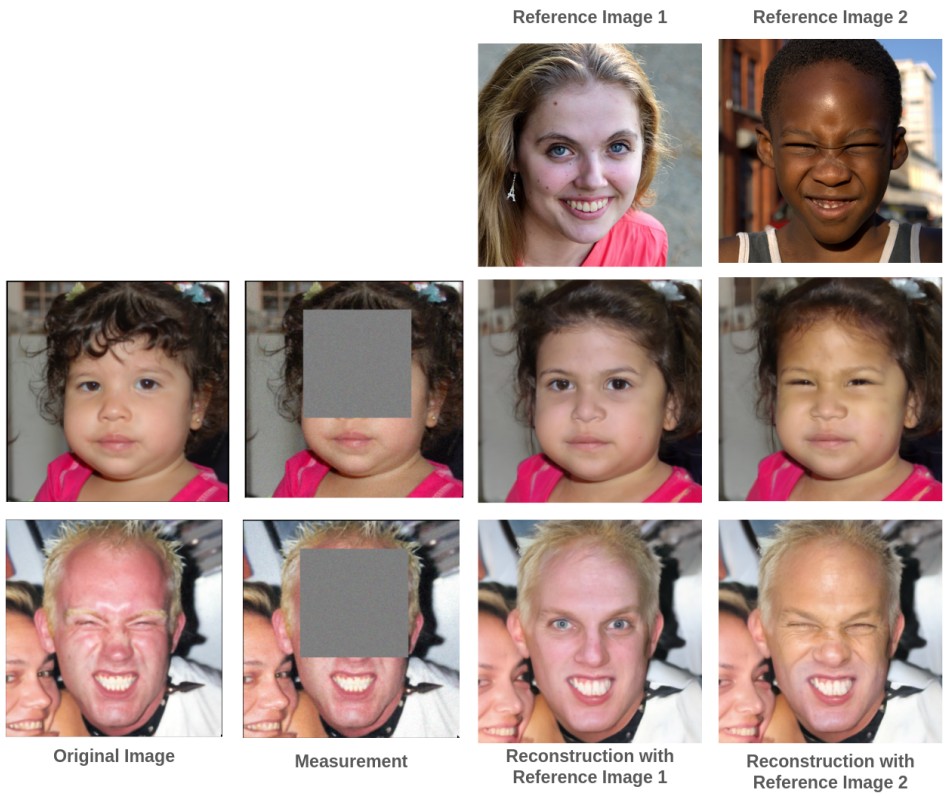

Figure 9: Demonstration of incorporating additional perceptual regularization.

Table 5: Comparison of our method with and without correction on FFHQ. Best results are highlighted in bold.

| | Ours-tweedie without correction | | | Ours-tweedie with correction | | |
|---|---|---|---|---|---|---|
| Tasks | PSNR↑ | SSIM↑ | LPIPS↓ | PSNR↑ | SSIM↑ | LPIPS↓ |
| Superresolution (4x) | 26.915 | 0.730 | 0.314 | **30.439** | **0.857** | **0.178** |
| Gaussian Blur | 28.896 | 0.788 | 0.275 | **30.402** | **0.853** | **0.175** |
| Inpainting (Box) | 15.604 | 0.617 | 0.361 | **24.025** | **0.859** | **0.131** |
| Motion Deblur | 25.123 | 0.538 | 0.370 | **30.003** | **0.854** | **0.179** |
| Nonlinear Deblur | 21.731 | 0.561 | 0.375 | **29.326** | **0.823** | **0.185** |
| Phase Retrieval | 11.978 | 0.181 | 0.726 | **27.944** | **0.793** | **0.209** |

## I.5 ABLATION STUDY

We perform the ablation study on the significance of our proposed correction steps. The results are presented in the Table 5.

## I.6 INFLUENCE OF DECAY SCHEDULE AND NFE EFFICIENCY

In our ADMM-PnP scheme, the size of the decay window for $\sigma^{(k)}$ determines the total number of iterations – and thus the speed of convergence. A shorter window (small $W$) drives $\sigma^{(k)}$ down more quickly, often reaching convergence in fewer steps but at the risk of settling in a suboptimal local minimum.

To study this trade-off, we sweep

$$W \in \{5, 10, 50, 100, 200, 300, 400, 500\}$$

for each task. Since each iteration of Ours-tweedie uses 11 score evaluations (10 for the DC update and 1 for the Tweedie's lemma based denoiser), these W value translate to

$$\text{Number of Function Evaluations (NFE)} = \{55, 110, 550, 1100, 2200, 3300, 4400, 5500\}$$

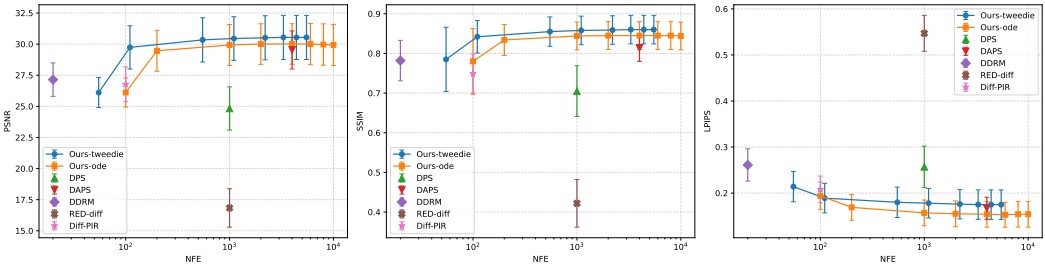

Figure 10: Performance with respect to NFE for Superresolution task (FFHQ)

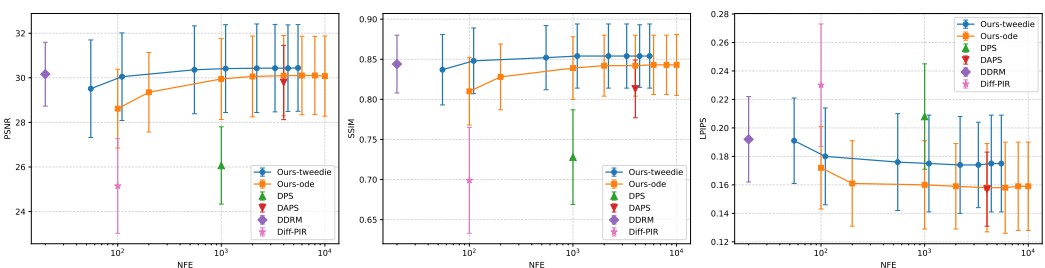

Figure 11: Performance with respect to NFE for Gaussian deblurring task (FFHQ)

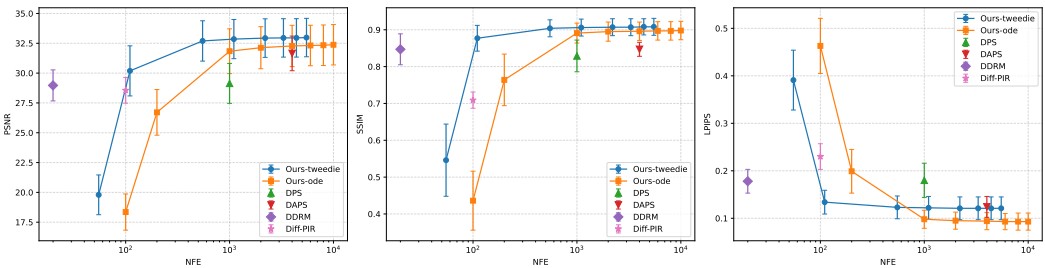

Figure 12: Performance with respect to NFE for Inpainting with random missings (FFHQ)

By contrast, each Ours-ode iteration costs 20 NFEs, giving

Number of Function Evaluations (NFE) $= \{100, 200, 1000, 2000, 4000, 6000, 8000, 10000\}$

Figures 10-15 plot mean±std. (standard deviation) performance of our methods and all baselines against NFE over 100 images of FFHQ dataset. For most tasks, quality saturates after just 10 iterations (110 NFE for Ours-tweedie, 200 NFE for Ours-ode), showing a rapid decay schedule suffices to achieve near-peak results. However, on the hardest inverse problems (phase retrieval and nonlinear blur), gradually decaying noise (larger $W$) and more NFEs yield significantly better reconstructions–far outpacing every baseline. Thus, while aggressive schedules excel on simple tasks, challenging problems benefit from extended iteration and gentler annealing; given enough NFEs, our approach establishes state-of-the-art performance across most of the tasks.

## I.7   MORE QUALITATIVE RESULTS

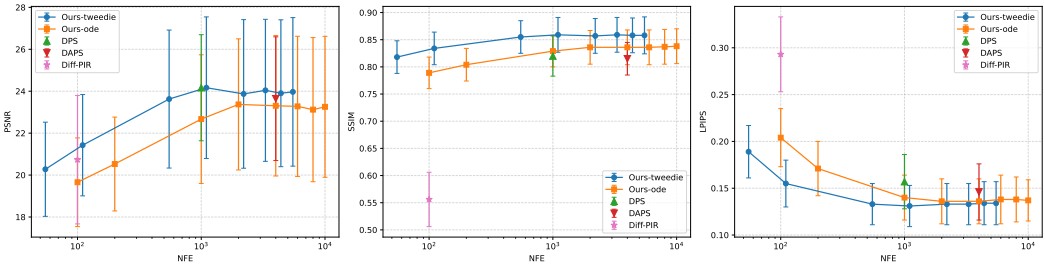

Figure 13: Performance with respect to NFE for Inpainting with box missing (FFHQ)

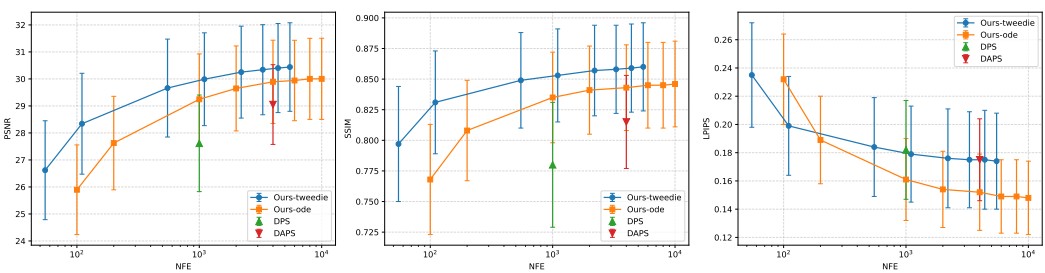

Figure 14: Performance with respect to NFE for Motion blur (FFHQ)

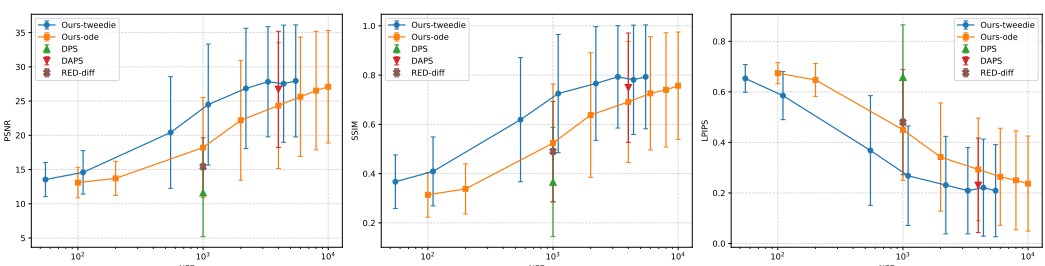

Figure 15: Performance with respect to NFE for Phase retrieval (FFHQ)

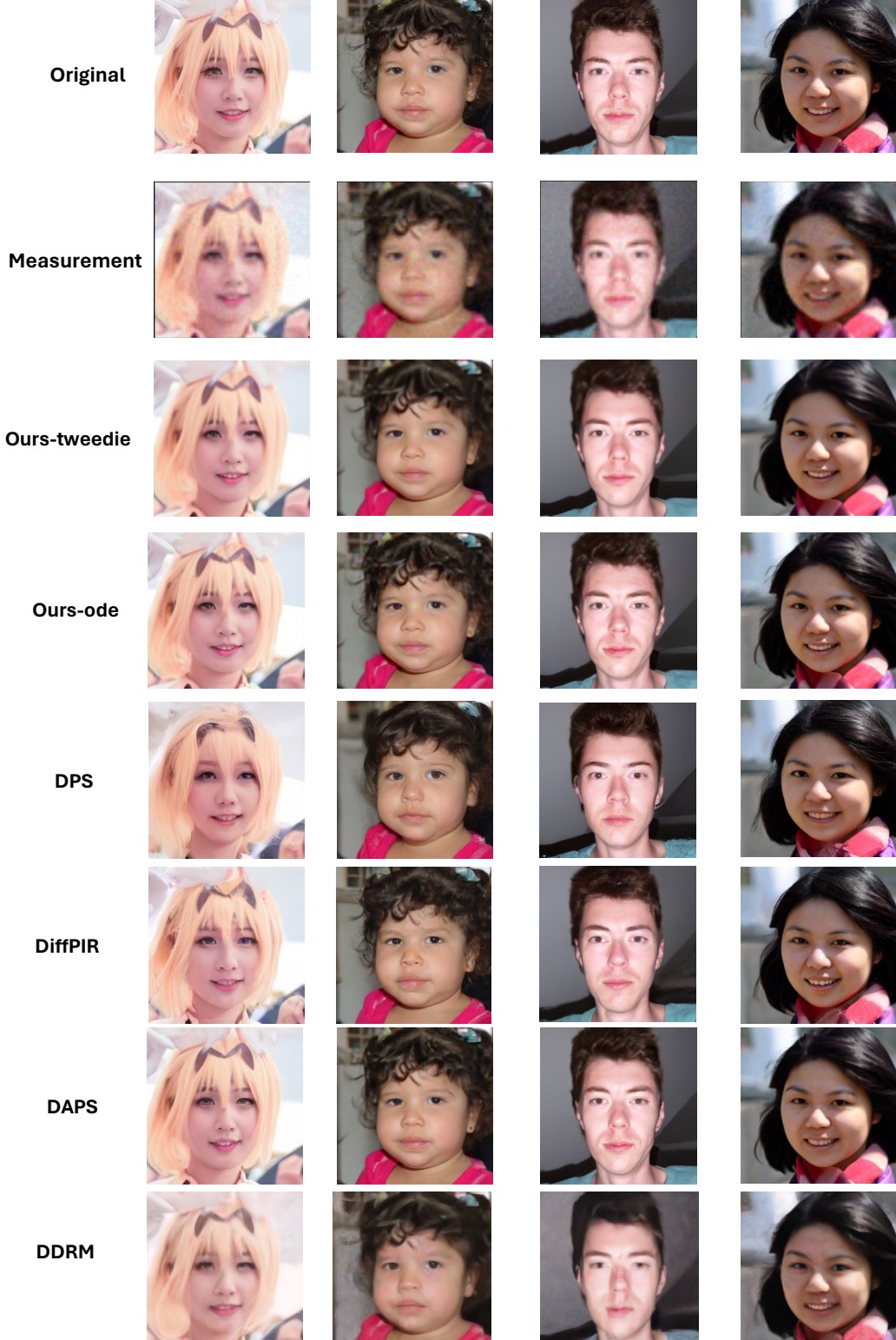

Figure 16: Recovery under $4\times$ superresolution task on FFHQ

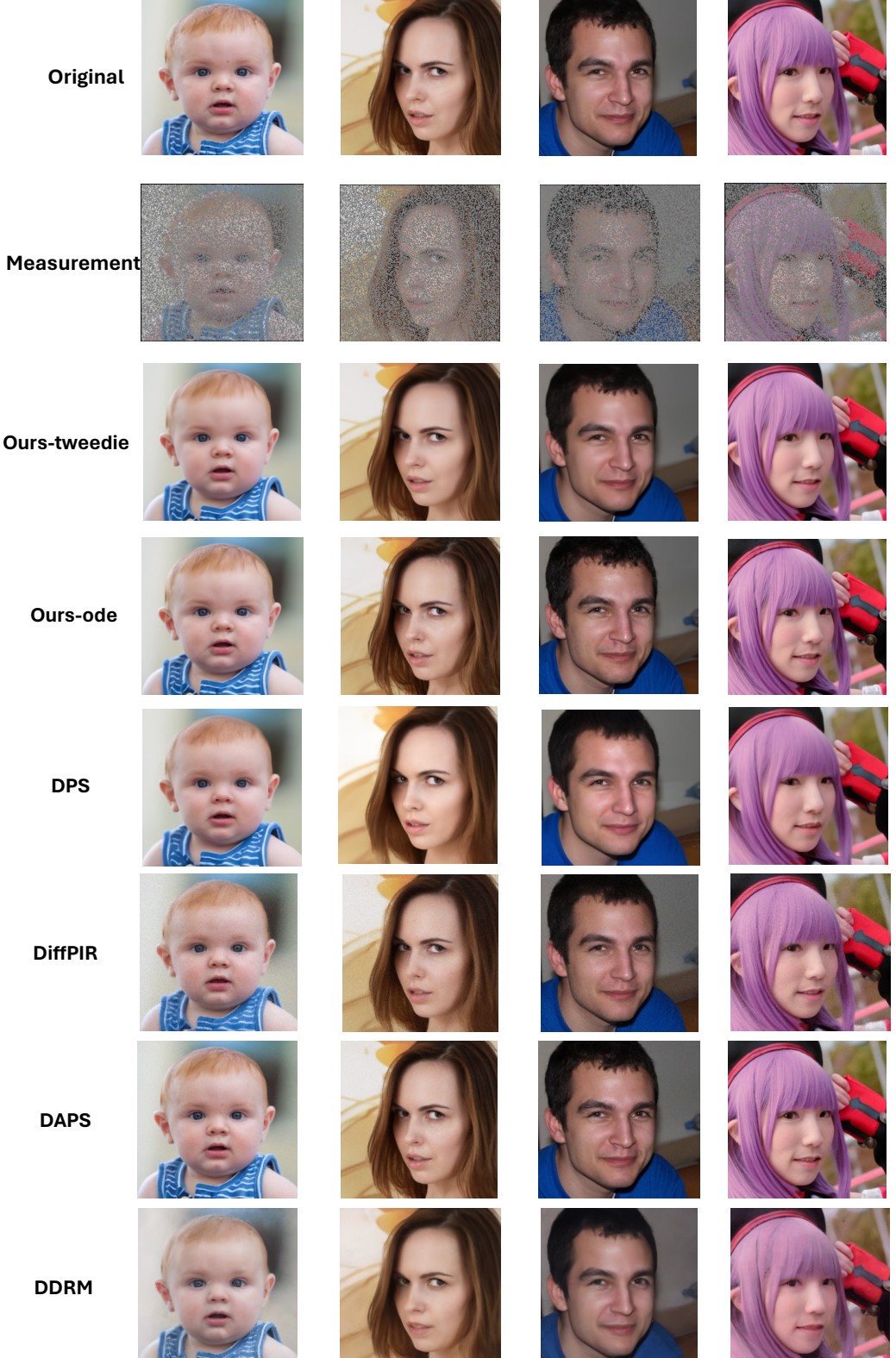

Figure 17: Recovery under inpainting with random missings on FFHQ

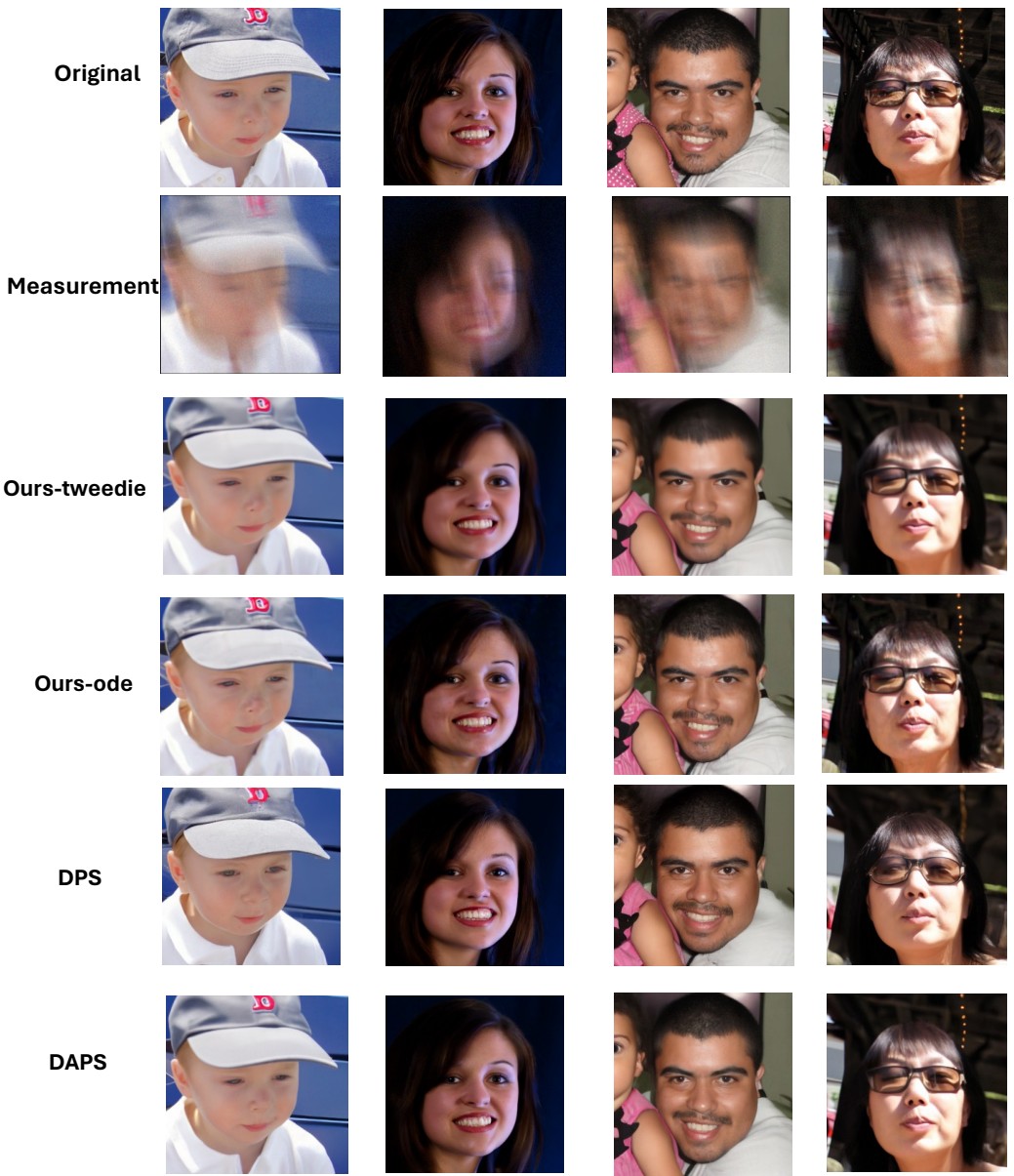

Figure 18: Recovery under motion blur task on FFHQ

