# OpenReview forum: "Taming Score-Based Denoisers in ADMM: A Convergent Plug-and-Play Framework"
_ICLR.cc/2026/Conference — ICLR 2026 Poster_

### Official Review · Reviewer_zbpZ · 2025-10-23

**Soundness:** 3
**Presentation:** 3
**Contribution:** 4
**Rating:** 6
**Confidence:** 4

**Summary:**

The paper introduces a new plug-and-play ADMM framework that integrates score-based denoisers derived from pretrained diffusion models. The contribution lies in two main aspects: (1) addressing the  manifold mismatch between the noise-trained score manifolds and the actual ADMM iterates, particularly due to dual-variable effects and (2) the lack of convergence guarantees when using score-based denoisers within ADMM.

The author propose using a three-stage denoiser called AC–DC (Auto-Correction + Directional Correction + Score Denoising), where in the AC step,  Gaussian noise is added to align iterates with score manifolds, in the DC step  conditional Langevin dynamics is used to refine the direction toward the true data manifold and in the score denoising, pre-trained score model is applied. They provide theoretical results for the cases of convex data-fidelity with weakly contractive denoisers and noncovex case.

Experiments on image inverse problems (super-resolution, inpainting, motion deblurring) show performance gains over methods like DDRM, DiffPIR, and DPS.

**Strengths:**

The combination of ADMM with a diffusion-based denoiser that explicitly accounts for manifold mismatch  is conceptually fresh. Previous works used diffusion models in proximal steps or as score priors, but rarely in a primal–dual ADMM context with noise manifold adjustment.

The authors provide theoretical analysis regarding the convergence of the proposed denoiser.

The experimental results demonstrate that the proposed denoiser is indeed effective in solving inverse problems.

The approach tackles a real gap between diffusion-based inference and optimization-based inverse problem solvers.

**Weaknesses:**

Some of the assumptions are strong. Theorem 2 requires stationarity of the inner Langevin DC step at each iteration and smoothness/coercivity of –log p(x); these are hard to ensure in high-dimensional image spaces. Is there anyway authors could test this on the empirical images or a toy problem?

**Questions:**

Could the author comment on the computation complexity added by using the AC-DC denoiser as opposed to simply apply the score-based denoiser with adjusting to the noise manifold?

---

> ### Author Response · Authors · 2025-11-20
>
> We would like to thank the reviewer for their comments and concerns.
> __(Strong Assumptions).__ Indeed, as the reviewer correctly pointed out, the convergence to the stationarity of Langevin DC steps at each iteration is a strong assumption. In the updated PDF, we have derived the results when DC uses fixed $J$ steps (Appendix E.2 ). The fixed step number $J$ would result in additional term in $\delta_k$ and $c_k$ that depends on three factors: a) the step size b) number of iterations $J$, and c) the Wasserstein distance between initialization and stationary distribution but decays exponentially with $J$.
>
> Regarding the log-pdf assumption, we appreciate the reviewer’s concern about the assumptions. Appendix I.2 now includes empirical evaluations that verify (i) bounded empirical Lipschitz constants of the score and (ii) approximately linear growth of $\langle \mathbf{x},-\nabla \log p(\mathbf{x})$ with $||\mathbf{x}||_2^2$. These diagnostics show that Assumption 2 and Assumption 3 should be reasonable in practice.
>
> __(Computational Complexity).__ Thanks for bringing this up and it is a good point to clarify. AC-DC denoiser does incur more score evaluations in each iteration as compared to  directly using denoising without DC correction. By doing so, what we gain is that the added per-iteration computation improves the performance significantly. Nonetheless, if one views the total number of score evaluations against performance metrics, it appears our method attains much better performance relative to the baselines when using the same number of score evaluations (please see Appendix I.6). Our understanding is that the DC step, despite being a seemingly expensive Langevin process, can help reduce the total number of iterations needed by ADMM to attain good performance.

---

> > ### Comment · Reviewer_zbpZ · 2025-11-26
> >
> > I thank the authors for including Lemma 19. Experiment on the coercivity of score is convincing. I have adjusted my score.

---

> > > ### Author Response · Authors · 2025-11-27
> > >
> > > We would like to thank the reviewer for adjusting their scores. We appreciate the time and effort put into reviewing this work.

---

### Official Review · Reviewer_srVv · 2025-10-29

**Soundness:** 3
**Presentation:** 3
**Contribution:** 2
**Rating:** 4
**Confidence:** 4

**Summary:**

## Summary
This paper proposes an AC-DC denoiser within an ADMM plug-and-play (PnP) framework for inverse problems. The denoiser couples an annealed correction (AC) step with a drift-correction (DC) step inspired by score-based diffusion, and the theory aims to show convergence of ADMM-PnP with this denoiser under high-probability weak nonexpansiveness. Theoretical results include: (i) convergence under strong convexity of the fidelity term via a fixed-point argument; (ii) a high-probability weak nonexpansiveness result under a schedule for σ(k) that decays to zero, leading to convergence with fixed ρ; and (iii) a convergence result without convexity of ℓ by adopting an adaptive-ρ scheme (Chan et al., 2016), under boundedness assumptions on the data domain and the score, and an assumption of bounded gradients of ℓ.

**Strengths:**

## Strengths

1. The paper tackles an important and timely problem: how to safely and provably integrate score-based denoisers within PnP/ADMM. The AC-DC design is intuitive and practically relevant.

2. The high-probability analysis for weak nonexpansiveness is interesting; to my knowledge, there are relatively few works that attempt to rigorously control the stochasticity induced by score-based denoisers within PnP iterations.

3. The boundedness result under adaptive ρ and a vanishing σ(k) schedule is a useful step toward understanding convergence without convexity.

4. The empirical results suggest the method is competitive across a range of inverse problems.

**Weaknesses:**

## Major concerns

Despite the nice contributions, I have the following major concerns.

### 1 AC-DC denoiser

> Algorithm 1 injects noise inside the AC-DC denoiser. As implemented, Dσ is a stochastic operator: given the same input, it can return different outputs due to the injected noise (both in the AC step and in the DC sampling/evolution).
However, the convergence analysis treats Dσ as a deterministic mapping z → Dσ(z), i.e., a point-to-point operator. This is a mismatch. It seems that the current deterministic analysis does not rigorously cover the algorithm being evaluated experimentally.

### 2. Theorem 1

> 2.1 The proof strategy follows the fixed-point iteration approach of Ryu et al. This requires the fidelity ℓ to be µ-strongly convex. This is violated in many applications highlighted by the paper itself: deblurring, super-resolution, compressed sensing, MRI, and inpainting (e.g., rank-deficient A or non-strongly-convex penalties).

> 2.2 The parameter condition involving ε and µ may be restrictive. In particular, as ε → 1, the factor (1 + ε − 2ε^2) → 0, making the right-hand side ε/(µ(1 + ε − 2ε^2)) blow up. Then the condition 1/ρ > ε/(µ(1 + ε − 2ε^2)) is practically impossible to satisfy. This contradicts with the experimental setup, where Table 3 indicates ρ ≥ 100 in all cases, which violates the small-step requirement in the theorem.

### 3. Theorem 2

> The assumption “the DC step reaches the stationary distribution for each k” is strong.

### 4. Theorem 3

> 4.1 The analysis follows Chan et al. (2016): boundedness of the denoiser, vanishing noise schedules, and adaptive ρ yield convergence to a fixed point. While this shows stability, it does not characterize the limit point as a solution to any explicit optimization problem. As a result, the algorithm’s fixed point lacks interpretability: it is not known to minimize a well-defined objective nor to satisfy an equilibrium condition like a monotone inclusion. This is a conceptual limitation of the framework that should be acknowledged and discussed more candidly.

> 4.2 The assumption may be strong: The requirement that ||∇ℓ(x)||/√d ≤ R < ∞ for all x is generally false for common inverse problems (deblurring, super-resolution, compressed sensing, MRI, inpainting) unless x is constrained to a compact set. For instance, with ℓ(x) = (1/2σ2)||Ax − y||2, the gradient norm grows with ||x|| unless constrained.


## Overall assessment
This work addresses an important problem and proposes a practically relevant denoiser within PnP/ADMM, with nontrivial theoretical attempts. However, the current theoretical results rely on assumptions that do not match the algorithm as implemented (notably, treating a stochastic denoiser as deterministic and assuming stationarity per iteration), and on conditions that are violated in key applications and in the reported experiments (strong convexity of ℓ; parameter inequalities incompatible with large ρ; bounded gradient assumptions). I believe the paper would be significantly strengthened by a proper stochastic operator analysis and by reconciling the parameter/schedule assumptions with the experimental setup.

If the authors address some of the issues mentioned above, I would consider to increase my score.

**Questions:**

Please see the weaknesses part.

---

> ### Author Response · Authors · 2025-11-20
>
> We would like to thank the reviewer for their comments and questions.
>
>  __(Randomness in AC-DC denoiser).__ Indeed, the proposed AC-DC denoiser is a stochastic operator that injects noise. However, we would like to point out that our convergence analysis does account for this stochasticity of the denoiser. Note that both Theorem 2 and Theorem 3 establishes the convergence results for the proposed AC-DC denoiser with high probability $1-\eta$ due to this stochasticity.  To be more specific, we argued that in each iteration some conditions hold with a high probability $1-\eta_k$, and such probability decreases when $k$ grows larger (so that $\sum_k \eta_k$ converges to $\eta$). Hence, when considering all iterations, and by the union bound, we end up with an overall $1- \eta$ probability under which convergence holds.
>
>
> __(Theorem 1).__ Indeed, Theorem 1 is meant to cover the strongly convex case. Theorem 3 covers the nonconvex case.
>
> In terms of the parameters $\rho$ and $\epsilon$, we would like to clarify that the parameter $\epsilon$ is the contractive value of the residual of the denoiser ( see Assumption 1). This value depends on the denoiser being used, and the value $\epsilon\to1$ means that the residual of the denoiser doesn't contract irrespective of $\rho$ value. The denoiser with $\epsilon\ge 1$ can be considered as non-contractive and hence, the method with such denoiser won't converge. That is, the conditions are not contradicting each other. This result is consistent with the result of [Ryu et al., 2019]. Note that this $\mu$ value for our denoiser depends on the limiting value of $\sigma^{(k)}$ which can be ensured to be less than 1 with proper scheduling of $\sigma^{(k)}$ (see Theorem 2b ).
>
> __(Assumptions in Theorem 2).__ The reviewer has made a fair point. We initially used this “stationarity” assumption to simplify notations here. But in retrospect, this introduces a fairly strong assumption that we in fact do not need. In the updated PDF, we have derived convergence results corresponding to Theorem 2 and Theorem 3 that use a fixed and finite $J$ Langevin steps in DC in Appendix E.2. The fixed step number $J$ would result in additional term in $\delta_k$ and $c_k$ that depends on three factors: a) the step size b) number of iterations $J$, and c) the Wasserstein distance between initialization and stationary distribution but decays exponentially with $J$.
>
> __(Limit Point in Theorem 3).__ We agree. In fact, for all the theorems, our intention is to show stability other than stationarity. As the reviewer has mentioned, the use of PnP-denoiser corresponds to minimizing implicit objective which is not known and thus, in general only the fixed point convergence can be established. This is inherently the limitation of such a PnP framework, see, e.g., [Ryu et al., 2019, Nair et al., 2021 ,Liang et al., 2024,Sun et al., 202]. We have included such limitations in the updated pdf. However, we would like to mention that such a stability result or limit point result is not meaningless in terms of connecting to stationarity. For example, we noticed that the recent arXiv pre-print linked PnP’s limit points to KKT points when the denoisers are simple enough (e.g., linear denoisers) [Xu et al., 2025]. Such links have not been established for more complex PnP denoisers, to our knowledge. But we would like to argue that having a limit point convergence result as the first step towards more comprehensive convergence understanding is a meaningful progress.
>
> __(Boundedness Assumption in Theorem 3).__ We do agree that the boundedness assumption might be violated in some applications. However, boundedness is reasonable in some applications, e.g., MRI reconstruction, tomography, and natural image restoration problems. We have added a remark to clarify.
>
>
>
> __References__.
> [Ryu et al., 2019] Ryu, E., Liu, J., Wang, S., Chen, X., Wang, Z., & Yin, W. (2019, May). Plug-and-play methods provably converge with properly trained denoisers. In International Conference on Machine Learning (pp. 5546-5557). PMLR.
> [Nair et al., 2021] Nair, P., Gavaskar, R. G., & Chaudhury, K. N. (2021). Fixed-point and objective convergence of plug-and-play algorithms. IEEE Transactions on Computational Imaging, 7, 337-348.
> [Liang et al., 2024] Liang, W., Tu, Z., Lu, J., Tu, K., Ng, M. K., & Xu, C. (2024). Fixed-point convergence of multi-block PnP ADMM and its application to hyperspectral image restoration. IEEE Transactions on Computational Imaging.
> [Sun et al., 2021] Sun, Y., Wu, Z., Xu, X., Wohlberg, B., & Kamilov, U. S. (2021). Scalable plug-and-play ADMM with convergence guarantees. IEEE Transactions on Computational Imaging, 7, 849-863.
> [Xu et al., 2025] Xu, L., Cheng, L., Chen, J., Pu, W., & Fu, X. (2025). Radio Map Estimation via Latent Domain Plug-and-Play Denoising. arXiv preprint arXiv:2501.13472.

---

> > ### Comment · Reviewer_srVv · 2025-11-24
> >
> > Thank you for the clarification, especially for the "stationarity" assumption in Theorem 2. You have addressed some of my concerns, and I have increased my rating from 4 to 6.
> >
> > Regarding your response on “Limit Point in Theorem 3,” the claim that only fixed-point convergence can be established for PnP with complex denoisers, and that links to an objective are not known, is not fully accurate. There have been some prior works showing convergence to minimizers or stationary points of explicit (or implicit) objectives under specific classes of denoisers and algorithmic frameworks:
> >
> > - Hurault et al. [1] analyze PnP-HQS with a gradient-step denoiser (GS-DRUNet) and prove convergence to the minimum of an objective of HQS.
> > - Hurault et al. [2] introduce a proximal denoiser and show that the resulting PnP-DRS (equivalently, PnP-ADMM) converges to an objective via a Lagrangian, including nonconvex settings under suitable assumptions.
> > - Wei et al. [3] characterized cocoercive, conservative denoisers into proximal operators of weakly convex priors, and establish convergence of PnP-ADMM to an implicit objective in Poisson imaging inverse problems.
> >
> > These results do not cover diffusion-based or arbitrary black-box denoisers. However, they do provide convergence of PnP-ADMM to an implicit or explicit objective. These references may be worth mentioned. I encourage you to revise this part to accurately reflect the scope of existing theory and to cite [1–3] in the related-work and theory sections.
> >
> > References:
> >
> > [1] Hurault S, Leclaire A, Papadakis N. Gradient Step Denoiser for Convergent Plug-and-Play. ICLR 2022.
> >
> > [2] Hurault S, Leclaire A, Papadakis N. Proximal Denoiser for Convergent Plug-and-Play Optimization with Nonconvex Regularization. ICML 2022, PMLR: 9483–9505.
> >
> > [3] Wei D, Chen P, Xu H, et al. Learning Cocoercive Conservative Denoisers via Helmholtz Decomposition for Poisson Imaging Inverse Problems. NeurIPS 2025.

---

> > > ### Author Response · Authors · 2025-11-27
> > >
> > > We would like to thank the reviewer for adjusting their score and for the continued discussion on ADMM PnP convergence.
> > >
> > > We have now incorporated the references into “related works”. The references provided by the reviewer are interesting and relevant: they established convergence for a number of PnP denoisers, namely, gradient-step denoisers [1], the nonconvex counterpart of the gradient-step denoiser [2], and cocoercive conservative denoisers [3]. In particular, [1,2] associate their denoisers to an explicit regularizer (similar to [Xu et al 2025] did for linear denoisers) to show that PnP methods converge to minimizers (for convex cases) or stationary points (for nonconvex cases). The denoiser in [3] does not necessarily have explicit regularizers associated with them, but it still enjoys convergence to stationary points associated with an implicit objective.
> > >
> > > We have made the above clear in the updated theory section and related work section.
> > >
> > > __References:__.
> > > [1] Hurault S, Leclaire A, Papadakis N. Gradient Step Denoiser for Convergent Plug-and-Play. ICLR 2022.
> > > [2] Hurault S, Leclaire A, Papadakis N. Proximal Denoiser for Convergent Plug-and-Play Optimization with Nonconvex Regularization. ICML 2022, PMLR: 9483–9505.
> > > [3] Wei D, Chen P, Xu H, et al. Learning Cocoercive Conservative Denoisers via Helmholtz Decomposition for Poisson Imaging Inverse Problems. NeurIPS 2025.
> > > [Xu et al., 2025] Xu, L., Cheng, L., Chen, J., Pu, W., & Fu, X. (2025). Radio Map Estimation via Latent Domain Plug-and-Play Denoising. arXiv preprint arXiv:2501.13472.

---

### Official Review · Reviewer_JtYg · 2025-10-31

**Soundness:** 2
**Presentation:** 3
**Contribution:** 3
**Rating:** 4
**Confidence:** 4

**Summary:**

This paper proposes a novel Plug-and-Play (PnP) framework that integrates score-based denoisers into the ADMM optimization algorithm for solving inverse problems. The key contribution is a three-stage denoiser, termed AC-DC, designed to address the manifold mismatch problem: the fact that ADMM iterates (influenced by dual variables) do not lie on the noisy data manifolds on which the score functions were trained. The AC (Auto-Correction) stage adds Gaussian noise to the iterate, while the DC (Directional Correction) stage uses conditional Langevin dynamics to refine the iterate towards the correct noisy manifold before final score-based denoising. The authors provide a comprehensive convergence analysis, showing both fixed-point ball convergence under a constant step size and convergence under an adaptive step size schedule. Experiments across various inverse problems demonstrate improved performance over several baselines.

**Strengths:**

- The paper clearly identifies and addresses a significant, underexplored challenge in the PnP literature: the mismatch between the geometry of optimization iterates (especially in primal-dual methods like ADMM) and the manifolds on which score-based denoisers are trained. This is a pertinent and non-trivial issue.

- The proposed AC-DC denoiser is a novel and intuitive solution to the manifold mismatch problem. The idea of proactively "correcting" the iterate onto the score function's domain, rather than simply applying the denoiser, is innovative.

- The paper validates the method on a wide range of inverse problems (inpainting, deblurring, super-resolution, phase retrieval) and against a diverse set of modern baselines, showing consistent and often superior performance.

**Weaknesses:**

1. **Questionable Core Assumption**: A central weakness lies in the justification of the DC step. As outlined in the derivation leading to Eq. (10), the method assumes the residual s^(k) follows a Gaussian prior. This assumption appears to contradict the paper's primary motivation—that the noise in the ADMM iterate is not Gaussian and is off-manifold. If s^(k) can be reasonably modeled as Gaussian, the necessity of the complex DC correction is significantly undermined, as a simpler Gaussian denoiser might suffice. The paper lacks a compelling empirical or theoretical justification for this critical assumption.

2. **Lack of Clarity on Method Variants**: The distinction between the two presented variants, "Ours-tweedie" and "Ours-ode," is not clearly explained in the main text. The reader is left to infer the difference from the appendix, which is unsatisfactory. The core methodological description should explicitly state what these variants are.

3. **Insufficient Discussion of Related Work**: The proposed correction mechanism bears a conceptual resemblance to the Onsager correction in Denoising Approximate Message Passing (D-AMP) algorithms [1, 2], which also handles structured iteration noise. Furthermore, the nature of iteration noise in PnP algorithms has been explicitly studied in works like TFPnP [3] (Section 6.2 -- Iteration Noise of PnP Methods). The paper would be significantly strengthened by discussing these connections and clearly delineating how the proposed approach differs from or relates to these ideas.

4. **Inadequate Discussion of Computational Cost**: The AC-DC denoiser, particularly the DC step which involves multiple (J=10) Langevin steps, drastically increases the number of score function evaluations (NFE) per ADMM iteration compared to a standard PnP step. While the appendix includes an NFE analysis, the main text should frankly acknowledge this as a significant limitation and discuss the trade-off between performance and computational efficiency.

Given these issues, my initial rating is Borderline Reject. However, I would be inclined to raise my score if the authors can:
- Provide a convincing rationale for the Gaussian assumption in Equation (10),
- Clarify the differences between "Ours–tweedie" and "Ours–ode,"
- Thoroughly discuss relevant prior work (e.g., D-AMP and TFPnP), and
- Acknowledge and analyze the computational limitations of the AC-DC denoiser.

### References
[1] From Denoising to Compressed Sensing, TIT 2016
[2] Denoising AMP for MRI Reconstruction: BM3D-AMP-MRI, 2018
[3] TFPnP: Tuning-free Plug-and-Play Proximal Algorithms with Applications to Inverse Imaging Problems, JMLR 2022

**Questions:**

See above

---

> ### Author Response · Authors · 2025-11-20
>
> We appreciate the fair criticism, especially regarding the assumptions on the DC steps. We have revised them in the updated PDF. Below, we respond to your points in detail.
>
> __(Gaussian Assumption of Iterate Noise).__ This is a very valid point - and we agree. If the distribution of s is Gaussian, then using the DC step is not necessary. Here, let us clarify this step from two aspects, namely, practical implementation and convergence analysis.
>
> First, implementation. Note that the expression that guides our DC step is as follows $ \mathbf{z}^{(k)}\_{ac}=\tilde{\mathbf{z}}^{(k)} + \sigma^{(k)} \mathbf{n}\_{1}  =  \mathbf{z}\_{\sigma^{(k)}}  +  \sigma^{(k)}  ( \mathbf{n}\_{1} - \mathbf{n}\_{2}) + \mathbf{s}^{(k)} $
>
> Here, the ``noise’’ between the ac step and the dc target consists of two terms, i.e., $\sigma^{(k)} (\mathbf{n}\_1 -\mathbf{n}\_2)$ and $\mathbf{s}^{(k)}$. The first term is a Gaussian term coming from $\mathbf{z}_{\sigma^{(k)}}$ and the AC step. The second term $\mathbf{s}^{(k)}$ is from the optimization updates, whose distribution is unknown. Ideally, we would need the conditional PDF $p( \mathbf{z}^{(k)}\_{ac}|\mathbf{z}\_{\sigma^{(k)}})$ to perform the Langevin steps to reach the $\mathbf{z}\_{dc}$ that is on the target manifold. However, as the pdf of $\mathbf{s}^{(k)}$ is unknown, we could only approximate this conditional distribution. To be even more specific, the Gaussian approximation arises from the noisy manifold after the AC step, where the effective noise takes the form $\tilde{\mathbf{s}}^{(k)} = \sqrt{2} \sigma^{(k)} \mathbf{n} + \mathbf{s}^{(k)}$. Under mild conditions that $Var(\mathbf{s}^{(k)}) \ll(\sigma^{(k)})^{1/2}$ or when $\mathbf{s}^{(k)}$ is sub-Gaussian, the sum is well-approximated by a Gaussian random variable, enabling a local quadratic likelihood model for $p(\mathbf{z}\_{ac}^{(k)} | \mathbf{z}\_{\sigma^{(k)}})$). As such, if the first term $\sqrt{2}\sigma^{(k)} \mathbf{n}$ dominates the noise $\mathbf{s}^{(k)}$, using Gaussian is a reasonable approximation. The Gaussian approximation appears to be very effective to bring the noisy updates to iterates residing on Gaussian manifolds; see Appendix I.1 and Fig. 6. We have clarified this in the updated manuscript.
>
> Second, convergence analysis. We hope to clarify that the convergence analysis does not hinge on that $\mathbf{z}\_{dc}$ residing on the training manifolds of the diffusion model. As long as $\mathbf{z}\_{dc}$ is close to a stationary distribution that satisfies certain properties (see Theorem 4, Theorem 5), the process converges to a limit point (or a ball around it). Hence, the Gaussian approximation step does not damage limit point convergence. This is because our analysis focuses on the stability of the solution sequence and is not directly tied to the estimation accuracy of $\mathbf{x}$ (the latter is of course more ideal but current analytical tools lack such connections). We have articulated this in __Limitations in the updated version__.
>
> To give some empirical support of this DC step’s effectiveness and necessity. In addition to Fig. 5, we have added Appendix I.1 and Fig. 6. In addition, as demonstrated in Table 5, using only Gaussian denoising significantly underperforms our AC–DC method, attesting to the necessity of DC posterior correction.
>
>
> __(Clarification of the Nomenclature of Ours-Tweedie and Ours-ODE).__ We thank the reviewer for the suggestion. In the updated PDF, we have included brief details on what each variant is in Algorithm 1. We have also clarified them in “Proposed Approach”. These two variants only correspond to how we do the MMSE denoising part. Tweedie means that we directly apply the Tweedie’s lemma. ODE means we use a probability flow ODE to replace the Tweedie step. Both can be found in Algorithm 1 (line 8).
>
> __(Related Works).__ We appreciate the reviewer for providing us with these related works. We have included the discussion about these related work in the updated pdf. To be specific, [Metzler et al., 2016, Ekisoglu & Tanc, 2018] uses a denoiser based approximate message passing algorithm (D-AMP) that approximately Gaussianizes the residual via the Onsager correction under compressive sensing problem structures. On the other hand, [Wei et al., 2021] learns a reinforcement learning-based policy to automatically tune all internal parameters, including denoising strength. These are indeed interesting developments.
>
> Continued in the next comment.

---

> > ### Author Response · Authors · 2025-11-20
> >
> > __(Computational Limitations).__ As per the reviewer's feedback, we have included the limitation of our method, and a potential direction for further improvement. Indeed our method requires multiple score evaluation in a single iteration. On the other hand, we would like to remark that our method performs better than other methods under similar NFEs evaluation budgets. We have included Figs 10-15 in Appendix I.6 to demonstrate this comparison. Our understanding is that, although AC-DC’s per-iteration complexity is higher than methods that do not use DC steps, the DC steps do help the ADMM algorithm reduce the number of iterations needed to attain good performance. Hence, the ADMM AC-DC might end up with fewer iterations to run. We have added pointers in the main text to this discussion.
> >
> > __References__.
> > [Metzler et al., 2016] Metzler, Christopher A., Arian Maleki, and Richard G. Baraniuk. 2016. “From Denoising to Compressed Sensing.” doi:10.48550/arXiv.1406.4175.
> > [Ekisoglu & Tanc, 2018] Eksioglu, Ender M., and A. Korhan Tanc. 2018. “Denoising AMP for MRI Reconstruction: BM3D-AMP-MRI.” SIAM Journal on Imaging Sciences 11(3): 2090–2109. doi:10.1137/18M1169655.
> > [Wei et al., 2021] Wei, Kaixuan, Angelica Aviles-Rivero, Jingwei Liang, Ying Fu, Hua Huang, and Carola-Bibiane Schönlieb. 2021. “TFPnP: Tuning-Free Plug-and-Play Proximal Algorithm with Applications to Inverse Imaging Problems.” doi:10.48550/arXiv.2012.05703.

---

> > > ### Comment · Reviewer_JtYg · 2025-11-26
> > >
> > > Thanks for the response. I upgrade my score from 4 to 6.

---

> > > > ### Author Response · Authors · 2025-11-27
> > > >
> > > > We would like to thank the reviewer for adjusting their scores. We appreciate the time and effort put into reviewing this work.

---

### Official Review · Reviewer_uYpW · 2025-11-01

**Soundness:** 2
**Presentation:** 2
**Contribution:** 2
**Rating:** 4
**Confidence:** 2

**Summary:**

The paper proposes a new plug-and-play (PnP) ADMM method using score-based priors. Since score networks are trained on noisy manifolds, while ADMM iterates (due to dual variables) may not lie near those manifolds, score-based denoisers perform poorly, and the convergence theory is unclear. To address this, the paper proposes the AC–DC denoiser inside ADMM, consisting of three stages:
AC: add Gaussian noise to push iterates toward score training manifolds;
DC: short Langevin update guided by both the score and a quadratic prior; and
Final denoising via Tweedie score formula.
Under the strong convexity of the data fidelity term, the paper shows that the ADMM iteration is weakly nonexpansive and converges to a fixed point (up to a $\delta$ ball). Experiments on several inverse problems (inpainting, deblurring, SR, phase retrieval) show improvements over PnP baselines (DiffPIR, DDRM, DPS, RED-diff, DPIR, etc.).

**Strengths:**

- The paper addresses a real gap: score-based priors in ADMM are harder than proximal-gradient PnP, mainly due to dual variable noise geometry.
- The method conceptually simple: AC noise + short Langevin DC + score denoise.
- The paper extends PnP-ADMM convergence analysis to score models and Langevin steps.
- Broad experiments across common inverse problems show the practical utility of the method.

**Weaknesses:**

- Noise addition before score application already appears in recent PnP diffusion works (which the paper also acknowledges). In this respect, the novelty seems incremental.
- The DC step, a short Langevin refinement combining the score and a quadratic potential, structurally resembles diffusion posterior correction schemes, where noise injection is followed by brief score-based Langevin updates to pull iterates back toward the learned distribution. The paper’s approach integrates this idea into ADMM, but conceptually overlaps with prior noise-add-and-refine mechanisms.
- The AC and DC schedules appear heuristic. The issue of stability is unclear across tasks without tuning.
- Theorem 2 assumes that DC reaches the stationary distribution each iteration, which is unrealistic in practice with only a few Langevin steps.
- With strong convexity of the fidelity term, only convergence to a ball is ensured, not a point. Without strong convexity, an adaptive ADMM penalty schedule is required.
- The experiments use a pre-trained score model, so the advantages are partly dependent on that backbone.
- Many baselines are not ADMM-PnP, so the comparison may be apples-to-oranges from an optimization perspective.

**Questions:**

- The DC step convergence assumption seems very strong. In practice, you run a few steps; how sensitive is the method to DC iteration count?
- How is $\sigma^{(k)}$ chosen in the AC step? Is there a principled schedule, or is the choice purely empirical?
- Can the method diverge with fixed $\rho$ for non-convex cases? Any empirical failures?
- Can you provide some empirical evidence that dual variables distort score manifold geometry?
- Does AC–DC satisfy a consensus Equilibrium interpretation?

---

> ### Author Response · Authors · 2025-11-20
>
> We would like to thank the reviewer for the comments and questions - these are relevant questions that are worth clarifying. Here, we first respond the weakness part:
>
> __(Novelty)__ Please note that the novelty does not lie in adding noise before PnP diffusion. The research gaps in existing works are that, (1) no matter adding noise or not, the diffusion denoisers do not “project” the iterates back to the data manifold; and (2) existing diffusion PnP denoisers are hard to be combined with primal dual algorithms like ADMM and still enjoy convergence guarantees. Our work addresses the two gaps. We hope to highlight that integrating diffusion based PnP denoisers with ADMM is the most nontrivial part, as dual variables could make the denoising part rather hard (as the scores were not trained with unknown dual variables). The designed AC-DC denoiser mitigates the impact of dual variables significantly. Using diffusion PnP denoisers with ADMM is rather well motivated, as ADMM can flexibly use various regularization (traditional regularizers like L1, learned regularizers like perception based regularizer, and implicit regularizers like PnP denoisers) simultaneously, as we demonstrated in Appendix I.3 and Fig. 9 of the updated manuscript. Before our design, the existing diffusion PnP denoisers did not work well with ADMM, due to lack of considerations of dual variables.
>
> __(Convergence, Langeven Stationarity)__ The reviewer has made a fair point that assuming the DC step runs to reach stationarity is a bit strong. We made the assumption at the time to avoid introducing complex notations in analysis. Here we hope to clarify that i) in practice, we always only run 10 Langevin steps for the DC step, and ii) we have revised Theorem 2 (and Theorem 3) resulting Theorem 4 (and Theorem 5) to include finite step analysis of the DC step.
>
> __(Convergence results)__ Indeed, convergence to a ball or convergence with adaptive stepsize are both compromises compared to standard results in the traditional optimization convergence sense. However, we argue that the incorporation of learned scores in the optimization process is non-traditional by its nature. Studying the convergence of such a procedure under the primal-dual setting itself is quite interesting to us. Our result revealed that there is at least some degree of convergence that can be guaranteed if the denoisers are used properly. Such a revelation is interesting, in our humble opinion.
>
> __(Pre-trained Score, Baselines not ADMM-PnP)__ Please note that our focus is exactly using pre-trained score to solve inverse problems. The setting is as in [Chung et al., 2023, Zhang et al., 2024] and is a well-motivated setting. All baselines use the same pre-trained score. The reason why we did not find other ADMM-PnP baselines using diffusion is perhaps because diffusion based denoisers are hard to combine with ADMM, as we discussed above. This is exactly the gap that we address.
>
> Continued in the next comment.

---

> ### Author Response · Authors · 2025-11-20
>
> To answer your questions:
>
> __Q1 (Convergence of DC).__ As we mentioned, in practice, we only run 10 steps. We have also revised the theorem and proof by assuming finite steps of DC Langevin.
>
> __Q2 (Scheduling of $\sigma^{(k)}$).__ We used a principle-driven heuristic design. We choose a linear scheduling of $\sigma^{(k)}$ that decays from $10$ to $0.1$ over $W$ window. At iteration $k$, $\sigma^{(k)}=\max(0.1, 10 - (10-0.1) \cdot k/W)$. This design is based on the principle that the noise level typically decays over iterations so does the denoising strength, and similar annealing schedules are used in diffusion sampling as well [Karras et al., 2022]. And empirically, it works across all datasets without fine tuning.
>
> __Q3 (Divergence).__ No. With our hyperparameter settings (see Appendix H), we didn't see any divergence cases with fixed $\rho$ even for non-convex cases. Please note that we had a discussion on this and a potential future improvement in the conclusion and __limitations__ part.
>
> __Q4 (Illustrating Noisy Iterates).__ Good point and thanks for the suggestion. In the updated manuscript, we have illustrated a sample of noisy image obtained during the ADMM iteration in Appendix I.1. One can see that the noise present in such noisy iterates $\tilde{\mathbf{z}}^{(k)}$ is very different from Gaussian noise - which means that the iterate is not near any training manifold of the diffusion score. Hence, our method first corrects this mismatch and then denoises resulting in effective denoising (see illustration in Figure 6).
>
> __Q5 (Consensus Equilibrium).__ AC-DC does satisfy approximate Consensus Equilibrium. Our theoretical results show that our AC-DC PnP-ADMM converges to a ball of radius $r$ for some constant $r>0$. This means that for any $\delta>0$ there exists a constant $K_1>0$ such that for $ || \mathbf{x}^{(k)}-\mathbf{x}^* ||_2 \le r + \delta $   and $||\mathbf{u}^{(k)}-\mathbf{u}^*||_2 \le r+\delta$ all $k>K_1$. From equation 7c), we have $||\mathbf{x}^{(k+1)}-\mathbf{z}^{(k+1)}||_2 = ||\mathbf{u}^{(k+1)}-\mathbf{u}^{(k)}||_2 \le 2r + 2\delta$ for all $k>K_1$. This means there is approximate consensus equilibrium among $\mathbf{x}$ and $\mathbf{z}$ within radius $2r$.
>
> __References__.
> [Karras et al., 2022] Karras, T., Aittala, M., Aila, T., & Laine, S. (2022). Elucidating the design space of diffusion-based generative models. Advances in neural information processing systems, 35, 26565-26577.
> [Chung et al., 2023] Chung, H., Kim, J., Mccann, M. T., Klasky, M. L., & Ye, J. C. (2022). Diffusion posterior sampling for general noisy inverse problems. arXiv preprint arXiv:2209.14687.
> [Zhang et al.,] Zhang, B., Chu, W., Berner, J., Meng, C., Anandkumar, A., & Song, Y. (2025). Improving diffusion inverse problem solving with decoupled noise annealing. In Proceedings of the Computer Vision and Pattern Recognition Conference (pp. 20895-20905).

---

### Author Response · Authors · 2025-12-01
**Report to the Area Chair (AC)**

We would like to thank the AC for using their extra time to review the submissions and responses. Here, we briefly summarize the rebuttal process before the reverting.

## Score Changes During Rebuttal
1. __Reviewer JtYg__ (confidence=4): Score changed 4→__6__ after rebuttal (Our action: the reviewer's concerns about Gaussian assumption, related work, and computational cost were addressed).

2. __Reviewer srVv__ (confidence=4): Score changed 4→__6__ after rebuttal (Our action: the reviewer’s concerns on stochasticity and Theorem assumptions were resolved with finite-step analysis + added references in the related works.)

3. __Reviewer zbpZ__ (confidence=4): Score changed 6→__8__ after rebuttal (Our action: the reviewer’s concerns about the assumptions were resolved with finite-step analysis. In addition, the question regarding experimental validation of the coercivity and smoothness assumption + questions about computational complexity was addressed.)

4. __Reviewer uYpW__ (confidence=2): The __reviewer didn’t respond__ to our comments during the discussion period. (Our action: The novelty of the work,  questions on experimental setup, and concerns on assumptions with finite-step analysis were all addressed.)

In short, our original scores were 4464. After rebuttal (before reverting), the __scores were changed to 6684__, where the 4 was given by uYpW, who has confidence of 2 and never responded.

More detailed rebuttal summary is as follows:

## Reviewer Concerns and Responses

1. __Novelty, sensitivity of DC, and convergence interpretation (Reviewer uYpW)__
We clarified that the core novelty is enabling diffusion priors within ADMM by correcting dual-variable manifold mismatch—an issue not addressed in prior PnP diffusion methods. We showcased the flexibility of the ADMM framework to naturally handle additional regularization (Appendix I.3 / Fig. 9). We added finite-step Langevin convergence results replacing the original stationarity assumption of the DC, explained principled $\sigma^{(k)}$ scheduling, and demonstrated noisy-iterate mismatch (Appendix I.1 / Fig. 6). We also explained the connection with our results to approximate consensus-equilibrium.
__Outcome__: Reviewer didn’t respond and kept score 4 confidence 2.


2. __Gaussian approximation validity, unclear Tweedie vs. ODE variants, missing related work, and computational overhead (Reviewer JtYg)__.
We justified the Gaussian likelihood approximation after AC using bounded noise variance and sub-Gaussian residual arguments, and provided empirical evidence demonstrating its efficacy (Appendix I.1 / Fig. 6). We clarified Tweedie vs. ODE variants in Algorithm 1 and main text, added comparison to D-AMP [1,2] and TFPnP [3], and acknowledged computational cost while showing NFE-fair comparisons where AC-DC achieves superior accuracy per evaluation.
__Outcome:__ Reviewer satisfied and score upgraded 4→__6__.

3. __Stochastic vs. deterministic treatment in theory, strong assumptions, and interpretability of the limit point (Reviewer srVv)__.
We emphasized that our convergence analysis already takes account of the stochastic nature of the denoiser and provides the high-probability convergence results.  Finite-step DC proofs were added replacing its stronger convergence to stationary distribution assumption. We clarified the theoretical scope between strongly convex (Theorem 1) and non-convex settings (Theorem 3 with adaptive $\rho$), and expanded discussion on special PnP denoisers that result in  explicit/implicit objectives with references to recent theory.
__Outcome__: Reviewer satisfied and score upgraded 4→__6__. The reviewer followed up by asking, including some discussions of related works. We have included such discussion. But then the system was locked and scores were reverted.

4. __Strength and validity of assumptions and complexity (Reviewer zbpZ)__.
We replaced stationarity with finite-step convergence bounds (Appendix E.2), validated coercivity and smoothness assumptions experimentally (Appendix I.2), and clarified that although per-iteration cost increases, DC significantly reduces total ADMM iterations and yields better results under equal number of function evaluations (NFE) budgets.
__Outcome__: Reviewer satisfied and score upgraded 6→__8__.

## Overall Impact of the Rebuttal.
The rebuttal __resolved all major criticisms__, leading to __satisfaction and score upgrades among all three responded reviewers (one low-confidence reviewer didn’t respond)__ and __stronger theoretical & empirical clarity__. The updated version of the paper improves consensus among the reviewers and significantly strengthens the paper’s standing.

## References.
[1] From Denoising to Compressed Sensing, TIT 2016.
[2] Denoising AMP for MRI Reconstruction: BM3D-AMP-MRI, 2018.
[3] TFPnP: Tuning-free Plug-and-Play Proximal Algorithms with Applications to Inverse Imaging Problems, JMLR 2022

---

### Meta-Review · Area_Chair_nf3D · 2026-01-06

**Summary:**

The manifold mismatch issue exists when score-based generative models are directly integrated into optimization algorithms such as ADMM. In this paper, the authors propose ADMM plug-and-play (ADMM-PnP) with the AC-DC denoiser, a new framework that embeds a three-stage denoiser, including (1) auto-correction (AC) via additive Gaussian noise, (2) directional correction (DC) using conditional Langevin dynamics, and (3) score-based denoising, into ADMM.

The concerns of assumptions and computation cost from Reviewers JtYg, zbpZ, and srVv have been addressed and the corresponding scores were increased.
Reviewer uYpW with confidence=2 didn’t respond to authors' responses to the comments on the novelty of this work, experimental setup, and assumptions with finite-step analysis, which were reasonably  addressed by the authors.
Taking the above into account, the AC would suggest to accept this paper.

**Reviewer Scores:**

none

---

### Decision · Program_Chairs · 2026-01-26

Accept (Poster)